# *Cis*-aconitate therapy protects against influenza mortality by dual targeting of viral polymerase and ERK/AKT/NF-κB signaling

Adeline Cezard[1,2], Déborah Brea-Diakite[1,2], Virginie Vasseur[1,2], Alan Wacquiez [1,2], Loic Gonzalez [1,2], Ronan Le Goffic[3], Bruno Da Costa[3], Ambre Tinard [1,2], Delphine Fouquenet [1,2], Séverine Heumel[4], Arnaud Machelart[4], Eik Hoffmann [4], Priscille Brodin [4], François Trottein[4], Cyrille Mathieu [5], Lola Canus[5], Florentine Jacolin[5], Pierre-Olivier Vidalain[5], Laure Perrin-Cocon [5], Vincent Lotteau[5], Julien Burlaud-Gaillard[6], Dominique Tertigas[7], Michael G Surette[8], Antoine Legras[2,9], Damien Sizaret[10], Thomas Baranek [1,2], Christophe Paget [1,2], Antoine Guillon[1,2,11] & Mustapha Si-Tahar [1,2✉]

## Abstract

The influenza virus poses a significant global health challenge, causing approximately 500,000 deaths annually. Its ability to evade antiviral treatments and vaccine-induced immunity underscores the need for novel therapeutic approaches. Our study identifies *cis*-aconitate (*cis*-aco), a mitochondria-derived metabolite, as a potent dual-action agent against influenza, independently of its metabolic derivative, itaconate. *Cis*-aco impairs viral polymerase activity, resulting in decreased viral mRNA expression and protein synthesis, as observed for the influenza A/Scotland/20/74 (H3N2) strain. This antiviral effect was further confirmed across multiple influenza A and B strains, as well as in ex vivo human airway and lung organotypic models. Beyond its antiviral properties, *cis*-aco exhibits potent anti-inflammatory effects, disrupting key inflammatory cascades and reducing the secretion of inflammatory mediators. In a mouse model of influenza pneumonia, *cis*-aco mitigates viral replication, inflammation, and immune cell activation, significantly improving survival. Notably, its efficacy persists even when administered at later stages of infection, when oseltamivir/Tamiflu® is no longer effective. These findings position *cis*-aco as a promising influenza treatment, combining antiviral and anti-inflammatory benefits within a clinically relevant timeframe.

**Keywords** Influenza Virus; Therapy; Antiviral; Anti-inflammatory; Pneumonia
**Subject Categories** Metabolism; Microbiology, Virology & Host Pathogen Interaction; Pharmacology & Drug Discovery

See also: C Claus & I Kovacevic

## Introduction

Influenza viruses have long been major causes of morbidity and mortality, with heightened attention since the 1918 pandemic, driving extensive research into therapies (Centers for Disease Control and Prevention, National Center for Immunization and Respiratory Diseases (NCIRD), 2021). Current approaches, including vaccination and antivirals, often demonstrate limited effectiveness. The short duration of vaccine-induced immunity, combined with the intrinsic antigenic drift of influenza viruses, undermines sustained protection (World Health Organization, 2010). Skepticism also persists regarding the efficacy of approved anti-influenza drugs, especially when administered later in the course of infection (Aliberti et al, 2021; Jefferson et al, 2014; Gao et al, 2024). Therefore, developing innovative strategies effective even after infection onset is crucial.

The pathophysiology of influenza-related pneumonia stems from the intrinsic viral pathogenicity and the immune response. While a robust immune response is essential for viral clearance, excessive cellular recruitment and the release of cytotoxic molecules can lead to lung hyperinflammation, resulting in tissue damage, morbidity, and death (Herold et al, 2015; Tavares et al, 2017).

[1]INSERM, Centre d'Etude des Pathologies Respiratoires (CEPR), UMR 1100, Tours, France. [2]Université de Tours, Tours, France. [3]Université Paris-Saclay, INRAE, UVSQ, UMR892 VIM, Jouy-en-Josas, France. [4]Université de Lille, CNRS, INSERM, CHU Lille, Institut Pasteur de Lille, U1019 - UMR 9017 - CIIL - Center for Infection and Immunity of Lille, Lille F-59000, France. [5]Centre International de Recherche en Infectiologie (CIRI), Université de Lyon, Inserm, U1111, CNRS, UMR5308, Université Claude Bernard Lyon 1, Ecole Normale Supérieure de Lyon, Lyon 69007, France. [6]Plate-Forme IBiSA des Microscopies, PPFASB, Université de Tours and CHRU de Tours, Tours, France. [7]Department of Biochemistry and Biomedical Sciences, McMaster University, Hamilton, ON, Canada. [8]Department of Medicine, McMaster University, Hamilton, ON, Canada. [9]Service de Chirurgie Thoracique, CHRU de Tours, Tours, France. [10]Service de Pathologie, CHRU de Tours, Tours, France. [11]Service de Médecine Intensive -Réanimation, CHRU de Tours, Tours, France. ✉E-mail: mustapha.si-tahar@inserm.fr

Interestingly, the recent discovery of metabolic reprogramming of immune cells has opened avenues for innovative therapeutic approaches (O'Neill et al, 2016; Pearce and Pearce, 2018; Rambold and Pearce, 2018; Rao et al, 2019). We and others have demonstrated that hosts develop metabolic countermeasures in response to infection. (Guillon et al, 2020; O'Neill et al, 2016; Pålsson-McDermott and O'Neill, 2020; Guillon et al, 2022; Martínez-Reyes and Chandel, 2020; Soto-Heredero et al, 2020). For instance, using an integrated approach combining metabolomics, in vitro, and in vivo infection assays, we recently discovered the inhibitory effect of tricarboxylic acid cycle (TCA)-derived succinate on influenza virus infection. This inhibition is primarily associated with succinylation and nuclear retention of the viral nucleoprotein, although additional mechanisms may also contribute (Guillon et al, 2022).

Building on this finding, our investigation explored a broader range of host metabolites for their ability to regulate influenza virus infection in human lung epithelial cells. Among the TCA intermediates examined, cis-aconitate (cis-aco) stood out for its potent antiviral and anti-inflammatory properties. We further elucidated the molecular mechanisms underlying cis-aco's protective effect, and our in vivo experiments confirmed its efficacy in mitigating severe influenza pneumonia. Remarkably, cis-aco demonstrated therapeutic benefits even when administered at advanced stages of infection, when conventional treatment with oseltamivir (Tamiflu® (Davidson, 2018)) proved ineffective.

# Results

## Cis-aco inhibits both viral replication and production of inflammatory mediators in influenza virus-infected lung epithelial cells

Building on our previous work examining the anti-influenza effects of succinate (Guillon et al, 2022), we selected eight metabolites derived from glycolysis or the TCA cycle for evaluation. Human bronchial epithelial BEAS-2B cells were infected with influenza A virus (IAV; A/Scotland/20/74, H3N2) and treated with 3.4 mM of each metabolite at 4 hours (h) post-infection (p.i.) (Fig. 1A). This concentration was chosen as the highest dose without cytotoxic effects across all tested metabolites (Fig. EV1A,B). Notably, cis-aco treatment at this concentration had no significant impact on cell proliferation, mitochondrial mass, or ROS production (Fig. EV1C–E).

To assess the release of neo-virions, we measured neuraminidase (NA) activity in cell supernatants at 20 h p.i. Treatment with cis-aco resulted in a 60-fold reduction in NA activity ($P = 0.0005$), while itaconate treatment led to a smaller, yet significant, 3.8-fold reduction ($P = 0.0420$) (Fig. 1B). No significant changes in NA activity were observed with other metabolites. Concomitantly, cis-aco treatment also reduced the levels of the inflammatory cytokine interleukin-6 (IL-6) in IAV-infected epithelial cells ($P = 0.0098$) (Fig. 1C). We next evaluated the expression of a broader panel of pro-inflammatory cytokines and chemokines, including CCL2/MCP-1, CCL5/RANTES, CXCL1/GROα, CXCL10/IP-10, IL-6, and CXCL8/IL-8. In IAV-infected cells, the levels of these mediators were increased by four- to sevenfold compared to MOCK conditions (Fig. 1D, pink bars). Cis-aco treatment inhibited this inflammatory response by at least threefold (Fig. 1D, blue bars). Of

note, trans-aconitate exhibited similar antiviral and anti-inflammatory activities (Fig. EV2A–D), suggesting that both cis- and trans-isomers of aconitate possess anti-influenza properties.

Both influenza A and B viruses contribute to seasonal flu epidemics, with influenza A encompassing a diverse range of subtypes (Centers for Disease Control and Prevention, National Center for Immunization and Respiratory Diseases (NCIRD) 2025). To assess the broad-spectrum antiviral potential of cis-aco, we evaluated its activity against several influenza strains: influenza A/pandemic/2009 H1N1 (Fig. 1E), influenza A/Puerto Rico/8/1934 H1N1 (Fig. 1F), influenza B Yamagata strain B/Paris/234/2013 (Fig. 1G), in comparison with our "reference" strain influenza A/Scotland/20/74 H3N2 (Fig. 1H). Plaque-forming unit assays revealed a significant decrease in viral particle production in cells treated with cis-aco, irrespective of influenza virus type or subtype ($P < 0.05$; Fig. 1E–H).

The foregoing results were obtained using the immortalized bronchial epithelial BEAS-2B cell line, which can exhibit altered functional and metabolic profiles (Hughes et al, 2007; Geraghty et al, 2014). To better mimic lung physiology, we tested cis-aco in 2D cultures of primary bronchial epithelial cells (PBEC) isolated from non-cancerous bronchial tissue of lung cancer patients undergoing lobectomy. In these cells, cis-aco treatment significantly reduced IAV release in a dose-dependent manner ($P = 0.047$; Fig. 1I) and inhibited IAV-induced IL-6 production by up to 80% (Fig. 1J). Notably, it reduced viral titers of the Influenza B Victoria strain by more than 1 log ($P = 0.0313$; Fig. 1K). It also markedly decreased IAV-induced cell death by ~70%, as shown by SYTOX™ labeling (Fig. EV3A,B). While 2D primary bronchial epithelial cultures are closer to in vivo conditions than immortalized cell lines, they still lack the complexity of native tissues. To address this, we developed a 3D culture model using primary airway epithelial cells grown at an air–liquid interface (ALI), forming a differentiated, polarized epithelium with basal, goblet, and ciliated cells (Fig. 2A). This model exhibited a transepithelial electrical resistance of ~400 ohms/cm² and showed beating cilia (not shown). In this more complex system, cis-aco again demonstrated antiviral and anti-inflammatory effects, confirming its efficacy in a model that more closely mimics the in vivo environment (Fig. 2B–D).

To further model human lung physiopathology, we developed a standardized ex vivo lung organotypic slice culture at the ALI (Fig. 2E). Tissue explants were obtained from patients undergoing surgery for lung cancer (Table 1), predominantly male (6/8), with an average age of 73 [68–78] years and a history of smoking (6/8). Upon tissue infection, IAV localized to restricted areas (Fig. 2F), and infectious particles were detected in culture supernatants (Fig. 2G). Cis-aco treatment reduced viral titers by 95% (Fig. 2G).

## Cis-aco confers potent benefits in a murine model of influenza infection

Before evaluating the potential anti-influenza effects of cis-aco in vivo, we assessed its safety profile using repeated instillations in mice over 15 days. Cis-aco did not significantly affect body weight in either male or female mice compared to controls (Fig. EV4A). No changes in liver function were observed, as measured by serum ALAT levels (Fig. EV4B). Similarly, no alterations in fecal microbiota composition were detected between cis-aco-treated and control mice (Fig. EV4C). Flow cytometry analysis of immune

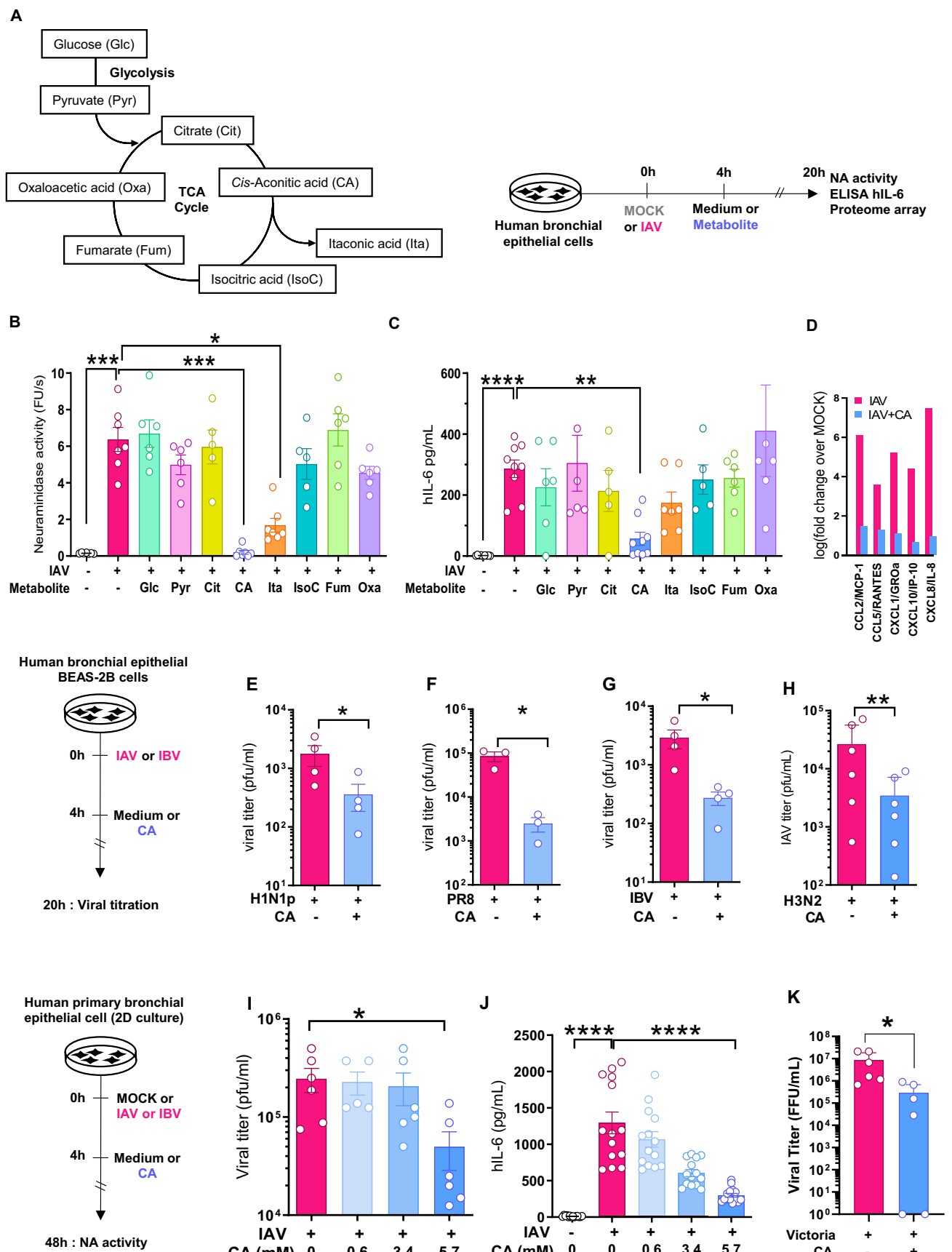

**Figure 1.  Anti-influenza properties of *cis*-aco among glycolysis and TCA cycle metabolites.**

(A–D) Human bronchial epithelial (BEAS-2B) cells were infected with influenza A/Scotland/20/74 (H3N2) virus at MOI = 1 (IAV) or left uninfected (MOCK) for 4 h, then treated (Metabolite) or untreated (Medium) with 3.4 mM of TCA cycle or glycolysis metabolites for 16 h. (A) Metabolites tested: *cis*-aco (CA), itaconic acid (Ita), oxaloacetic acid (Oxa), isocitric acid (IsoC), citrate (Cit), fumarate (Fum), pyruvate (Pyr), or glucose (Glc)). (B) Viral particle production assessed by neuraminidase activity assay. (C) hIL-6 levels in cell supernatants measured by ELISA. (D) Immune mediator levels in cell supernatants determined using a protein array. Pink bars represent fold changes in mediator secretion induced by IAV infection, and blue bars represent fold changes in IAV + CA conditions relative to MOCK. (E–H) BEAS-2B cells were infected with: (E) influenza A/pandemic/2009 H1N1 (H1N1p) strain, (F) A/Puerto Rico/8/1934 H1N1 (PR8) virus, (G) influenza B Yamagata (B/Paris/234/2013) virus (IBV), or (H) A/Scotland/20/74 (H3N2) strain. At 4 h p.i., cells were washed and treated or not for 16 h with 3.4 mM of *cis*-aco (CA). Production of infectious viral particles in cell supernatants was quantified by plaque-forming unit assay. (I, J) PBEC in two-dimensional (2D) liquid culture were infected with influenza A/Scotland/20/74 (H3N2) virus at MOI = 1 for 4 h, then treated or not with varying concentrations of *cis*-aco (CA) for 44 h. At 48 h p.i., viral titer (I) and hIL-6 levels (J) were measured in cell supernatants to assess viral particle production and pro-inflammatory cytokine release. (K) PBEC in two-dimensional (2D) liquid culture were infected with influenza B Victoria (B/Bretagne) virus at MOI = 1 for 4 h, then treated or not with 3.4 mM of *cis*-aco (CA) for 44 h. At 48 h p.i., viral titer was measured in cell supernatants to assess viral particle production. Data are presented as the mean ± SEM. Results represent cumulative data from 1 (D), 7 (B), 3 (G), or 4 (C, E, F, H, I, J) independent experiments. PBEC data (I–K) are shown as the mean ± SEM from single or duplicate samples from four independent individuals. The number of data points shown in each bar plot corresponds to the number of independent experiments performed for that condition. Statistical analyses were performed using the Kruskal–Wallis test with Dunn's multiple comparison test (I, J) or ratio paired *t* test (E, F, G, H), or the Wilcoxon matched-pairs signed rank test (K). Statistical significance: *$P < 0.05$, **$P < 0.01$, ***$P < 0.001$, ****$P < 0.0001$. Source data are available online for this figure.

cells in BAL fluid (Fig. EV4D) and blood (Fig. EV4E) confirmed that *cis*-aco had no impact on pulmonary or systemic immune responses, regardless of sex.

By mitigating various aspects of influenza pathogenesis, the preceding in vitro and ex vivo data (Figs. 1 and 2) suggest that *cis*-aco could prevent lung damage induced by IAV in vivo. Therefore, to assess its potential as an anti-influenza drug, we evaluated its effect on the mortality of infected mice, comparing it to the standard of care, oseltamivir (a neuraminidase inhibitor (Davidson, 2018)). While all untreated IAV-infected mice succumbed, those treated with 30 mg/kg *cis*-aco or 20 mg/kg oseltamivir within 20 min p.i. reached survival rates of 80% and 90%, respectively (Fig. 3A). Moreover, by 15 days post-treatment, surviving mice had regained their original body weight (Fig. 3B). However, delaying oseltamivir administration at 2 days p.i. abolished its efficacy (Fig. 3C,D), whereas *cis*-aco retained its curative effect under the same conditions in both female C57Bl/6 (Fig. 3C,D) and male BALB/c mice (Fig. EV5A,B).

## In vivo assessment of pathophysiological mechanisms modulated by *cis*-aco in influenza pneumonia

To better understand how *cis*-aco achieves its protective effect on survival, we next explored its impact on the pathophysiological responses associated with influenza infection (Fig. 4). Mice were intranasally infected with a lethal dose of influenza A/Scotland/20/1974 (H3N2) and treated with *cis*-aco 2 days p.i. Four days p.i., some mice were euthanized to analyze early events in influenza pathophysiology. *Cis*-aco treatment resulted in a significant reduction in viral load in lung tissues, with a 1-log decrease compared to controls ($P < 0.0001$; Fig. 4A). In addition, the relative expression of 50 mediators in the BAL fluids—including pro-inflammatory cytokines, interferons, growth factors, and proteases—was reduced in *cis*-aco-treated animals (Fig. 4B). Among these, the most pronounced reductions were observed in CCL2/MCP-1, C1qR1, G-CSF, CCL17/Tarc, and CCL20/MIP3α, while levels of resistin, myeloperoxidase, pentraxin 2/serum amyloid P, CXCL-10/IP-10, and CXCL9/Mig remained largely unchanged.

In a separate series of experiments, mice were sacrificed 8 days p.i. to assess lung injury caused by IAV. Severe pathology was characterized by increased leukocyte infiltration, including

neutrophils and NKT cells. While NK cell counts remained unchanged, their activation, as indicated by CD69 expression, was markedly increased following IAV infection. NKT cells similarly showed enhanced activation, whereas SiglecF+ alveolar macrophage numbers were significantly reduced (Fig. 4C). Remarkably, *cis*-aco treatment attenuated these immune perturbations, significantly limiting neutrophil and NKT cell infiltration while preventing the loss of alveolar macrophages ($P = 0.001$, $P = 0.01$, and $P = 0.05$, respectively; Fig. 4C). In addition, NK, NKT, and macrophage activation levels were all substantially lower in *cis*-aco–treated mice compared with infected, untreated controls ($P = 0.01$; Fig. 4C).

Histopathological analysis revealed significant reductions in alveolar wall thickening, hyaline membrane formation, epithelial necrosis, and leukocyte infiltration in IAV-infected mice treated with *cis*-aco compared to untreated animals (Fig. 4D). Quantitative assessment confirmed that *cis*-aco significantly alleviated tissue damage caused by IAV infection ($P < 0.01$; Fig. 4E). To further evaluate IAV-triggered inflammation on a broader scale, we measured NF-κB activity in NF-κB-luciferase transgenic mice infected with IAV and treated or not with *cis*-aco (Fig. 4F,G). By day 8 p.i., both airway and systemic inflammation were visible (Fig. 4F, central picture). In contrast, NF-κB activation was significantly reduced in *cis*-aco-treated animals ($P < 0.03$; Fig. 4F, right picture, and 4G). Overall, *cis*-aco suppressed critical aspects of influenza pathogenesis, including viral replication, inflammatory signaling, secretion of inflammatory mediators, and immune cell infiltration.

## Mechanistic insights into the anti-influenza action of *cis*-aco in inhibiting influenza virus replication and inflammatory signaling

To gain molecular insights into how *cis*-aco inhibits the production of viral particles, we analyzed its effects on various stages of IAV replication cycle. IAV is an enveloped virus with a genome made up of negative sense, single-stranded RNA and its life cycle comprises the following stages: (i) entry into the host cell; (ii) nuclear import of ribonucleoproteins (vRNP); (iii) transcription of the viral genome; (iv) replication and translation of viral proteins; (v) nuclear export of vRNP; and (vi) assembly and budding at the host cell plasma membrane.

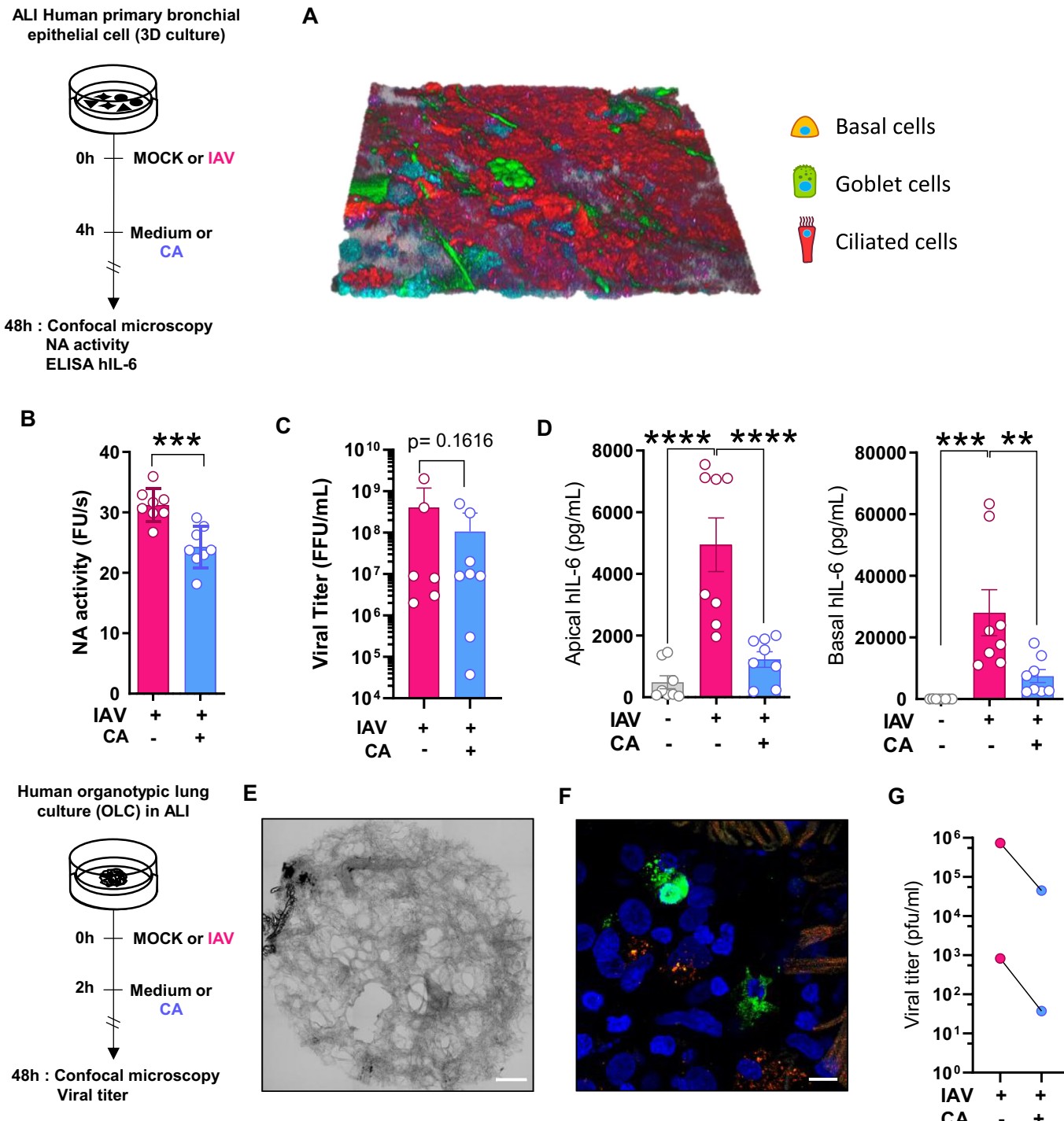

## Impact of cis-aco on virus budding and viral protein expression

Transmission and scanning electron microscopy (TEM and SEM) of IAV-infected bronchial epithelial cells revealed a marked reduction in virus budding following *cis*-aco treatment (Fig. 5A). To determine whether this was due to impaired budding or to reduced viral material production, we quantified viral proteins and mRNA in cells, treated or not with *cis*-aco 4 h p.i. Confocal microscopy (Fig. 5B,C) and western blotting (Fig. 5D,E) revealed

significant reductions (~75%, $P < 0.03$, Fig. 5E) in NP, NS1, M1, and PA protein expression at 8 h p.i. while qRT-PCR showed a tenfold decrease in viral NP and M1 mRNA levels at 6 h p.i. ($P < 0.03$, Fig. 5F). Given that *cis*-aco was applied at 4 h p.i., it is unlikely that it affects the early stages of viral entry, including cell fusion and nuclear trafficking (completed within 1 h p.i. (Dou et al, 2018)). These findings suggest that *cis*-aco inhibits IAV genome transcription, leading to reduced viral protein expression and neo-

**Figure 2.  *Cis*-aco demonstrates antiviral and anti-inflammatory properties in human primary bronchial epithelial cells (PBEC) and human organotypic lung cultures (OLCs) under air–liquid interface (ALI) conditions.**

(A–D) PBEC in three-dimensional (3D) culture were infected with influenza A/Scotland/20/74 (H3N2) virus at an MOI = 1 for 4 h, followed by treatment with or without 5.7 mM *cis*-aco (CA) for 44 h. (A) At 48 h p.i., tissue composition was validated using 3D confocal microscopy. Basal cells (yellow), goblet cells (green), ciliated cells (red), and nuclei (blue) were labeled to confirm tissue structure. (B) Neuraminidase activity, (C) viral titer, and (D) hIL-6 levels were measured in apical or basal cell supernatants. (E–G) OLC were infected with 2 × 10⁴ pfu of influenza A/Scotland/20/74 (H3N2) virus (IAV) and treated or not with 3.4 mM of *cis*-aco at 2 h p.i. (E) Alveolar structures were visualized using light microscopy (scale bar: 500 µm). (F) IAV infection was confirmed *via* confocal microscopy, with viral nucleoprotein (green), nuclei (blue), and α-tubulin (red) labeled at 48 h p.i. (scale bar: 10 µm). (G) Viral titers were measured in OLC supernatants at 48 h p.i. Data are presented as the mean ± SEM. PBEC data (A: microscopy) represent duplicates from three independent patients analyzed in a single experiment. Panels (B–D) represent four independent experiments. The number of data points shown in each bar plot corresponds to the number of independent experiments performed for that condition. Representative images of OLC from two patients are shown in (E, F), and viral titers in (G) were measured using pooled supernatants from five OLC derived from two patients. Statistical analyses were conducted using the unpaired *T* test (B, C) or ordinary one-way ANOVA test (D). Statistical significance: *$P < 0.05$, **$P < 0.01$, ***$P < 0.001$, ****$P < 0.0001$. Source data are available online for this figure.

**Table 1.  Patients' characteristics for OLC preparations.**

| Patients' characteristics | ($n = 8$) |
|---|---|
| Male sex, n (%) | 6 (87) |
| Age (year), median [IQR] | 73 [68–78] |
| Body Weight (kg), median [IQR] | 74 [58–95] |
| Body Mass Index (kg/m²), median [IQR] | 25 [19–29] |
| Active or former smokers, n (%) | 6 (75) |
| Pak-year, n (%) | 42 [39–53] |
| FEV1% predicted, median [IQR] | 92 [82–101] |
| FEV1/FVC ratio, median [IQR] | 81 [69–100] |
| COPD, n (%) | 1 (13) |

*IQR* interquartile range, *FEV1* forced expiratory volume in 1 second, *FVC* forced vital capacity, *COPD* chronic obstructive pulmonary disease.

virion production. This hypothesis was confirmed by a minigenome assay in HEK293T cells. The minigenome assay measures IAV polymerase activity without using infectious virus, thereby avoiding the cytopathic effects of infection. HEK293T cells are transfected with plasmids encoding the polymerase complex (PB1, PB2, PA) and nucleoprotein (NP), along with a luciferase reporter minigenome, so that luciferase expression directly reflects polymerase activity (Te Velthuis et al, 2018). Treatment with *cis*-aco reduced luciferase activity by ~75% ($P < 0.03$, Fig. 5G), indicating inhibition of IAV polymerase activity.

### Anti-inflammatory effects of cis-aco

We next investigated whether *cis*-aco's ability to decrease inflammation (e.g., IL-6, Fig. 1C,D) was a secondary consequence of its antiviral action or due to intrinsic immunomodulatory properties.

IAV infection in lung epithelial cells activates intracellular signaling pathways that upregulate the expression of inflammatory cytokines (Dai et al, 2011; Gaur et al, 2011; Yu et al, 2020). Consistent with this, we observed increased phosphorylation of extracellular signal-regulated kinase (ERK)1/2, Protein kinase B (PKB; also known as AKT), and the p65 subunit of the nuclear factor-kappa B (NF-κB) at 20 h p.i. compared to non-infected cells (MOCK) (Fig. 6A). *Cis*-aco treatment appeared to inhibit the accumulation of these phosphorylated proteins (Fig. 6A,B). To further assess whether *cis*-aco possesses inherent anti-inflammatory properties, we stimulated lung epithelial cells with different inflammatory agonists: (i) Poly(I:C) (PIC), a TLR3 agonist that

mimics viral RNA, (ii) Phorbol 12-myristate 13-acetate (PMA), which activates protein kinase C signaling, and (iii) TNFα, a major inflammatory cytokine. As expected, all agonists induced IL-6 secretion (Fig. 6C). Interestingly, *cis*-aco treatment inhibited IL-6 release in a dose-dependent manner (2.3-fold decrease after TNFα stimulation, $P < 0.04$, and >fourfold decrease after PIC or PMA stimulation, $P < 0.01$; Fig. 6C). These findings highlight the intrinsic anti-inflammatory properties of *cis*-aco, independently of its antiviral activity.

### In vivo confirmation of the anti-inflammatory activity of cis-aco

To examine the anti-inflammatory effects of *cis*-aco in vivo, we used a murine model of acute lung injury induced by bacterial lipopolysaccharide (LPS; (Matute-Bello et al, 2008)). To this end, NF-κB-luciferase transgenic mice were treated or not with *cis*-aco 15 min post-LPS challenge, and NF-κB activity was measured as a proxy of inflammation (Fig. 6D,E). At 24 h post-stimulation, lung inflammation induced by LPS (Fig. 6D, center picture) was significantly reduced in *cis*-aco-treated animals (~75%, $P < 0.03$; Fig. 6D, right picture and Fig. 6E). Consistent with this observation, TNFα levels in the BAL fluids of LPS-challenged mice were also significantly lower in *cis*-aco-treated animals (70% reduction, $P < 0.008$, Fig. 6F). These in vivo experiments thus suggest that *cis*-aco can inhibit inflammatory signaling, resulting in decreased production of pro-inflammatory cytokines.

### The anti-influenza properties of *cis*-aco are independent of itaconate

*Cis*-aco is a TCA intermediate that can be converted into itaconate by the mitochondrial enzyme *cis*-aco decarboxylase (CAD), also known as ACOD1 or Irg1 (Fig. 7A) (Michelucci et al, 2013). To verify whether the anti-influenza effect of *cis*-aco is dependent on this conversion, we used a silencing approach targeting the CAD gene. This specific siRNA transfection effectively suppressed CAD expression in bronchial epithelial BEAS-2B cells compared to a control scramble siRNA (Fig. 7B). Interestingly, even under CAD silencing conditions, *cis*-aco retained its antiviral and anti-inflammatory properties, as determined through quantification of IAV infectious particles, neuraminidase activity, and IL-6 release (Fig. 7C–E). These findings suggest that *cis*-aco inhibits IAV infection in lung epithelial cells independently of its conversion into itaconate by CAD. These in vitro findings were further supported by in vivo experiments, where CAD-deficient mice

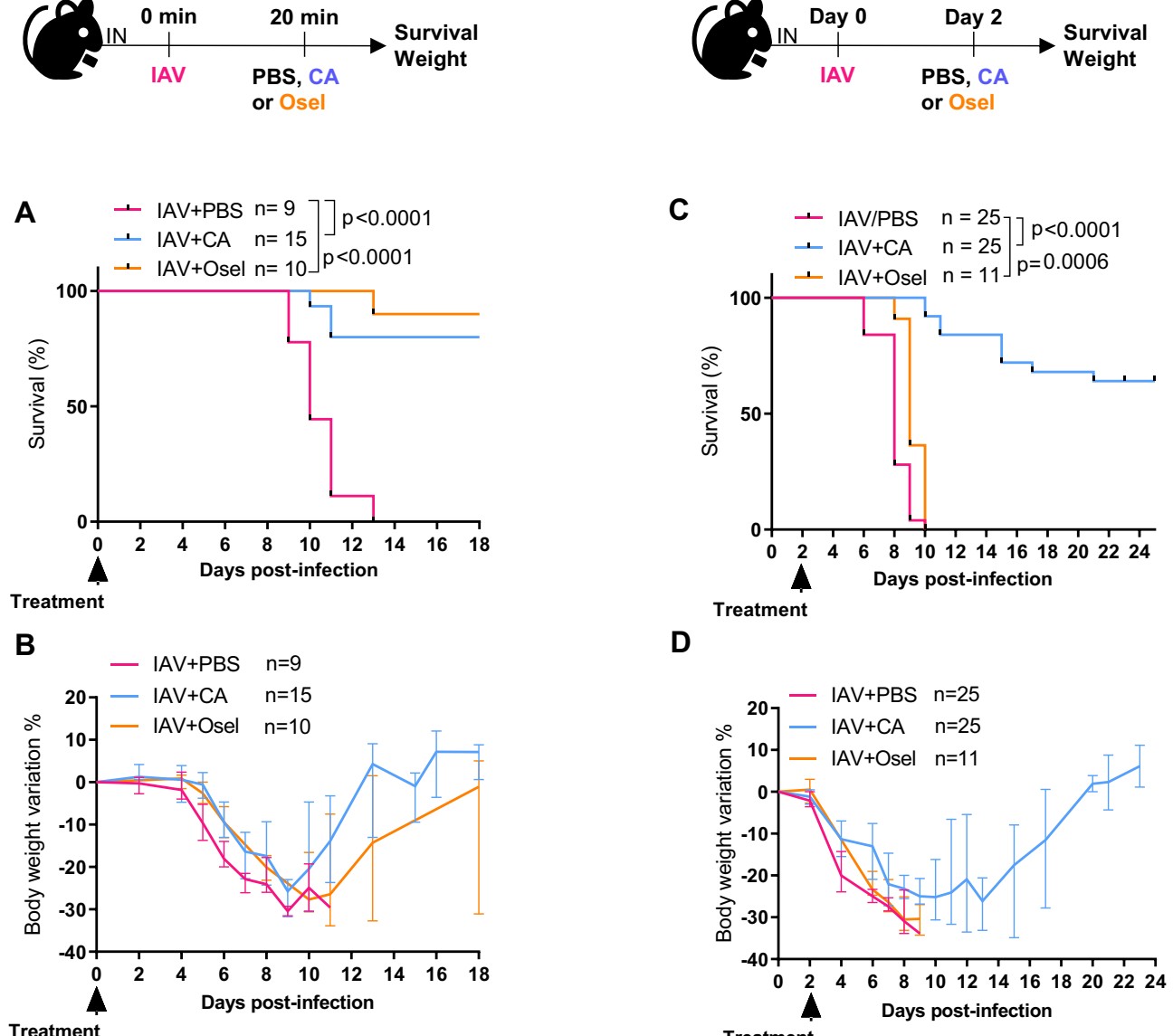

**Figure 3. *Cis*-aco provides superior protection against influenza infection in mice compared to Oseltamivir, even with delayed treatment.**

Seven-week-old female C57Bl/6 mice were intranasally infected with 200 pfu of influenza A/Scotland/20/74 (H3N2) virus (IAV) and treated intranasally with either 30 mg/kg of *cis*-aco (CA, blue) or 20 mg/kg of Oseltamivir (Osel, orange) at: (A, B) 20 min p.i. or (C, D) 2 days p.i. Animal survival rates (A, C) and body weight loss (B, D) were monitored daily. Data are presented as the mean ± SEM from at least two independent experiments. The number of mice in each treatment group ("*n*") is indicated. Statistical analysis was performed using the Log-rank (Mantel–Cox) test. Statistical significance: *$P < 0.05$, **$P < 0.01$, ***$P < 0.001$, ****$P < 0.0001$. Source data are available online for this figure.

infected intranasally with IAV and treated with *cis*-aco displayed survival rates similar to wild-type mice (Fig. 7F) (Demars et al, 2021).

## *Cis*-aco mitigates mortality in IAV-infected mice in a clinically relevant timeframe

The preceding data demonstrate that *cis*-aco prevents influenza-induced lung damage by reducing viral infection and inflammation. To assess its therapeutic potential in a clinically relevant context, we considered the typical delay between symptom onset and

treatment in human influenza cases (Seymour et al, 2017). This delay is critical for evaluating the efficacy of anti-influenza therapies in mice in "real-world" conditions. To complement our experimental study, we examined a prospective clinical trial (Lhommet et al, 2020) involving patients with community-acquired pneumonia (CAP) caused by influenza A and B viruses. Among the 153 CAP patients, 37% had viral pneumonia, 24% had bacterial pneumonia, and 20% had co-infections. IAV was the predominant pathogen, found in 33% of cases. Detailed characteristics of these CAP patients, particularly those with IAV infection, are provided in Table 2. The median [IQR] symptom-to-

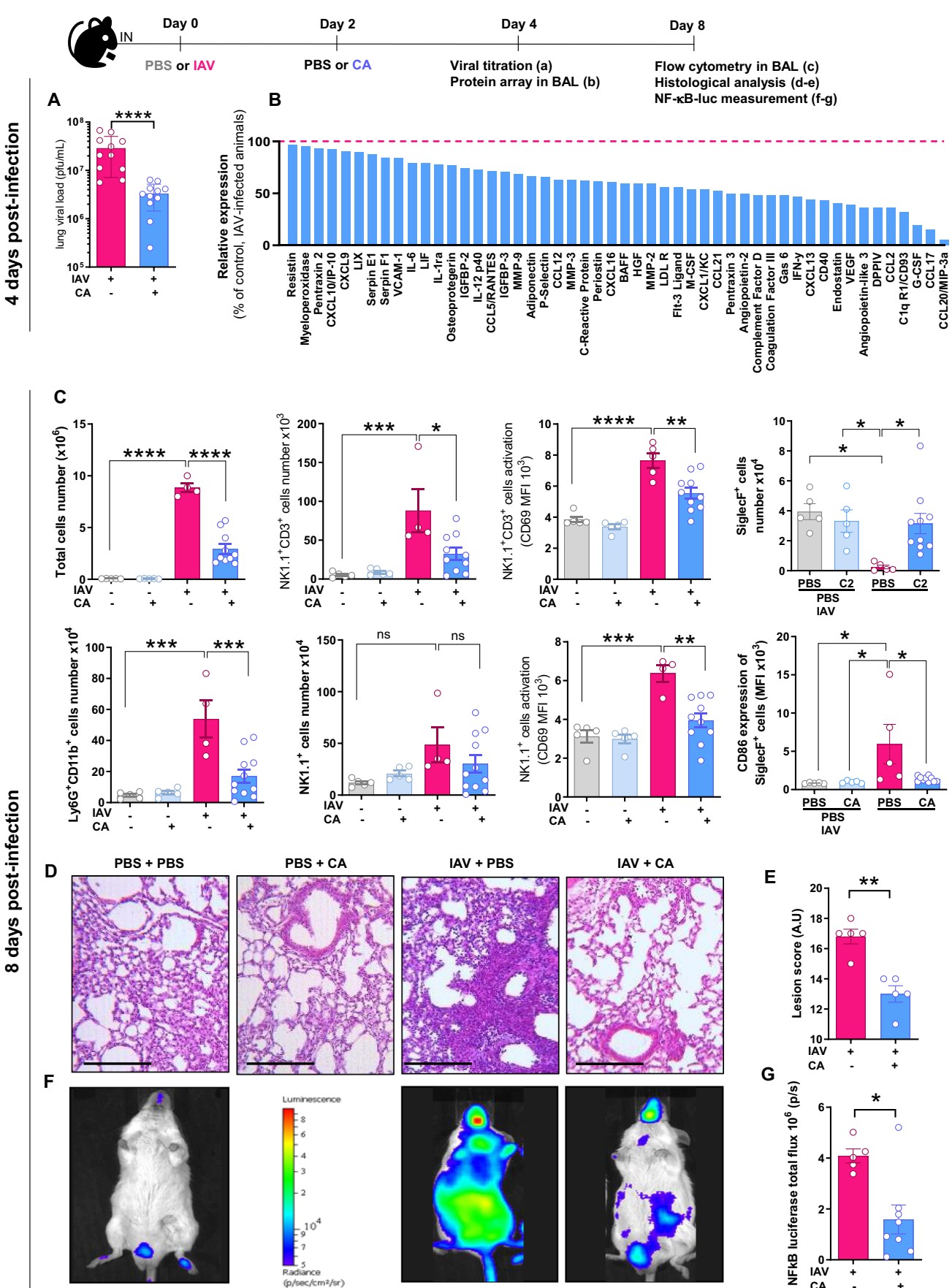

**Figure 4.   *Cis*-aco reduces viral load, lung inflammation, and tissue damage in IAV-infected mice.**

(A–E) Seven-week-old female C57Bl/6 mice were intranasally infected with 200 pfu of influenza A/Scotland/20/74 (H3N2) virus (IAV) and treated or not 2 days p.i. with 30 mg/kg of *cis*-aco (CA) intranasally. Mice were euthanized at either 4 or 8 days p.i. (A, B) At day 4 p.i., (A) the viral load in lung tissues was quantified using a PFU assay, and (B) the levels of 50 mediators were measured in BAL fluids. (C–E) At day 8 p.i., (C) the number and activation status of immune and inflammatory cells in BAL fluids were determined by flow cytometry. (D, E) Lung sections were stained with H&E, and tissue lesions were assessed and scored (scale bar: 200 μm). (F, G) NF-κB transgenic Balb/c mice ($n = 9$) were intranasally infected with 300 pfu of IAV and treated 2 days p.i. with 30 mg/kg CA. At 8 days p.i., mice were anesthetized, and luciferin was administered intranasally (0.75 mg/kg). Bioluminescence was quantified using the IVIS imaging system. Statistical analyses were performed using the one-way ANOVA test (C), or the Mann–Whitney test (A, E, G). Data are presented as the mean ± SEM and include results from 1 experiment (B, F) or 3 (A) or 2 (C, D, E, G) independent experiments. The number of data points shown in each bar plot corresponds to the number of independent experiments performed for that condition. Statistical significance: *$P < 0.05$, **$P < 0.01$, ***$P < 0.001$, ****$P < 0.0001$. Source data are available online for this figure.

hospitalization time was 3 [2–7] days for all CAP cases and 4 [3–6] days for those attributed to influenza (A or B) infection (Fig. 8A).

To simulate this clinically relevant delay, *cis*-aco treatment was initiated in mice on day 4 p.i., by which time IAV infection had already progressed to pneumonia, with substantial production of inflammatory mediators (Fig. 4B). To more closely mirror clinical conditions, a second dose of *cis*-aco was administered on day 5 p.i. In line with previous clinical observations (Rodríguez et al, 2011; Hernu et al, 2018) and reinforcing our data from Fig. 3, delayed oseltamivir treatment did not prevent IAV-induced mortality (Fig. 8B). In contrast, *cis*-aco treatment on days 4 and 5 improved survival, increasing the survival rate from 0% to approximately 50% (Fig. 8B). These findings highlight the potent curative effects of *cis*-aco against IAV infection within a clinically relevant timeframe.

## Discussion

Influenza pathophysiology is multifactorial, involving direct viral cytopathic effects and dysregulation of the host immune response, which together contribute to lung damage. Currently, the treatment of influenza remains an unmet medical need, as the effectiveness of antiviral therapies is limited, especially when administered late in the course of infection. Moreover, strategies that reduce inflammation without controlling viral replication have been associated with increased mortality (Ni et al, 2019). In this study, we show that *cis*-aco treatment addresses both key aspects of influenza pathophysiology. *Cis*-aco effectively inhibits major human influenza strains by impairing viral RNA and protein expression, while simultaneously reducing influenza-induced inflammatory signaling. Guided by a clinical perspective, we demonstrated the protective anti-influenza effects of *cis*-aco in human lung tissue explants and in infected mice treated within a clinically relevant timeframe, surpassing the efficacy window of the standard of care, oseltamivir/Tamiflu.

By screening a range of related metabolites using an in vitro model of IAV infection in human bronchial epithelial cells, the primary target cells for IAV (Lechner et al, 1982; Reddel et al, 1988; Benam et al, 2019), *cis*-aco emerged as the most promising molecule. It is synthesized in the TCA cycle from citric acid through the action of the mitochondrial enzyme aconitate hydratase (also named aconitase) and can be converted by the enzyme *cis*-aconitate decarboxylase (CAD; also called ACOD1 or Irg1) into itaconate. This latter metabolite is central in linking the innate immune response to cell metabolism, including during IAV infection (Michelucci et al, 2013; Sethy et al, 2019; Sohail et al, 2022). However, the anti-influenza effects of *cis*-aco were not due to its conversion into itaconate, as antiviral and anti-inflammatory

activities persisted despite efficient silencing of CAD using siRNA. These in vitro findings were further confirmed in vivo, with CAD-deficient mice infected intranasally with IAV and treated with *cis*-aco displaying survival rates comparable to wild-type mice. Collectively, these data demonstrate that the anti-influenza properties of *cis*-aco are independent of its conversion to itaconate.

To further elucidate the mechanism of action of *cis*-aco, we investigated its effects on various stages of the IAV life cycle. Our results show that *cis*-aco impairs viral polymerase activity, leading to decreased levels of viral RNA and proteins and preventing the formation of new viral particles. These antiviral effects were confirmed against both influenza A and B viruses, which is particularly noteworthy given that current neuraminidase inhibitors are less effective against influenza B viruses compared to influenza A (Jefferson et al, 2014; Burnham et al, 2013). However, further research is required to fully elucidate the mechanism of *cis*-aco, given its likely complex effects on both viral replication and inflammatory cell signaling.

To further explore the cellular mechanisms underlying *cis*-aco's antiviral activity, we first examined the global metabolic landscape in influenza virus-infected mice. Untargeted metabolomic profiling of murine lungs revealed that, despite broad metabolic reprogramming during infection, levels of both itaconate and *cis*-aconitate remained largely unchanged compared with mock-infected animals (Appendix Fig. S1a,b). We then analyzed specific TCA cycle–related enzymes. In mouse lungs, the influenza virus led to increased CAD expression and reduced IDH2 levels (Appendix Fig. S2). Similarly, BEAS2 cells showed a 27-fold increase in CAD and a 2.7-fold decrease in IDH2 (Appendix Fig. S3a). These observations were further confirmed in primary human models, including airway epithelial cells, alveolar macrophages, and lung tissue explants (Appendix Fig. S4). Despite only a few mitochondrial genes being significantly altered, CAD was consistently induced in all models upon influenza virus challenge.

*Cis*-aco treatment counteracted this metabolic imbalance. In both human bronchial epithelial cells and mouse lungs, *cis*-aco reduced CAD expression (Appendix Figs. S2 and S3) and upregulated key TCA cycle enzymes (CS, ACO2, IDH2, SDHA) regardless of infection status (Appendix Fig. S3a) while restoring IDH2 expression in infected lungs (Appendix Fig. S2). Although *cis*-aco did not significantly alter TCA cycle metabolite levels in human bronchial epithelial cells (Appendix Fig. S3b), Seahorse analysis revealed enhanced mitochondrial respiration and ATP production (Appendix Fig. S3c–g). Importantly, serum itaconate levels remained unchanged in both naïve and infected mice (not shown), confirming that *cis*-aco's antiviral effects are independent of itaconate accumulation.

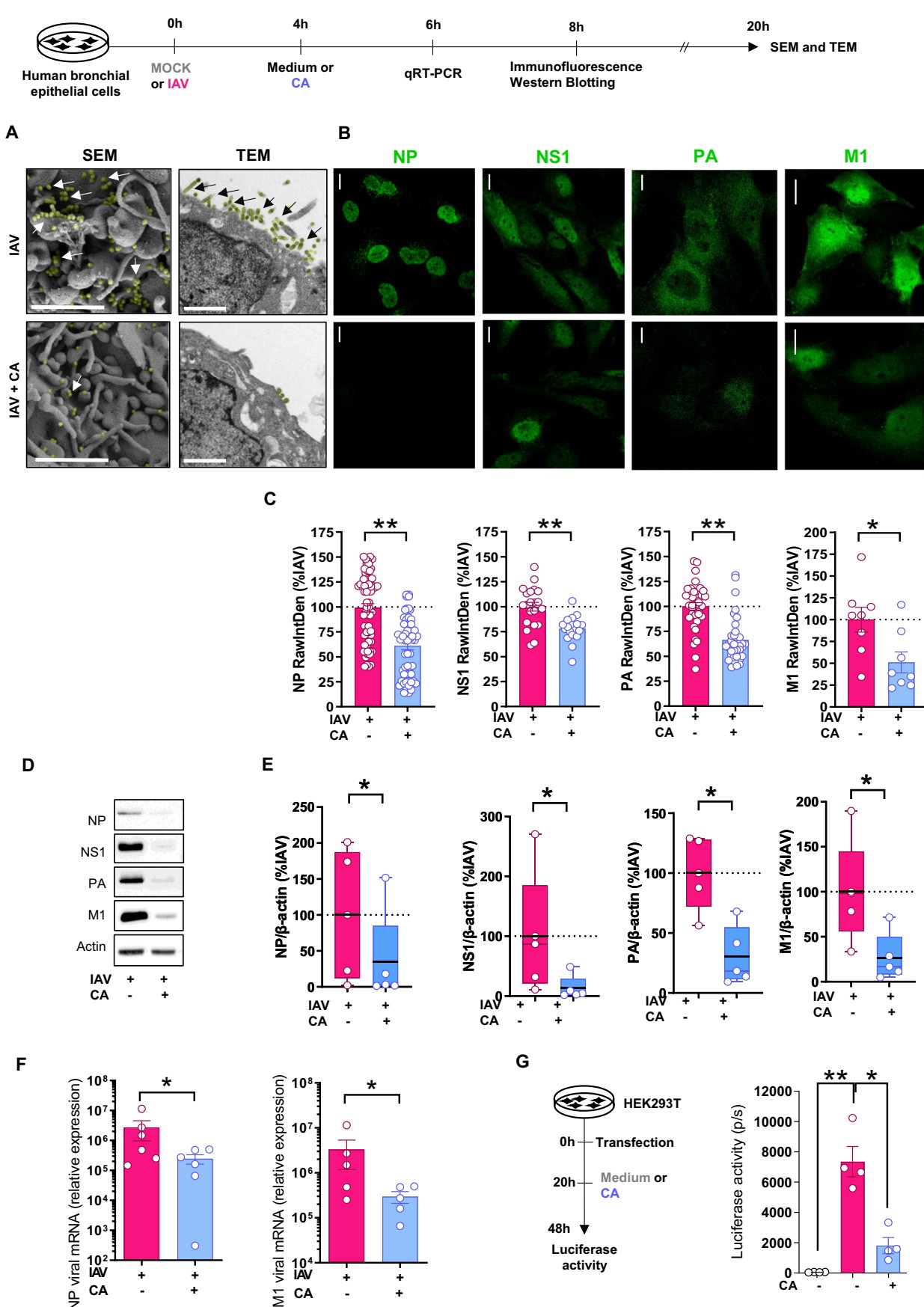

**Figure 5.  Anti-influenza virus properties of *cis*-aco involve the inhibition of viral polymerase activity.**

BEAS-2B cells were infected with influenza A/Scotland/20/74 (H3N2) virus at a MOI = 5 (**A**) or MOI = 1 (**B–F**) for 4 h, then washed and treated with 3.4 mM of *cis*-aco (**CA**) or left untreated (Medium). (**A**) Representative images from transmission electron microscopy (upper panel) and scanning electron microscopy (lower panel) show IAV particles budding at 20 h p.i., indicated by arrows (scale bar: 1 µm). (**B–E**) At 8 h p.i., viral protein (green) expression and trafficking were analyzed by (**B, C**) confocal microscopy (scale bar: 20 µm) and (**D, E**) Western blotting to detect viral NP, NS1, and PA proteins. (**C**) Raw integrated density (RawIntDen), calculated as the sum of all pixel values in the region of interest, was measured and normalized to the mean of the IAV condition for each experiment. (**E**) Relative protein levels were normalized to the mean value of "IAV condition" samples, with β-actin as a loading control. (**F**) At 6 h p.i., IAV transcription was quantified by RT-qPCR, measuring M1 viral mRNA levels. (**G**) A minigenome assay was performed in HEK-293T cells to test the effect of *cis*-aco on viral polymerase activity. Cells were transfected with plasmids encoding PA, PB1, PB2, NP, and the reporter plasmid pPolI-WSN-NA-firefly luciferase. At 20 h post-transfection, cells were treated with 0 or 3.4 mM *cis*-aco (**CA**) and luciferase activity was measured at 48 h post-transfection. Results are presented as the mean ± SEM from 3 (**A–C**), 4 (**D, E, G**), or 5 (**F**) independent experiments. The number of data points shown in each bar plot corresponds to the number of independent experiments performed for that condition. Statistical analyses were performed using the Kruskal–Wallis test with Dunn's multiple comparison test (**G**), the Mann–Whitney test (**C**), or the Wilcoxon matched-pairs rank test (**E, F**). Statistical significance: *$P < 0.05$, **$P < 0.01$, ***$P < 0.001$, ****$P < 0.0001$. Source data are available online for this figure.

While both *cis*-aco and itaconate share immunomodulatory and antiviral properties (Michelucci et al, 2013; Sethy et al, 2019; Sohail et al, 2022), they likely act through distinct mechanisms. Itaconate has been reported to inhibit SDH and IDH2, suppress mitochondrial respiration, reduce ROS, and enhance AKT phosphorylation. In contrast, our study shows that *cis*-aco upregulates SDHA and IDH2, enhances mitochondrial respiration and ATP production and does not reduce ROS (Fig. EV1E) nor activate AKT phosphorylation (Fig. 6A,B). Taken together, these findings indicate that *cis*-aco acts *via* a mechanism distinct from, and in part opposite to that of itaconate.

Severe viral pneumonia is closely associated with lung hyperinflammation, which can lead to acute respiratory distress syndrome (ARDS), as well as significant morbidity and mortality due to respiratory failure. While antiviral therapy and supportive measures, such as mechanical ventilation, are standard treatments, the use of immune-modulatory agents remains controversial (Ni et al, 2019). Proponents argue that anti-inflammatory treatments could reduce lung inflammation, whereas opponents caution that such treatments might interfere with immune responses, potentially delaying viral clearance and increasing mortality risk. This debate was particularly evident during the A/H1N1 2009–2010 influenza pandemic, when anti-inflammatory therapies like steroids were linked to higher mortality rates (Brun-Buisson et al, 2011; Kim et al, 2011; Matthay and Liu, 2011). As a result, anti-inflammatory therapies are not recommended for influenza-related pneumonia, and the disease was excluded from the largest randomized clinical trial assessing steroids in severe community-acquired pneumonia (i.e., the CAPE COD study) (Dequin et al, 2023).

Given this context, *cis*-aco's dual action as both an antiviral and anti-inflammatory agent holds considerable therapeutic potential. By simultaneously reducing hyperinflammation and enhancing viral clearance, *cis*-aco could offer a comprehensive treatment strategy. It exerts its anti-inflammatory effects by inhibiting key pro-inflammatory pathways, including ERK, AKT, and NF-κB, which are also crucial for viral replication and often hijacked by IAV (Schmolke et al, 2009; Luig et al, 2010; Schreiber et al, 2020; Yan et al, 2018; Botwina et al, 2020; Haasbach et al, 2017; Pleschka et al, 2001; Börgeling et al, 2014, 38). For instance, IAV activation of the ERK pathway facilitates the nuclear export of vRNPs (Yan et al, 2018; Botwina et al, 2020; Haasbach et al, 2017; Pleschka et al, 2001). By modulating these cellular signaling factors, *cis*-aco may impair IAV replication more potently than existing antiviral drugs that primarily target viral components (Li et al, 2019; Schräder et al,

2018; Müller et al, 2012). Furthermore, because *cis*-aco acts on cellular pathways rather than solely targeting viral factors, it has the potential to limit the risk of drug resistance by reducing selective pressure on the virus itself (Adamson et al, 2021).

To confirm the protective mechanisms of *cis*-aco in a more representative and challenging context, we conducted in vivo experiments using mice infected with a lethal dose of IAV (Guillon et al, 2022; Le Goffic et al, 2006). *Cis*-aco mitigated all key aspects of influenza pathology, including reducing viral replication, controlling excessive inflammatory cytokine production, decreasing immune cell recruitment and activation, and minimizing tissue lesions.

Beyond its efficacy against infectious diseases, our study also emphasizes the broader therapeutic potential of *cis*-aco. Its potent anti-inflammatory properties position it as a promising candidate for managing non-infectious pulmonary inflammatory conditions, such as asthma or fibrosis. Local administration of *cis*-aco locally *via* inhalation could enhance its therapeutic effects while reducing potential systemic side effects (Banat et al, 2023).

Preclinical studies are essential for advancing potential treatments, but traditional models—such as mouse and other animal models or in vitro cell cultures—often fail to accurately predict efficacy in patients. To improve clinical translation, we used a multimodal strategy that included: (i) human lung samples from individuals at high risk for influenza (e.g., elderly), (ii) a dosing schedule mimicking clinical treatment (instead of prophylactic or simultaneous treatment, as typically used), and (iii) comparison with an FDA-approved drug. We demonstrated *cis*-aco antiviral effects not only in human bronchial epithelial cell lines but also in advanced ex vivo lung culture models such as primary epithelial cells and human organotypic lung cultures. The latter preserves the complex lung tissue architecture and cellular diversity, providing a more relevant evaluation context than conventional cell cultures (Lam et al, 2023). Next, we compared *cis*-aco with oseltamivir, the most commonly recommended anti-influenza drug (Davidson, 2018). While oseltamivir is effective at early infection stages, its limited therapeutic window (Hong et al, 2020; Ding et al, 2017) and delayed administration in many patients restricts its efficacy. For example, only 20% of patients receive oseltamivir within 2 days of symptom onset, while viral pneumonia hospitalization typically occurs 4–5 days post-symptom onset (Hernu et al, 2018; Garot et al, 2022)). These findings highlight the importance of evaluating anti-influenza treatments at day 4 post-infection to better reflect clinical realities.

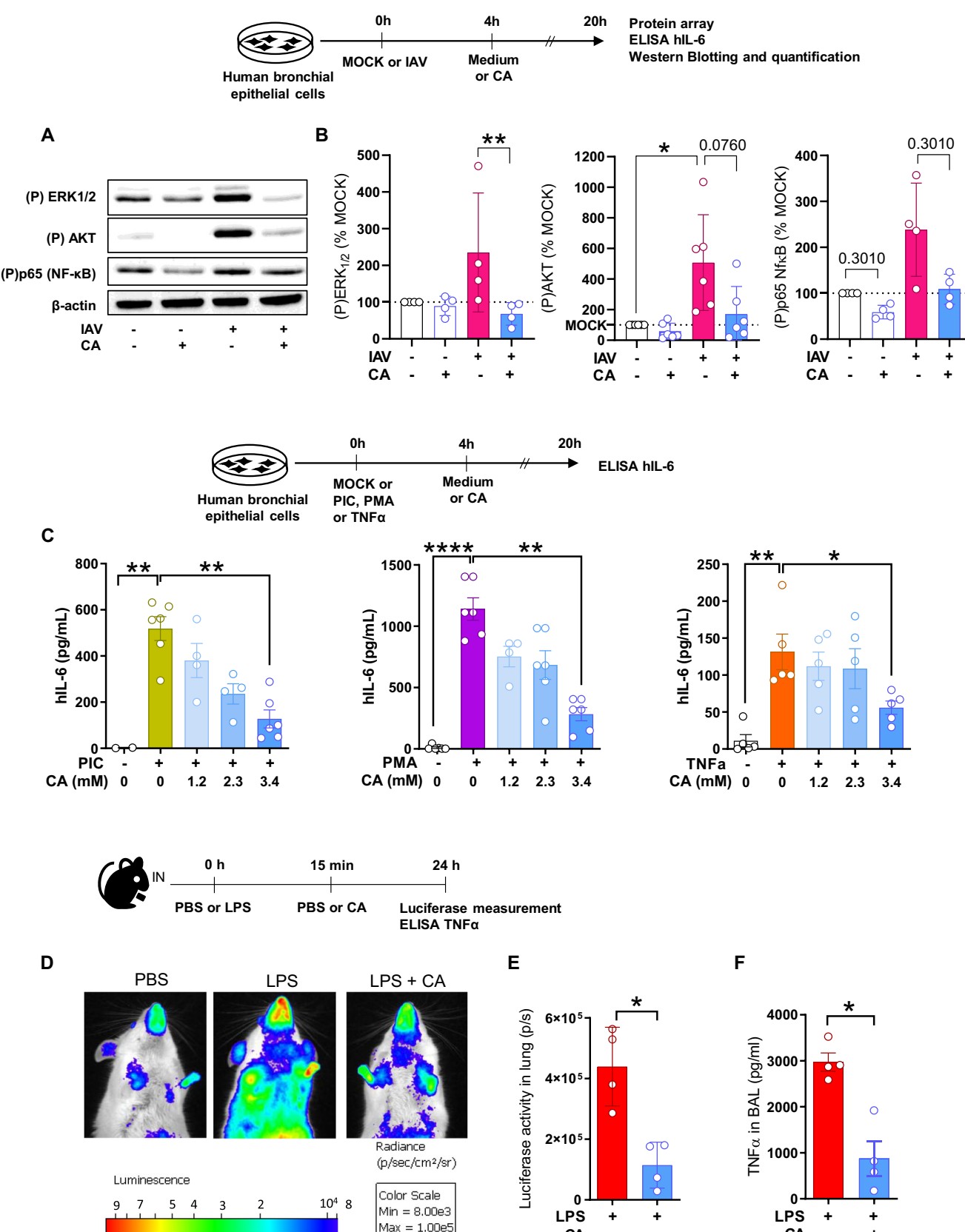

**Figure 6.** *Cis*-aco reduces pro-inflammatory responses and signaling.

BEAS-2B cells were infected or not with the influenza A/Scotland/20/74 (H3N2) virus at an MOI = 1 for 4 h, and subsequently treated or not with 3.4 mM of *cis*-aco (CA) for 16 h (**A, B**). (**A**) Representative western blotting showing phosphorylated forms (P) of ERK1/2, AKT, and p65 proteins with β-actin as a loading control. (**B**) Signal quantification of proteins and normalization were performed for each experiment. (**C**) BEAS-2B cells were stimulated or not (MOCK) with 2 μg/mL Poly(I:C) (PIC), 50 nM Phorbol 12-myristate 13-acetate (PMA), or 20 ng/mL Tumor necrosis factor alpha (TNFα) for 4 h, followed by treatment with increasing doses of *cis*-aco (CA) for 16 h. IL-6 levels in cell supernatants were measured by ELISA. (**D–F**) NF-κB transgenic Balb/c mice were instilled with 10 μg LPS and treated intranasally with either PBS or 30 mg/kg *cis*-aco (CA) 15 min post-stimulation. At 24 h post-instillation, bioluminescence was measured using the IVIS imaging system after intranasal administration of luciferin. Data are presented as the mean ± SEM. Results reflect cumulative data from a single experiment (**D–F**; n = 4 mice per condition) or at least four independent experiments (**A–C**). The number of data points shown in each bar plot corresponds to the number of independent experiments performed for that condition. Statistical analyses were performed using the Friedman test (**B** and TNFα data in **C**), the Kruskal–Wallis test (**C**), or the Mann–Whitney test (**E, F**). Statistical significance: *$P < 0.05$, **$P < 0.01$, ***$P < 0.001$, ****$P < 0.0001$. Source data are available online for this figure.

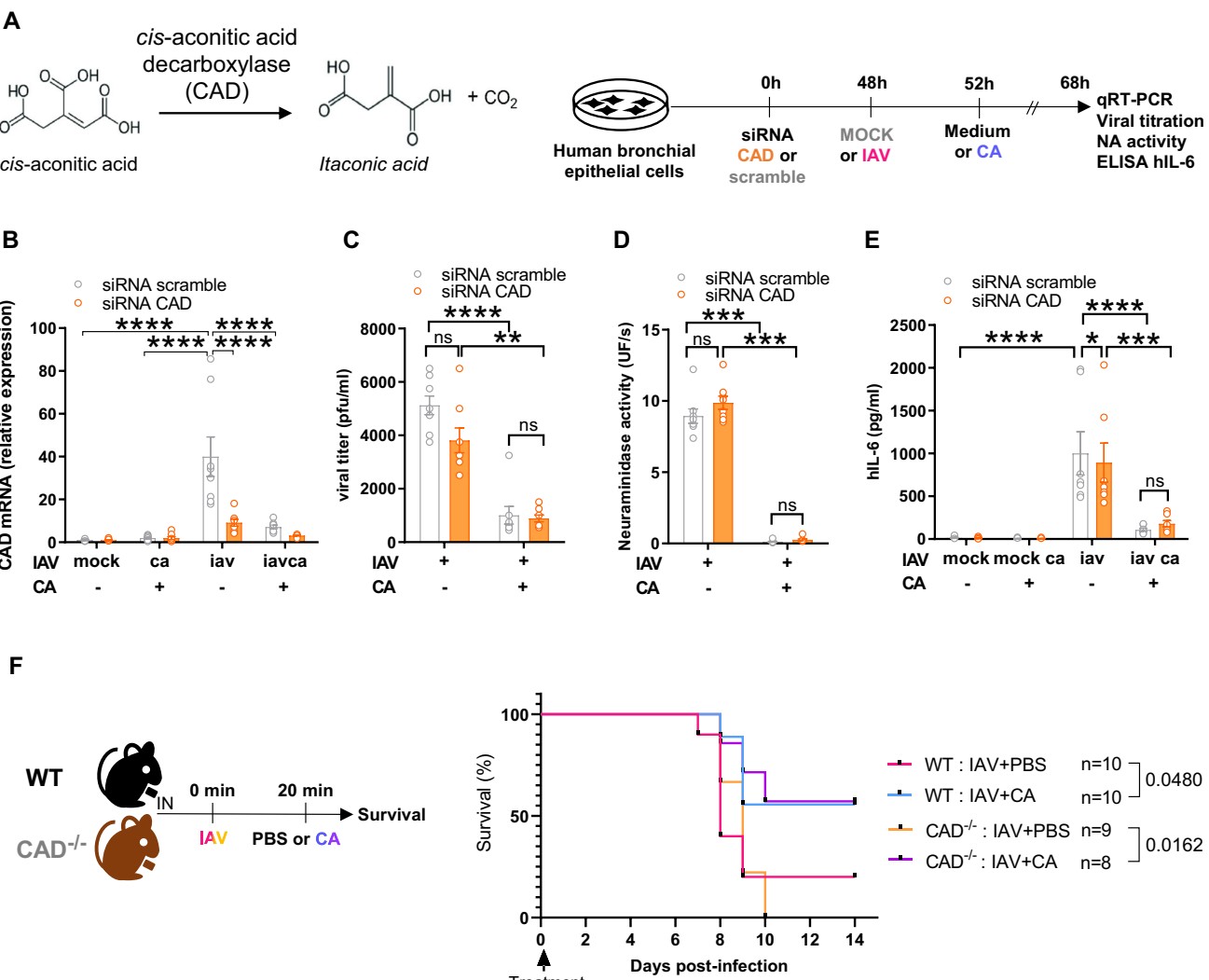

**Figure 7.** Anti-influenza activity of *cis*-aco is independent of itaconate.

(**A–E**) BEAS-2B cells were transfected with either *cis*-aconitate decarboxylase (CAD) or control (scramble) siRNA. At 48 h post-transfection, cells were infected or not with A/Scotland/20/74 (H3N2) virus (IAV) at an MOI = 1 for 4 h, and subsequently treated or not with 3.4 mM *cis*-aco (CA) for 16 h. (**A**) CAD catalyzes the decarboxylation of *cis*-aco to produce itaconate. (**B**) Gene knockdown was confirmed by RT-qPCR. (**C, D**) IAV particles production was measured by a plaque-forming units (pfu) assay (**C**) and a neuraminidase activity assay (**D**). (**E**) IL-6 levels in cell supernatants were quantified by ELISA. (**F**) CAD-deficient and wild-type (WT) mice were infected intranasally with 100 pfu of A/Scotland/20/74 (H3N2) virus (IAV) and treated intranasally or not with 30 mg/kg of *cis*-aco (CA) 20 min p.i. Animal survival was monitored daily. Data are presented as the mean ± SEM and are cumulative from a single experiment (**F**), or 3 (**B**), or 4 (**C–E**) independent experiments. The number of data points shown in each bar plot corresponds to the number of independent experiments performed for that condition. Statistical analyses were performed using the Mixed-effects model test (**B, E**) or the Kruskal–Wallis test with Dunn's multiple comparison test (**C–E**), or the Log-rank (Mantel–Cox) test (**F**). Statistical significance: *$P < 0.05$, **$P < 0.01$, ***$P < 0.001$, ****$P < 0.0001$. Source data are available online for this figure.

Interestingly, in our IAV-infected mouse model, oseltamivir's therapeutic effect declines markedly when administered at 2 or 4 days p.i., aligning with observations in critically ill influenza patients (Hernu et al, 2018). In contrast, *cis*-aco demonstrated a significant survival benefit, with ~70% survival when administered at day 2 p.i. and ~50% survival at day 4. These results are promising as similar efficacy is rarely achieved with FDA-approved influenza drugs under similar experimental conditions (Marathe et al, 2016). The superior efficacy of *cis*-aco across a larger treatment window underscores its potential as a major advancement in influenza therapy, allowing greater flexibility and effectiveness.

Our current findings significantly advance the identification and understanding of host-derived metabolites with antiviral activity, building on our previous studies on succinate (Guillon et al, 2022). Succinate mediates its anti-influenza effects by inducing the succinylation of a single amino acid (K87) on the IAV nucleoprotein, thereby impairing the trafficking of viral ribonucleoprotein complexes and disrupting the replication cycle. However, succinate has several limitations compared to *cis*-aco: it requires concentrations five times higher to achieve optimal anti-IAV inhibition, is ineffective against influenza B viruses, does not modulate infection-associated inflammatory signaling (Guillon

et al, 2022), and is not effective in vivo at later stages of infection (e.g., day 2 post-infection). In contrast, *cis*-aco offers the advantage of dual antiviral and anti-inflammatory actions, positioning it as a more potent endogenous molecule for controlling influenza infection.

Alongside *cis*-aco and succinate, other metabolites such as fumarate (Olagnier et al, 2020) and itaconate (O'Carroll and O'Neill, 2021) are increasingly recognized as endogenous mediators with remarkable antimicrobial activities. This discovery points to an ancient, conserved mechanism among primitive organisms, likely evolved to balance cellular proliferation and defense through a unique molecular system. This hypothesis aligns with theories on the origins of life, where metabolites were not only crucial for growth and development but also served as protective agents against environmental challenges (Scossa and Fernie, 2020). In a broader context, our findings further support the concept that metabokines are part of a wider family of host defense mediators— including cytokines, chemokines, bioactive lipids, and reactive oxygen species—that collectively orchestrate responses to external threats (Paludan et al, 2021; Chiang and Serhan, 2020).

In conclusion, our study demonstrates that *cis*-aco, a host-derived metabolite, exhibits potent anti-influenza properties with significant translational potential due to its natural origin and anticipated low toxicity. As illustrated in Fig. 9, *cis*-aco shows: (i) antiviral activity through inhibition of the IAV polymerase, (ii) strong anti-inflammatory and anti-cell death effects, (iii) broad-spectrum action against both influenza A and B viruses, and (iv) protective efficacy surpassing that of the reference drug oseltamivir. Altogether, our results pave the way for the development of *cis*-aco-based therapies for influenza virus infections. Further studies are warranted to investigate *cis*-aco pharmacokinetics and explore its potential in combination with existing antivirals.

**Table 2. Characteristics of patients hospitalized for community-acquired pneumonia (CAP).**

| | CAP (all pathogens) (n = 153) | CAP (Influenza) (n = 31) |
|---|---|---|
| Sex (male) [n (%)] | 108 (70.6) | 18 (58.1) |
| Age (years) | 62 [51–73] | 60 [48–66] |
| BMI (kg/m$^2$) | 27 [23–32] | 29.5 [27–35.5] |
| COPD [n (%)] | 108 (24.2) | 3 (9.7) |
| Asthma [n (%)] | 9 (5.9) | 3 (9.7) |
| Chronic heart failure [n (%)] | 22 (14.4) | 8 (25.8) |
| Chronic renal failure [n (%)] | 12 (7.8) | 4 (12.9) |
| Diabetes mellitus [n (%)] | 27 (17.6) | 5 (16.1) |
| Smoking status Current smoker [n (%)] | 57 (37.2) | 13 (41.9) |
| Alcohol dependence [n (%)] | 29 (18.9) | 4 (12.9) |
| Immunosuppression [n (%)]a | 41 (26.8) | 6 (19.4) |
| Cough [n (%)] | 133 (86.9) | 28 (90.3) |
| Expectoration [n (%)] | 59 (38.6) | 9 (29) |
| Dyspnea [n (%)] | 140 (91.5) | 29 (93.5) |
| Respiratory rate (IQR/min) | 32 [26–38] | 31 [25–37.5] |
| Thoracic pain [n (%)] | 35 (22.9) | 5 (16.1) |
| Fever [n (%)] | 115 (75.2) | 27 (87.1) |
| Chills [n (%)] | 60 (39.2) | 12 (38.7) |
| Arthralgia/myalgia [n (%)] | 35 (22.9) | 10 (32.3) |
| Hemoptysis [n (%)] | 12 (7.8) | 1 (3.2) |
| Diarrhea [n (%)] | 30 (19.6) | 2 (6.5) |

Quantitative data are reported as the median value and interquartile range [IQR] and qualitative value are reported as n (%).
aDefined as solid cancer, hemopathy, organ transplant, bone marrow transplant, HIV infection, and splenectomy.

## Methods

**Reagents and tools table**

| Reagent/resource | Reference or source | Identifier or catalog number |
|---|---|---|
| **Experimental models** | | |
| C57Bl/6 mice | Janvier Labs | C57BL/6JRj/Female SPF4 |
| CAD-deficient C57Bl/6 mice | Dr. Priscille Brodin (co-author of the study) | N/A |
| BALB/c NF-κB transgenic mice | Dr. Ronan Le Goffic (co-author of the study) | N/A |
| BEAS-2B cell line | ATCC® | CRL-9609 |
| A549 cell line | ATCC® | CCL-185 |
| MDCK.2 cell line | ATCC® | CRL-2935 |
| HEK293T cell line | ATCC® | CRL-11268 |
| Influenza virus B/Paris/234/2013 (Yamagata) lineage | European Virus Archive Global (EVAg). | 014V-01887 |

| Reagent/resource | Reference or source | Identifier or catalog number |
|---|---|---|
| Influenza virus B Victoria (B/Bretagne) | European Virus Archive Global (EVAg). | 014V-01882 |
| Influenza virus A/Scotland/20/74 | Gift from Pr. Sylvie van Der Werf, Pasteur Institute, France | N/A |
| Influenza virus A/PR/8/34 (PR8) | Kindly provided by Dr. Georg Kochs (Freiburg University, Germany) | N/A |
| Influenza virus A/H1N1 pdm09 | Dr. François Trottein (co-author of the study) | N/A |
| **Antibodies** | | |
| Anti-influenza A Virus Nucleoprotein antibody | Abcam | ab128193 |
| Anti-influenza A Virus Nucleoprotein antibody FITC | Abcam | ab20921 |
| Anti-Influenza A PA antibody | Invitrogen | PA532223 |
| Anti-Influenza NS1 antibody | Gift from Dr. Daniel Marc, INRAE, Nouzilly, France | N/A |
| β-actin Monoclonal Antibody | ThermoFisher Scientific | MA5-15739 |
| Anti-Phospho-Akt (Ser473) | Cell Signaling Technology | 4060 |
| Anti-Phospho-ERK1/2 | Cell Signaling Technology | 4377 |
| Phospho-NFκB p65 (Ser536) Monoclonal Antibody (T.849.2) | ThermoFisher Scientific | MA5-15160 |
| Anti-Mouse IgG (whole molecule)–Peroxidase antibody | Sigma-Aldrich | A9044 |
| Anti-Rabbit IgG (whole molecule)–Peroxidase antibody | Sigma-Aldrich | A9169 |
| APC-eFluor780-conjugated anti-CD45 (30-F11) | ThermoFisher Scientific | 47-0451-82 |
| CD86 (B7-2) Monoclonal Antibody (GL1), FITC | eBiosciences | 11-0862-82 |
| MHC Class II (I-A) Monoclonal Antibody (NIMR-4), PE | eBiosciences | 12-5322-81 |
| CD11b Monoclonal Antibody (M1/70), PerCP-Blueine5.5 | eBiosciences | 45-0112-82 |
| CD335 (NKp46) Monoclonal Antibody (29A1.4), eFluor 450 | eBiosciences | 48-3351-82 |

| Reagent/resource | Reference or source | Identifier or catalog number |
|---|---|---|
| APC Rat Anti-Mouse Ly-6G antibody | BD Biosciences | 560599 |
| PE/Blueine7 anti-mouse CD11c Antibody | BioLegend | 117318 |
| CD8a Monoclonal Antibody (53-6.7), eFluor 450 | eBiosciences | 48-0081-82 |
| PE/Blueine7 anti-mouse/human CD11b Antibody | BioLegend | 101216 |
| FITC anti-mouse CD4 Antibody | BioLegend | 130308 |
| BB700 Mouse Anti-Mouse NK-1.1 Antibody | eBiosciences | 566502 |
| CD69-APC, mouse Antibody | Miltenyi | 130-115-576 |
| PE/Blueine7 anti-mouse CD3ε Antibody | BioLegend | 100320 |
| CD3e Monoclonal Antibody (145-2C11), PE | eBiosciences | 12-0031-83 |
| F4/80 Monoclonal Antibody (BM8), eFluor 660 | eBiosciences | 50-4801-82 |
| Dihydrorhodamine 123 | Sigma-Aldrich | 109244-58-8 |
| MitoTracker™ Red CM-H2Xros | ThermoFisher Scientific | M7513 |
| V450 Mouse anti-Ki-67 | BD Biosciences | 561281 |
| Anti-p63 Antibody | Abcam | 124762 |
| Anti-tubulin Antibody | Sigma | T6793 |
| Anti-Mucin5Ac | ThermoFisher Scientific | MA5-12178 |
| Goat anti-rabbit antibody AF 546 | ThermoFisher Scientific | A11035 |
| Goat anti-mouse antibody AF 488 | ThermoFisher Scientific | A21121 |
| Goat anti-mouse antibody AF 647 | ThermoFisher Scientific | A21242 |
| **Oligonucleotides and other sequence-based reagents** | | |
| Viral M sequence: sense 5'-3' | Eurofins Genomics | AAG ACC AAT CCT GTC ACC TCT GA |
| Viral M sequence: antisense 5'-3' | Eurofins Genomics | CAA AGC GTC TAC GCT GCA GTC C |
| Viral NP sequence: sense 5'-3' | Eurofins Genomics | CTCTTGTTCGCACCGGAATG |
| Viral NP sequence: antisense 5'-3' | Eurofins Genomics | GGCTACGGCAGGTCCATAC |
| Human CAD sequence: sense 5'-3' | Eurofins Genomics | CGT GTT ATT CAG AGG AGC AAG AG |
| Human CAD sequence: antisense 5'-3' | Eurofins Genomics | AGC ATA TGT GGG CGG GAG |
| **Chemicals, enzymes, and other reagents** | | |
| Sodium Pyruvate (100 mM) (Gibco™) | ThermoFisher Scientific | 11360070 |

| Reagent/resource | Reference or source | Identifier or catalog number |
| --- | --- | --- |
| Lipopolysaccharide from Escherichia coli 0111:B4 | Invivogen | tlrl-eblps |
| DL-Isocitric acid trisodium salt hydrate | Sigma-Aldrich | I1252 |
| Triethyl citrate | Sigma-Aldrich | 14849 |
| Sodium fumarate dibasic | Sigma-Aldrich | F1506 |
| D-(+)-Glucose | Sigma-Aldrich | G7021 |
| Itaconic acid, +99%, ACROS Organics™ | Fisher Scientific | 10457700 |
| Oxaloacetic acid | Sigma-Aldrich | O4126 |
| cis-aconitic acid | Sigma-Aldrich | A3412 |
| trans-aconitic acid | Sigma-Aldrich | 122750 |
| Poly(I:C) LMW 25 mg | Invivogen | tlrl-picw |
| Phorbol 12-myristate 13-acetate | Sigma-Aldrich | 16561-29-8 |
| Recombinant Human TNF-alpha Protein | R&D systems | 210-TA |
| ActinRed™ 555 ReadyProbes™ Reagent | ThermoFisher Scientific | R37112 |
| NucBlue™ Fixed Cell ReadyProbes™ Reagent | ThermoFisher Scientific | R37606 |
| 2'-(4-Methylumbelliferyl)-α-D-N-acetylneuraminic acid sodium salt hydrate | Chemodex | M0096 |
| Protease Inhibitor Cocktail | Sigma-Aldrich | P8340 |
| PhosphoSafe Extraction Reagent | Sigma-Aldrich | 71296 |
| Gibco™ Ham's F-12 Nutrient Mix | Fisher Scientific | 31765027 |
| Gibco™ MEM | Fisher Scientific | 31095029 |
| Gibco™ GlutaMAX™ Supplement | Fisher Scientific | 13462629 |
| Gibco™ HEPES (1 M) | Fisher Scientific | 11560496 |
| BEGM™ Bronchial Epithelial Cell Growth Medium BulletKit™ | Lonza | CC-3170 |
| Trypsin 0.25%/EDTA 0.02% in PBS | PAN BIOTECH | P10-020100 |
| Trypsin, TPCK Treated | ThermoFisher Scientific | 20233 |
| Trypsin / Lys-C Mix, Mass Spec Grade | Promega | V5072 |
| MEM Eagle with Earle's BSS (2X) | Lonza | BE12-668F |
| Crystal Violet Oxalate | RAL Diagnostics | 361490 |
| Formaldehyde, 37 wt % sol. in water, stab. with 5-15% methanol | Acros Organics | 119690010 |

| Reagent/resource | Reference or source | Identifier or catalog number |
| --- | --- | --- |
| Avicel® RC 581 Stabilizer | FMC BioPolymer | N/A |
| Annexin V-FITC kit | Miltenyi Biotech | 130-092-052 |
| SYTOX™ Green nucleic acid stain | Invitrogen | S7020 |
| Propidium iodide | Sigma-Aldrich | P4170-25MG |
| True-Nuclear™ Transcription Factor Buffer Set | Biolegend | 424401 |
| BD Cytofix/Cytoperm™ Fixation/Permeabilization Solution Kit | BD Biosciences | BDB554714 |
| Red Blood Cell Lysing Buffer Hybri-Max™ | Sigma-Aldrich | R7757 |
| TB Green® Premix Ex Taq™ | Takara | RR420L |
| 50% EM Glutaraldehyde | TAAB Laboratory Equipment | G045 |
| Uranyl acetate | Merck | 8473 |
| Osmium tetroxide 4% solution | Electron Microscopy Science | 19150 |
| Oseltamivir phosphate | Sigma-Aldrich | 204255-11-8 |
| Gibco™ optiMEM | Fischer Scientific | 31985070 |
| Invitrogen™ Lipofectamine™ RNAiMAX Transfection Reagent | Fischer scientific | 13-778-150 |
| ON-TARGETPlus human smartpool IRG1 | Dharmacon | L-180668-01-0005 |
| MISSION® negative control scramble | Sigma-Aldrich | SIC001 |
| Gibco™ Milieu Hibernate™-A | Fischer Scientific | 12087586 |
| Primocin | Invivogen | ant-pm-05 |
| Bovine Albumin Fraction V | Fischer Scientific | 15260037 |
| PureCol | Advanced BioMatrix | 5005-B |
| Human Fibronectin Stabilized Solution | PromoCell | C-43060 |
| Heparin | StemCell | 7980 |
| Hydrocortisone | StemCell | 7925 |
| Soybean Trypsin Inhibitor | Sigma-Aldrich | T-9128 |
| Isoproterenol | Sigma-Aldrich | I-6504 |
| Bovine pituitary extract | Fischer Scientific | 11568866 |
| Epidermal Growth Factor (EGF) | Fischer Scientific | 10134762 |
| Serum-free keratinocyte medium (Gibco) | Fischer Scientific | 11590526 |

| Reagent/resource | Reference or source | Identifier or catalog number |
|---|---|---|
| Ca²⁺/Mg²⁺-free Hank's Balanced Salt Solution | Gibco | 88284 |
| Proteinase type XIV | Sigma-Aldrich | P5147-100MG |
| Hexamethyldisilazane | Sigma-Aldrich | 440191-100 ML |
| Fugene HD transfection reagent | Promega | E2311 |
| Firefly Luciferase Assay System | Promega | E1500 |
| Fluorescent dye SYTOX™ | Fischer Scientific | 10768273 |
| PneumaCult™Ex Kit | StemCell | 5001 |
| PneumaCult™Ali Kit | StemCell | 5008 |
| Hoechst 33342 | Invitrogen | H3570 |
| **Software** | | |
| GraphPad Prism | GraphPad Software | https://www.graphpad.com/scientific-software/prism/ |
| VenturiOne | Applied Cytometry | https://www.appliedcytometry.com/venturi/ |
| FUJI FILM Multigauge | Bioz | https://www.bioz.com/ |
| LightCycler 480 SW V.1.5 | Roche | https://lifescience.roche.com/ |
| ImageJ | Imagej | https://imagej.net/Welcome |
| BioStation IM software (v2.12) | Nikon | https://www.nikon.com/products/microscope-solutions/ |
| Leica LasX Life Sciences software | Leica Microsystems | https://www.leica-microsystems.com |
| Digital Micrograph V.3 software | Gatan | https://www.gatan.com/products/tem-analysis |
| MagMAX Express 96-Deep Well Magnetic Particle Processor software | Applied Biosystems | 4472991 |
| GentleMACS dissociator software | Miltenyi Biotec | 130-093-235 |
| ProCyte Dx hematocytometer software | Idexx | https://www.idexx.fr |
| IncuCyte® two-color incubator imaging system software | Essen Biosciences, Sartorius | https://www.essenbioscience.com |
| MF ChemiBis 3.2 software | DNR BioImaging Systems | https://hvdlifesciences.at/dnr-bio-imaging-systems.html |
| Zeiss Ultra plus FEG-SEM scanning electron microscope software | Zeiss | https://www.zeiss.com |
| **Other** | | |
| LIVE/DEAD™ Fixable Aqua Dead Cell Stain Kit | ThermoFisher Scientific | L34966 |
| Phusion™ High-fidelity DNA polymerase | ThermoFisher Scientific | 16237911 |
| Pierce™ BCA Protein Assay Kit | ThermoFisher Scientific | 23225 |

| Reagent/resource | Reference or source | Identifier or catalog number |
|---|---|---|
| CellTiter 96® AQueous One Solution Cell Proliferation Assay | Promega | G3582 |
| Nucleospin® RNA | Macherey-Nagel | 740955 |
| High Capacity cDNA reverse transcription kit | Applied Biosystems | 4368813 |
| Human IL6 ELISA DuoSet | R&D Systems | DY206 |
| Mouse MPO ELISA DuoSet | R&D Systems | DY3667 |
| Human Cytokine Array Kit | R&D Systems | ARY005B |
| Mouse XL Cytokine Array | R&D Systems | ARY028 |
| SequalPrep normalization kit | Thermo Fisher Scientific | A1051001 |
| MagMAX™ DNA Multi-Sample Kit | Thermo Fisher Scientific | 4413020 |

## Methods and protocols

### Viruses

The influenza strains used in this study were initially provided by partner laboratories and subsequently amplified in M. Si-Tahar's laboratory. Specifically, the mouse-adapted influenza A/Scotland/20/74 (H3N2) strain was kindly provided by Prof. Sylvie van der Werf's team at the Pasteur Institute, Paris, France; the influenza A/PR/8/34 (H1N1) strain by Dr. Georg Kochs at Freiburg University, Germany; the pandemic H1N1 strain by Dr. François Trottein at the Center for Infection and Immunity of Lille; and the influenza B/Paris/234/2013 (Yamagata lineage) and B Victoria (B/Bretagne) strains were obtained through the European Virus Archive Global (EVAg).

### Cell line culture

In vitro experiments were performed using human bronchial epithelial BEAS-2B cells, except for plaque assays which used Madin-Darby Canine Kidney (MDCK) cells, and the minigenome assay which used HEK-293T. These cells were cultured in either F-12K Medium (BEAS-2B) or MEM (HEK-293T and MDCK) supplemented with 10% FBS, 100 U/ml penicillin, and 100 μg/ml streptomycin. All cells were mycoplasma-free. BEAS-2B cells were infected in medium without FBS for 4 h with IAV at MOI = 1 (except for TEM and SEM analysis, for which an MOI = 5 was applied). Cells were also stimulated in medium with FBS with 2 μg/ml Poly(I:C) or in medium without FBS with 2 μg/ml PMA or with 20 ng/ml TNFα. Four hours after the challenge, cells were washed with PBS and incubated for 4 h or 16 h with different concentrations of metabolites (cis-aconitate derivative cis-aconitic acid, trans-aconitic acid, itaconic acid, glucose, pyruvate, oxaloacetic acid, fumarate, isocitric acid) diluted in medium without FBS.

### Studies involving human participants

Informed consent was obtained from all participants in accordance with the World Medical Association Declaration of Helsinki, and

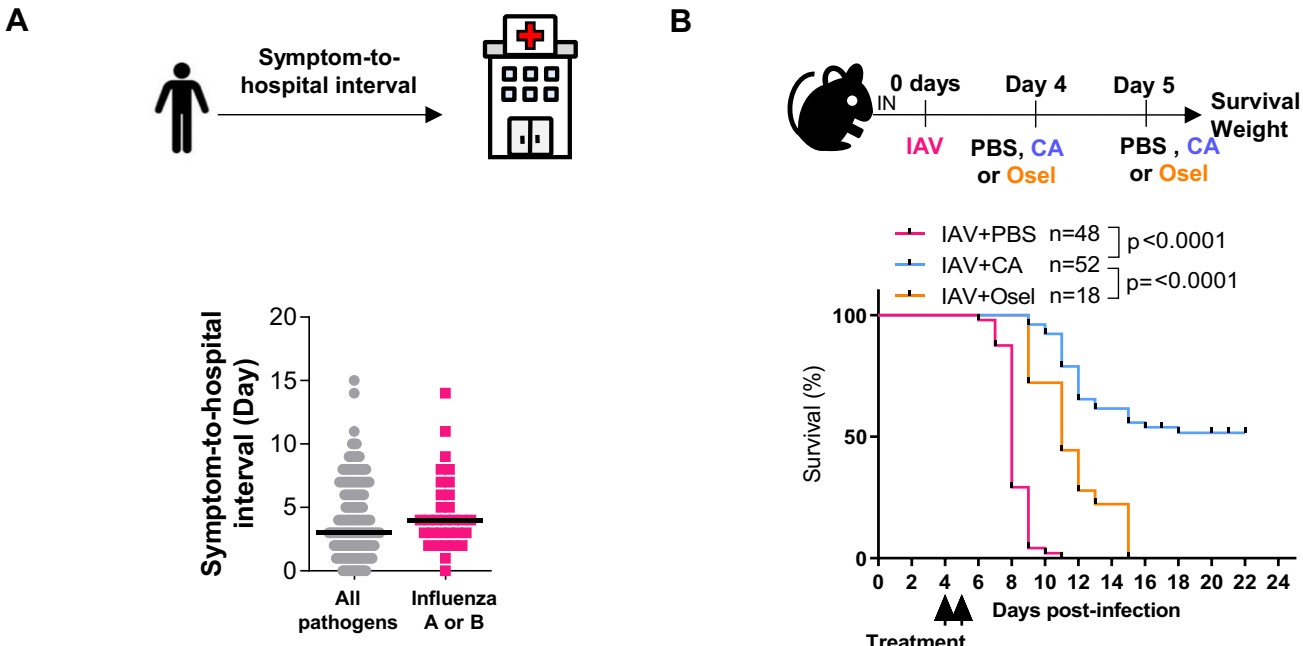

**Figure 8. Cis-aco protects mice from influenza infection within a clinically relevant timeframe.**

(A) Time between the onset of symptoms and the first hospital admission in patients hospitalized for community-acquired pneumonia (CAP). Each dot represents an individual patient with the median indicated by the black line. (B) Seven-week-old female mice were intranasally infected with 200 pfu of influenza A/Scotland/20/74 (H3N2) virus (IAV) and treated intranasally 4 and 5 days p.i. with 30 mg/kg cis-aco (CA, blue) or 20 mg/kg Oseltamivir (Osel, orange), or left untreated (pink). Survival was monitored daily. Data in (B) are presented as the mean ± SEM and are cumulative from three independent experiments; the number of mice ("n") is indicated. Statistical analysis was performed using the Log-rank (Mantel–Cox) test. Statistical significance: *$P < 0.05$, **$P < 0.01$, ***$P < 0.001$, ****$P < 0.0001$. Source data are available online for this figure.

the study also conformed to the principles set out in the U.S. Department of Health and Human Services Belmont Report.

Tissue and cell collections were declared to the French Ministry of Graduate Study, Research, and Innovation (DC-2008-308, MESRI). Lung lobes were collected immediately following surgical resection at CHRU of Tours. All experiments adhered to The Code of Ethics of the World Medical Association and were approved by the Ethics Committee of the CHRU of Tours. Lung donors for this study varied in age, gender, medical history, and the cause of resection (see Table 1). Prospective data collection presented in Table 2 was conducted in a single center over an 18-month period. The study complied with French law for observational studies and with the STROBE guidelines for observational studies. The study was approved by the ethics committee of the French Intensive Care Society (CE SRLF 13–28), was approved by the "Commission Nationale de l'Informatique et des Libertés" (CNIL) for the treatment of personal health data. We gave written and oral information to patients or their next-of-kin. Patients or next-of-kin gave verbal informed consent, as approved by the ethics committee. Eligible patients were adults hospitalized in ICU for CAP. Pneumonia was defined as the presence of an infiltrate on a chest radiograph and one or more of the following symptoms: fever (temperature ≥ 38.0 °C) or hypothermia (temperature < 35.0 °C), cough with or without sputum production, or dyspnea or altered breath sounds on auscultation. Community-acquired infection was defined as an infection occurring within 48 h of admission. Cases of pneumonia due to inhalation or infection with Pneumocystis,

pregnant women, and patients under guardianship were not included. Cases with $PaO2 \geq 60$ mmHg in ambient air or with the need for oxygen therapy ≤4 L/min or without mechanical ventilation (invasive or non-invasive) were not included.

### Human primary bronchial epithelial cells (PBEC) culture

PBEC were isolated from normal bronchial tissues of lung cancer patients undergoing lobectomy at the university hospital (CHRU) of Tours. The cancer-free tissues were washed and incubated for 2 h at 37 °C with 0.018% (w/v) proteinase XIV in Ca2 + /Mg2 + -free Hank's Balanced Salt Solution. Epithelial cells were scraped from the luminal surface, washed, and cultured in serum-free keratinocyte medium supplemented with 2.4 ng/ml epidermal growth factor, 25 µg/ml bovine pituitary extract, 1 µM isoproterenol, 100 U/ml penicillin, and 100 µg/ml streptomycin on 6-well plates coated with 30 µg/ml PureCol, 10 µg/ml bovine serum albumin, and 5 µg/ml fibronectin. During the first week, 1/500 Primocin was added to the medium. After reaching near-confluence, cells were trypsinized and stored in liquid nitrogen.

For mucociliary differentiation, PBEC were cultured in PneumaCult EX medium, with a 3–4 day proliferation step before stimulation with 2 µg/ml Poly(I:C) or 50 nM PMA. The stimulation medium was a 1:1 mixture of BEGM and complete DMEM/F12, supplemented with 100 U/ml penicillin, 100 µg/ml streptomycin, 12.5 ml 1 M HEPES, and 5 ml GlutaMAX™.

For ALI differentiation, PBEC were cultured submerged on semipermeable transwell inserts (0.4 µm pore size) coated with

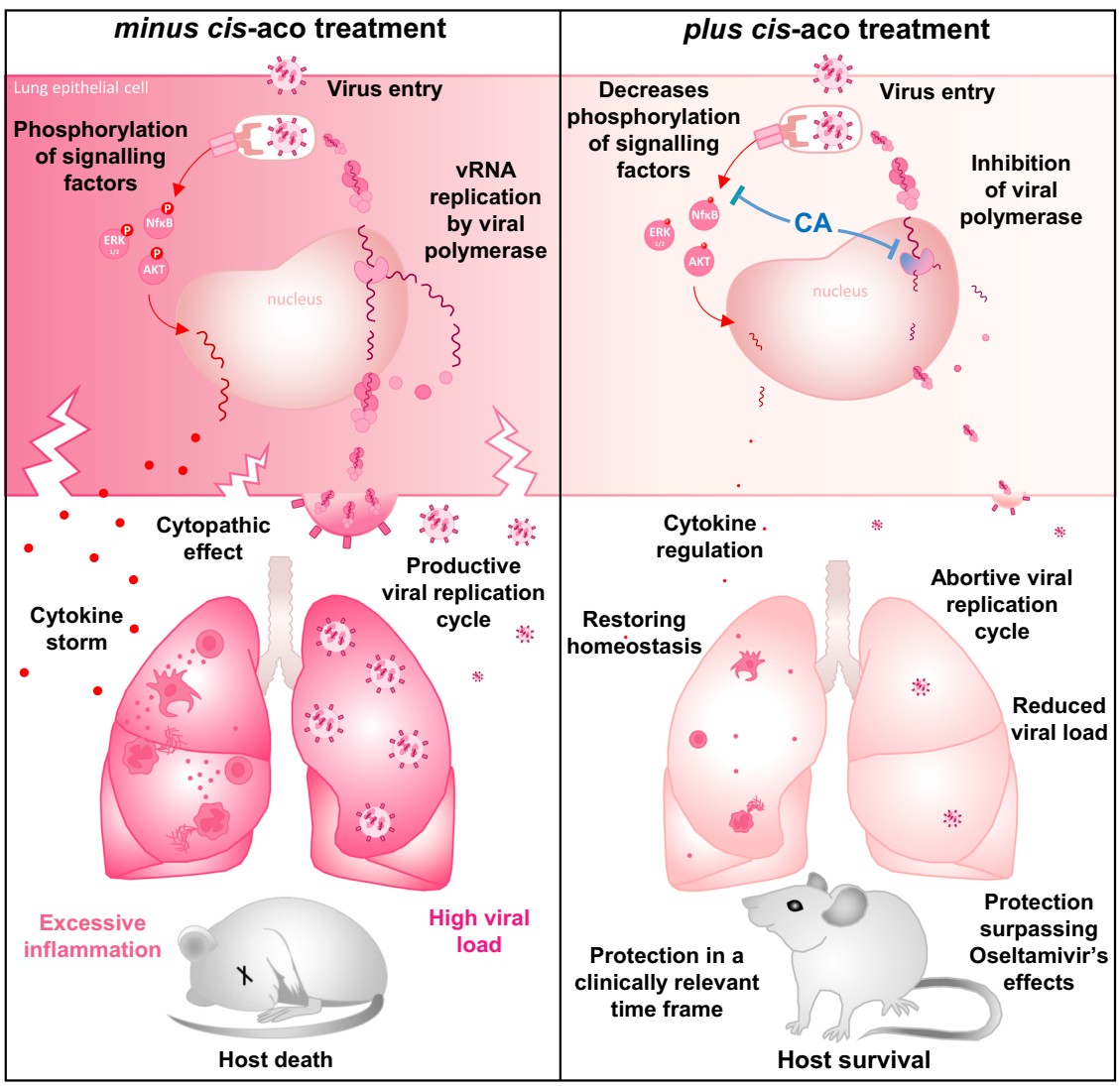

**Figure 9. Anti-influenza mechanism of action of *cis*-aco.**

*Cis*-aco (CA) impairs IAV polymerase activity, reducing viral mRNA expression and protein synthesis, thereby preventing effective virus replication. In addition, *cis*-aco downregulates inflammatory pathways triggered by various stimuli. In vivo, these combined antiviral and anti-inflammatory effects decrease viral load and mitigate excessive inflammation, providing superior protection against mortality compared to the reference anti-influenza drug Oseltamivir. Source data are available online for this figure.

collagen and fibronectin. Once confluent, the apical medium was removed, and cells were cultured at ALI for 3 weeks.

### Human organotypic lung culture (OLC)

The preparation of OLC was carried out according to a protocol derived from (Ferren et al, 2021), using human lung resections from surgical patients at the CHRU of Tours, collected in Hibernate medium containing 1/500 Primocin. Lung explants were sliced using the McIlwain® tissue chopper (Campden Instruments) at 500 μm thickness and placed back in Hibernate medium with 1/500 Primocin for slice dissociation under a dissection microscope before their immediate transfer to ALI. Individualized OLC were placed on semipermeable Millicell® cell culture inserts with PTFE membranes (0.4 μm pore size; Merck) already pre-activated with 1 ml of OLC culture medium. The culture medium is a volume-to-

volume mix of BEGM and DMEM medium with 100 U/ml penicillin and 100 μg/ml streptomycin. OLC were infected by drop deposition with $2.10^4$ pfu of A/Scotland/20/74 (H3N2) virus (IAV). After 2 h, OLC were treated with 3.4 mM of *cis*-aco. At 48 h p.i., the OLC subnatants were collected to measure viral titers.

### Animal care, handling, and study approvals

C57Bl/6 female mice (~8 weeks old) were purchased from Centre d'Elevage R. Janvier (Le Genest Saint-Isle, France) and housed under specific-pathogen-free conditions at Tours University animal facility (France), with ad libitum access to food and water. NF-κB luciferase transgenic BALB/C mice were generated by backcrossing NF-κB luciferase transgenic B10.A mice (a kind gift from Prof. Richard Flavell, Howard Hughes Medical Institute) with BALB/C mice to produce transgenic mice with white fur, minimizing light

absorption. C57BL/6NJ wild-type mice and C57BL/6NJ-Acod1e-m1(IMPC)J/J (CAD-/-) mice, deficient in CAD expression, were purchased from The Jackson Laboratory (Bar Harbor, ME, USA). All mice were maintained and bred at the Pasteur Institute of Lille, France (agreement B59-350009).

All procedures involving C57Bl/6 mice were conducted in compliance with European animal welfare regulations. Experiments adhered to the ethical standards set by the French government and were approved by our local and national ethics committees (CEEA.19, APAFIS#201604071220401.V2-4885, 2016111512369894 V3 - 7590). Studies using NF-κB transgenic BALB/c mice were approved by the Animal Care and Use Committee at the "Centre de Recherche de Jouy-en-Josas" (COMETHEA) under the relevant institutional authorization (Ministère de l'éducation nationale, de l'enseignement supérieur et de la recherche), authorization number: 2015100910396112v1 (APAFIS#1487). Experiments involving C57BL/6NJ-Acod1em1(IMPC)J/J (CAD-/-) mice deficient in CAD expression were ethically approved by the French Committee on Animal Experimentation and the Ministry of Education and Research (APAFIS#10232-2017061411305485 v6, approved on 14/09/2018).

### Neuraminidase (NA) assay

The assay measures the release of a 4-methylumbelliferone fluorescent product from the 2′-(4-Methylumbelliferyl)-α-D-N-acetylneuraminic acid sodium salt hydrate (MU-NANA) substrate. 67 μL of cell supernatant was incubated with 33 μL of MU-NANA (50 μM) in black 96-well black microplates. Fluorescence was immediately measured in a kinetic assay over 1 h at $Ex = 355$ nm and $Em = 460$ nm.

### Protein-array and ELISA

Protein array and DuoSet ELISA (Human IL-6, and mouse MPO and ALT) were performed according to the manufacturer's instructions (R&D Systems or Clinisciences for ALT ELISA).

### siRNA transfection

In total, $1.25 \times 10^5$ BEAS-2B cells were seeded in a 12-well plate the day before transfection with specific siRNA or negative control scramble siRNA. Each siRNA stock was diluted to 50 nM in 100 μL of optiMEM (Gibco) containing RNAiMax reagent (ratio siRNA:RNAiMax of 1:3). After 5 min of incubation at room temperature, 100 μL of each siRNA-mix was added to 900 μL of fresh medium per well. Gene knockdown efficacy was evaluated by RT-qPCR after 48 h (medium replaced after 24 h).

### RNA isolation and RT-qPCR

Cells in 6-well plates were lysed with 350 μL RA1 buffer (Macherey-Nagel) and 1/100 diluted β-mercaptoethanol. Total RNA was extracted using the NucleoSpin® RNA kit (Macherey-Nagel), including DNase digestion. RNA concentration was measured with a Nanodrop 2000. cDNA synthesis was performed from 500 ng RNA using the High-Capacity cDNA Reverse Transcription Kit, with IAV M1-specific sense primer or random primers. mRNA levels were quantified by RT-qPCR on a LightCycler 480 (Roche) using 10 ng cDNA, 10 μM primers, and 10 μL SYBR® Premix Ex Taq in a 20 μL reaction volume. Reactions were performed in duplicate, and the thermal protocol included initial denaturation at 95 °C for 30 s, followed by 40 cycles of denaturation (95 °C for 5 s)

and annealing/extension (60 °C for 20 s). Melting curves were generated to verify reaction specificity.

### IAV titration by plaque-forming units assay

Titrations in culture media and mouse lungs were performed as previously described (Blanc et al, 2016).

### IAV titration by $TCID_{50}$

Briefly, 96-well plates were seeded with MDCK cells 2 days prior to inoculation. Cells were washed with sterile PBS, and 200 μL of supernatant samples were added. Samples were serially diluted in MEM medium up to a $10^{-11}$ dilution. The assay was performed in eight replicates, with the last column of each plate serving as cell control without virus. Plates were incubated at 34–37 °C in a humidified 5% $CO_2$ atmosphere for 2 days. Following incubation, the inoculum was removed, cells were washed with PBS, and fixed for 25 min with 0.3% crystal violet in 20% methanol. Cells were then washed twice with PBS. $TCID_{50}$ per milliliter was calculated using a $TCID_{50}$ calculator (by Marco Binder; adapted at TWC).

### Transmission electron microscopy

Cells were washed with PBS, detached using trypsin, and centrifuged. They were fixed for 24 h in 4% paraformaldehyde and 1% glutaraldehyde in 0.1 M phosphate buffer (pH 7.2). After washing in PBS, cells were post-fixed with 2% osmium tetroxide for 1 h. Samples were dehydrated in graded ethanol and propylene oxide solutions, then impregnated with a 1:1 mixture of propylene oxide/Epon resin and left overnight in pure resin. The samples were embedded in Epon resin and polymerized at 60 °C for 48 h. Ultra-thin sections (90 nm) were cut using a Leica EM UC7 ultra-microtome, stained with 2% uranyl acetate and 5% lead citrate, and analyzed with a JEOL 1011 transmission electron microscope using Digital Micrograph software.

### Scanning electron microscopy

Cells were washed with PBS, detached using trypsin, and centrifuged. They were fixed for 24 h in 4% paraformaldehyde and 1% glutaraldehyde in 0.1 M phosphate buffer (pH 7.2). After washing in PBS, samples were post-fixed with 2% osmium tetroxide for 1 h. Samples were dehydrated in a graded ethanol series, then dried in hexamethyldisilazane. The dry samples were placed onto carbon disks and coated with 40 Å of platinum using a GATAN PECS 682 apparatus. Observations were made with a Zeiss Ultra Plus FEG-SEM scanning electron microscope.

### Confocal fluorescence microscopy

BEAS-2B cells were cultured in 12-well plates on cover slides. After treatments, cells were fixed with 4% formaldehyde for 30 min at room temperature and permeabilized with 0.1% Triton X-100 in PBS for 30 min. After blocking with PBS containing 1% BSA and 0.1% Tween 20 for 1 h, cells were stained for 2 h at room temperature with anti-NP-FITC (1/30), anti-NS1 (1/200), and anti-PA (1/50) antibodies. Anti-rabbit-AF488 (2 h at room temperature) served as the secondary antibody for NS1, and anti-mouse-AF488 was used for PA.

PBECs were fixed with 4% formaldehyde for 10 min at 4 °C, then permeabilized with cold methanol for 10 min at 4 °C. After blocking with PBS containing 1% bovine serum albumin and

0.3% Triton X-100 for 10 min, cells were stained for 2 h at room temperature with anti-p63 (1/100), anti-Tubulin (1/100), and anti-Mucin 5AC (1/1000) antibodies. Secondary antibodies used were anti-rabbit-AF546 (for p63), anti-mouse-AF488 (for Mucin 5AC), and anti-mouse-AF647 (for Tubulin). Nuclei were stained with NucBlue reagent for 5 min.

OLC were fixed overnight at 4 °C in 4% formaldehyde. Aldehyde groups were quenched with two 10-min incubations in PBS with 0.1% glycine, followed by permeabilization with 0.5% Triton X-100 in PBS for 15 min at room temperature. After 2 h of saturation in PBS with 1% BSA, 0.5% Triton X-100, cells were stained overnight at 4 °C with anti-Tubulin (1/500) and anti-NP-FITC (1/50) antibodies. The secondary antibody anti-mouse-AF647 was applied for 2 h at room temperature for Tubulin staining. Nuclei were stained with Hoechst reagent (1/2000) for 10 min. Samples were analyzed, and 3D reconstructions were generated using a Leica SP8 confocal microscope with Leica LasX Life Sciences Software.

### Western blotting

Cells in six-well plates were lysed with 150 µL of RIPA buffer (with protease inhibitors or PhosphoSafe Extraction Reagent). After centrifugation at 12,000× g for 10 min, protein concentration was measured using the Pierce™ BCA Kit. Ten µg of protein were mixed with Laemmli buffer, heated at 100 °C for 5 min, and separated on 12% SDS-PAGE gels. Proteins were transferred to nitrocellulose membranes and probed with primary antibodies: anti-NP (1/500), anti-NS1 (1/1000), anti-PA (1/1000), anti-(P)ERK1/2 (1/2000), anti-(P)AKT (1/1000), anti-(P)p65 (1/3000), or anti-β-actin (1/5000). HRP-conjugated secondary antibodies were used for detection, followed by ECL. Protein bands were visualized using an automated imaging system and analyzed with FUJI FILM MultiGauge software.

### Minigenome assay

The minigenome studies were performed in 24-well plates. Briefly, HEK-293T cells were transfected with together with 50 ng of pRF483-PA-RT, 50 ng of pRF483-PB2-RT, 100 ng of pRF483-NP-RT, 50 ng of pRF483-PB1-RT and 150 ng of reporter plasmid pPolI-WSN-NA-firefly luciferase which contains a firefly luciferase ORF flanked by the noncoding regions of the NA segment under the control of human polymerase I promoter. As a negative control, HEK-293T cells were transfected with the same plasmids, with the exception of the PB1 plasmid. The procedure used the Fugene HD transfection reagent according to the manufacturer's instructions. In all, 20 h post-transfection cells were treated with different concentrations of cis-aco. 48 h post-transfection, cells were washed twice with PBS and lysed in 100 µl of lysis buffer provided with the Firefly Luciferase Assay System. Firefly luciferase activities were measured on 20 µl of cell extracts, using the Firefly luciferase substrate provided with the above-mentioned kit and a Centro luminometer (Berthold).

### Incucyte® and cell death assays

PBEC were seeded in 96-well plates in the presence of 2 µM of the fluorescent dye SYTOX™, or 1 µM of propidium iodide (PI) which binds to DNA and rapidly penetrates dying cells upon membrane permeabilization, and then infected with IAV. Real-time cell death assays were performed using an IncuCyte® two-color incubator imaging system. The images obtained were analyzed using the software supplied with the IncuCyte imager, which enables precise analysis of the number of SYTOX™-positive cells present in each image. Experiments were carried out using a minimum of two separate wells for each experimental condition and a minimum of four image fields per well.

### Animal infection and fluid collection

Seven-week-old C57Bl/6 mice (female or male) were intranasally challenged with 10 µg LPS or 200 pfu A/Scotland/20/74 (H3N2) IAV, and treated with 0.6 mg cis-aco (30 mg/kg) at various time points. Blood was collected on the sacrifice day, centrifuged at 10,000× g for 10 min for serum analysis or heparinized for analysis on a ProCyte Dx hematocytometer. Airways were lavaged with 4 × 0.5 ml PBS for BAL collection, and lungs were perfused with 10 ml PBS injected into the heart. The left lung was fixed in 4% paraformaldehyde for histology. Right lungs were digested enzymatically using the gentleMACS dissociator, and after centrifugation, lung suspensions and BAL fluids were stored at -80 °C for subsequent inflammatory mediator analysis. Leukocytes were isolated, red blood cells lysed, and leukocytes counted via flow cytometry.

BALB/c NF-κB transgenic mice were used in Ronan Le Goffic's lab (Chevalier et al, 2021). Mice were challenged with 10 µg LPS or infected with 300 PFU A/Scotland/20/74 (H3N2) IAV. At 1 or 8 days p.i., luciferin (0.75 mg/kg) was administered intranasally, and luciferase activity was measured using the IVIS system.

CAD-deficient C57Bl/6 mice were used in Priscille Brodin and François Trottein's labs. CAD-deficient and wild-type mice (20 animals per group) were infected with 100 PFU A/Scotland/20/74 (H3N2) IAV and treated with 30 mg/kg cis-aco 20 min p.i. Body weight loss and survival were monitored daily.

### Genomic DNA extraction and 16S rRNA sequencing analysis

Genomic DNA was extracted from mouse fecal pellets, and 16S rRNA sequencing was performed as previously described (Dhaliwal et al, 2024). The taxonomy of each amplicon sequence variant (ASV) was assigned based on the SILVA database v1.3.8 (Quast et al, 2013). ASVs unclassified at the kingdom or phylum level or ASVs classified as Eukaryota or Mitochondria were excluded. Aitchison distances were measured using the microbiome (http://microbiome.github.io) and phyloseq packages (McMurdie and Holmes, 2013) in RStudio v4.1.2.

### Flow cytometry analysis

BAL, lungs, or human bronchial epithelial cells were dispensed into round-bottomed 96-well plates and centrifuged at 300× g at 4 °C for 5 min. Samples were further stained using specific antibodies and appropriate isotype controls (listed in the Reagents table). For each antibody, one well was seeded for the Fluorescence Minus One Control. Dead cells were excluded using the LIVE/DEAD cell staining kit (Invitrogen). Flow cytometry data were acquired on a MACSQuant® Analyzer (Miltenyi Biotec), and analyses were performed using the VenturiOne software (Applied Cytometry). The gating strategy is presented in Appendix Fig. S5.

### Histopathology

Lungs were collected after BAL, and airways were washed and placed in 4% paraformaldehyde in PBS. Lung sections of ~4 µm thickness were cut and stained with hematoxylin-eosin at the LAPV

**The paper explained**

**Problem**

Influenza causes substantial global mortality each year. Current anti-viral treatments are limited by viral resistance and require early administration to be effective. Severe disease arises from both viral replication and excessive host inflammation, yet most existing therapies primarily target the virus.

**Results**

We show that the mitochondrial metabolite *cis*-aconitate has a dual antiviral and immunomodulatory activity against multiple influenza A and B strains. It inhibits viral replication and simultaneously dampens harmful inflammatory responses. In mouse models of influenza pneumonia, *cis*-aconitate reduces viral burden and inflammation and significantly improves survival, even when administered at late stages of infection.

**Impact**

*Cis*-aconitate represents a promising host-directed therapeutic strategy that overcomes key limitations of current antivirals. Its dual mechanism of action and extended treatment window support further development toward clinical application.

number of replicates (*n*) and the statistical tests used, are provided in the figure legends as well as in the Dataset EV1. Statistical significance is indicated as follows: *$P < 0.05$, **$P < 0.01$, ***$P < 0.001$, ****$P < 0.0001$. For in vitro experiments, "*n*" refers to the number of independent experiments, whereas for in vivo studies, "*n*" refers to the number of individual animals.

## Data availability

This study includes no data deposited in external repositories.

The source data of this paper are collected in the following database record: biostudies:S-SCDT-10_1038-S44321-026-00379-8.

## Peer review information

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

(Amboise, France). A study pathologist examined the tissue sections using light microscopy on a Leica Diaplan microscope in a blinded experimental protocol. All histopathological findings were graded in a semi-quantitative fashion on a scale of 0 to 4 (0: absent, 1: mild, 2: moderate, 3: severe, 4: extremely severe).

### Cell proliferation and cytotoxicity assays

Cells in 96-well plates were washed twice with PBS and incubated for 1 h at 37 °C with 100 μL of MTS reagent diluted 1/5 for the cell proliferation test. Optical density was measured at 490 nm. Cells were stained for 15 min at 4 °C with Live/Dead (1/1000$^e$), anti-Ki67 (1/100$^e$) or MitoTracker (1/5000$^e$), or 5 min at 37 °C with Dihydrorhodamine(DHR)-123 (1/100$^e$) before flow cytometry analysis.

### Mitochondrial respiration assay

Cells were seeded in poly-L-Lysine-coated, XF 24-wells cell culture microplates (Agilent), at $5 \times 10^4$ cells/well in 250 μL F-12K medium with Glutamax supplemented with 10% FCS (BioSera) and 1% penicillin/streptomycin. After 24 h, the medium was replaced, and cells were treated or not with 3.4 mM *cis*-aco for 20 h. The MitoStress Test assay (Agilent) was performed in 500 μL of Seahorse assay medium (XF DMEM pH 7.4 + 10 mM Glucose, 2 mM Glutamine, and 1 mM sodium pyruvate) following the manufacturer's recommendations. The oxygen consumption rate (OCR) was measured using the Extracellular Flux Analyzer (XFe24 Seahorse Agilent) at the basal stage, after injection of 1.5 μM Oligomycin, 0.5 μM FCCP, and 1 μM of Rotenone + Antimycin A. Cell numbers were determined post-assay by Hoechst staining and counted using the Cytation 1 image reader (Biotek), and OCR values were normalized to cell number.

### Statistical analyses

Statistical analyses were performed using GraphPad Prism. Data are presented as mean ± SEM. Statistical details, including the

with proinflammatory signaling and respiratory distress in influenza-infected mice. J Biol Chem 297:100885

Chiang N, Serhan CN (2020) Specialized pro-resolving mediator network: an update on production and actions. Essays Biochem 64:443–462

Dai X, Zhang L, Hong T (2011) Host cellular signaling induced by influenza virus. Sci China Life Sci 54:68–74

Davidson S (2018) Treating influenza infection, from now and into the future. Front Immunol 9:1946

Demars A, Vitali A, Comein A, Carlier E, Azouz A, Goriely S, Smout J, Flamand V, Van Gysel M, Wouters J et al (2021) Aconitate decarboxylase 1 participates in the control of pulmonary Brucella infection in mice. PLoS Pathog 17:e1009887

Dequin P-F, Meziani F, Quenot J-P, Kamel T, Ricard J-D, Badie J, Reignier J, Heming N, Plantefève G, Souweine B et al (2023) Hydrocortisone in severe community-acquired pneumonia. N Engl J Med 388:1931–1941

Dhaliwal J, Tertigas D, Carman N, Lawrence S, Debruyn JC, Wine E, Church PC, Huynh HQ, Rashid M, El-Matary W et al (2024) Outcomes following acute severe colitis at initial presentation: a multi-centre, prospective, paediatric cohort study. J Crohns Colitis 18:233–245

Ding Y, Cao Z, Cao L, Ding G, Wang Z, Xiao W (2017) Antiviral activity of chlorogenic acid against influenza A (H1N1/H3N2) virus and its inhibition of neuraminidase. Sci Rep 7:45723

Dou D, Revol R, Östbye H, Wang H, Daniels R (2018) Influenza A virus cell entry, replication, virion assembly and movement. Front Immunol 9:1581

Ferren M, Favède V, Decimo D, Iampietro M, Lieberman NAP, Weickert J-L, Pelissier R, Mazelier M, Terrier O, Moscona A et al (2021) Hamster organotypic modeling of SARS-CoV-2 lung and brainstem infection. Nat Commun 12:5809

Gao Y, Guyatt G, Uyeki TM, Liu M, Chen Y, Zhao Y, Shen Y, Xu J, Zheng Q, Li Z et al (2024) Antivirals for treatment of severe influenza: a systematic review and network meta-analysis of randomised controlled trials. Lancet 404:753–763

Garot D, Tchatat Wangueu L, Larrat C, Guillon A (2022) Mechanically ventilated COVID-19 patients failed to meet the criteria for the Berlin definition of ARDS. Infection 50:545–546

Gaur P, Munjhal A, Lal SK (2011) Influenza virus and cell signaling pathways. Med Sci Monit 17:RA148–RA154

Geraghty RJ, Capes-Davis A, Davis JM, Downward J, Freshney RI, Knezevic I, Lovell-Badge R, Masters JRW, Meredith J, Stacey GN et al (2014) Guidelines for the use of cell lines in biomedical research. Br J Cancer 111:1021–1046

Guillon A, Arafa EI, Barker KA, Belkina AC, Martin I, Shenoy AT, Wooten AK, Lyon De Ana C, Dai A, Labadorf A et al (2020) Pneumonia recovery reprograms the alveolar macrophage pool. JCI Insight 5:133042

Guillon A, Brea-Diakite D, Cezard A, Wacquiez A, Baranek T, Bourgeais J, Picou F, Vasseur V, Meyer L, Chevalier C et al (2022) Host succinate inhibits influenza virus infection through succinylation and nuclear retention of the viral nucleoprotein. EMBO J 41:e108306

Haasbach E, Müller C, Ehrhardt C, Schreiber A, Pleschka S, Ludwig S, Planz O (2017) The MEK-inhibitor CI-1040 displays a broad anti-influenza virus activity in vitro and provides a prolonged treatment window compared to standard of care in vivo. Antivir Res 142:178–184

Hernu R, Chroboczek T, Madelaine T, Casalegno J-S, Lina B, Cour M, Argaud L, On behalf of the "Flu in Lyon ICUs" Study Group (2018) Early oseltamivir therapy improves the outcome in critically ill patients with influenza: a propensity analysis. Intensive Care Med 44:257–260

Herold S, Becker C, Ridge KM, Budinger GRS (2015) Influenza virus-induced lung injury: pathogenesis and implications for treatment. Eur Respir J 45:1463–1478

Hong E-H, Song J-H, Kim S-R, Cho J, Jeong B, Yang H, Jeong J-H, Ahn J-H, Jeong H, Kim S-E et al (2020) Morin hydrate inhibits influenza virus entry into host cells and has anti-inflammatory effect in influenza-infected mice. Immune Netw 20:e32

Hughes P, Marshall D, Reid Y, Parkes H, Gelber C (2007) The costs of using unauthenticated, over-passaged cell lines: how much more data do we need?. Biotechniques 43:575, 577–578

Jefferson T, Jones MA, Doshi P, Del Mar CB, Hama R, Thompson MJ, Spencer EA, Onakpoya I, Mahtani KR, Nunan D et al (2014) Neuraminidase inhibitors for preventing and treating influenza in healthy adults and children. Cochrane Database Syst Rev 2014:CD008965

Kim S-H, Hong S-B, Yun S-C, Choi W-I, Ahn J-J, Lee YJ, Lee H-B, Lim C-M, Koh Y, Korean Society of Critical Care Medicine H1N1 Collaborative (2011) Corticosteroid treatment in critically ill patients with pandemic influenza A/H1N1 2009 infection: analytic strategy using propensity scores. Am J Respir Crit Care Med 183:1207–1214

Lam M, Lamanna E, Organ L, Donovan C, Bourke JE (2023) Perspectives on precision cut lung slices-powerful tools for investigation of mechanisms and therapeutic targets in lung diseases. Front Pharm 14:1162889

Le Goffic R, Balloy V, Lagranderie M, Alexopoulou L, Escriou N, Flavell R, Chignard M, Si-Tahar M (2006) Detrimental contribution of the Toll-like receptor (TLR) 3 to influenza A virus-induced acute pneumonia. PLoS Pathog 2:e53

Lechner JF, Haugen A, McClendon IA, Pettis EW (1982) Clonal growth of normal adult human bronchial epithelial cells in a serum-free medium. Vitro 18:633–642

Lhommet C, Garot D, Grammatico-Guillon L, Jourdannaud C, Asfar P, Faisy C, Muller G, Barker KA, Mercier E, Robert S et al (2020) Predicting the microbial cause of community-acquired pneumonia: can physicians or a data-driven method differentiate viral from bacterial pneumonia at patient presentation?. BMC Pulm Med 20:62

Li C-C, Wang X-J, Wang H-CR (2019) Repurposing host-based therapeutics to control coronavirus and influenza virus. Drug Discov Today 24:726–736

Luig C, Köther K, Dudek SE, Gaestel M, Hiscott J, Wixler V, Ludwig S (2010) MAP kinase-activated protein kinases 2 and 3 are required for influenza A virus propagation and act via inhibition of PKR. FASEB J 24:4068–4077

Marathe BM, Wong S-S, Vogel P, Garcia-Alcalde F, Webster RG, Webby RJ, Najera I, Govorkova EA (2016) Combinations of oseltamivir and T-705 extend the treatment window for highly pathogenic influenza A(H5N1) Virus infection in mice. Sci Rep 6:26742

Martínez-Reyes I, Chandel NS (2020) Mitochondrial TCA cycle metabolites control physiology and disease. Nat Commun 11:102

Matthay MA, Liu KD (2011) Con: corticosteroids are not indicated for treatment of acute lung injury from H1N1 viral pneumonia. Am J Respir Crit Care Med 183:1127–1128

Matute-Bello G, Frevert CW, Martin TR (2008) Animal models of acute lung injury. Am J Physiol Lung Cell Mol Physiol 295:L379–L399

McMurdie PJ, Holmes S (2013) phyloseq: an R package for reproducible interactive analysis and graphics of microbiome census data. PLoS ONE 8:e61217

Michelucci A, Cordes T, Ghelfi J, Pailot A, Reiling N, Goldmann O, Binz T, Wegner A, Tallam A, Rausell A et al (2013) Immune-responsive gene 1 protein links metabolism to immunity by catalyzing itaconic acid production. Proc Natl Acad Sci USA 110:7820–7825

Müller KH, Kakkola L, Nagaraj AS, Cheltsov AV, Anastasina M, Kainov DE (2012) Emerging cellular targets for influenza antiviral agents. Trends Pharmacol Sci 33:89–99

Ni Y-N, Chen G, Sun J, Liang B-M, Liang Z-A (2019) The effect of corticosteroids on mortality of patients with influenza pneumonia: a systematic review and meta-analysis. Crit Care 23:99

O'Carroll SM, O'Neill LAJ (2021) Targeting immunometabolism to treat COVID-19. Immunother Adv 1:ltab013

Olagnier D, Farahani E, Thyrsted J, Blay-Cadanet J, Herengt A, Idorn M, Hait A, Hernaez B, Knudsen A, Iversen MB et al (2020) SARS-CoV2-mediated suppression of NRF2-signaling reveals potent antiviral and anti-inflammatory activity of 4-octyl-itaconate and dimethyl fumarate. Nat Commun 11:4938

O'Neill LAJ, Kishton RJ, Rathmell J (2016) A guide to immunometabolism for immunologists. Nat Rev Immunol 16:553–565

Pålsson-McDermott EM, O'Neill LAJ (2020) Targeting immunometabolism as an anti-inflammatory strategy. Cell Res 30:300–314

Paludan SR, Pradeu T, Masters SL, Mogensen TH (2021) Constitutive immune mechanisms: mediators of host defence and immune regulation. Nat Rev Immunol 21:137–150

Pearce EJ, Pearce EL (2018) Immunometabolism in 2017: driving immunity: all roads lead to metabolism. Nat Rev Immunol 18:81–82

Pleschka S, Wolff T, Ehrhardt C, Hobom G, Planz O, Rapp UR, Ludwig S (2001) Influenza virus propagation is impaired by inhibition of the Raf/MEK/ERK signalling cascade. Nat Cell Biol 3:301–305

Quast C, Pruesse E, Yilmaz P, Gerken J, Schweer T, Yarza P, Peplies J, Glöckner FO (2013) The SILVA ribosomal RNA gene database project: improved data processing and web-based tools. Nucleic Acids Res 41:D590–D596

Rambold AS, Pearce EL (2018) Mitochondrial dynamics at the interface of immune cell metabolism and function. Trends Immunol 39:6–18

Rao M, Dodoo E, Zumla A, Maeurer M (2019) Immunometabolism and pulmonary infections: implications for protective immune responses and host-directed therapies. Front Microbiol 10:962

Reddel RR, Ke Y, Gerwin BI, McMenamin MG, Lechner JF, Su RT, Brash DE, Park JB, Rhim JS, Harris CC (1988) Transformation of human bronchial epithelial cells by infection with SV40 or adenovirus-12 SV40 hybrid virus, or transfection via strontium phosphate coprecipitation with a plasmid containing SV40 early region genes. Cancer Res 48:1904–1909

Rodríguez A, Díaz E, Martín-Loeches I, Sandiumenge A, Canadell L, Díaz JJ, Figueira JC, Marques A, Alvarez-Lerma F, Vallés J et al (2011) Impact of early oseltamivir treatment on outcome in critically ill patients with 2009 pandemic influenza A. J Antimicrob Chemother 66:1140–1149

Schmolke M, Viemann D, Roth J, Ludwig S (2009) Essential impact of NF-kappaB signaling on the H5N1 influenza A virus-induced transcriptome. J Immunol 183:5180–5189

Schräder T, Dudek SE, Schreiber A, Ehrhardt C, Planz O, Ludwig S (2018) The clinically approved MEK inhibitor Trametinib efficiently blocks influenza A virus propagation and cytokine expression. Antivir Res 157:80–92

Schreiber A, Boff L, Anhlan D, Krischuns T, Brunotte L, Schuberth C, Wedlich-Söldner R, Drexler H, Ludwig S (2020) Dissecting the mechanism of signaling-triggered nuclear export of newly synthesized influenza virus ribonucleoprotein complexes. Proc Natl Acad Sci USA 117:16557–16566

Scossa F, Fernie AR (2020) The evolution of metabolism: How to test evolutionary hypotheses at the genomic level. Comput Struct Biotechnol J 18:482–500

Sethy B, Hsieh C-F, Lin T-J, Hu P-Y, Chen Y-L, Lin C-Y, Tseng S-N, Horng J-T, Hsieh P-W (2019) Design, synthesis, and biological evaluation of itaconic acid derivatives as potential anti-influenza agents. J Med Chem 62:2390–2403

Seymour CW, Gesten F, Prescott HC, Friedrich ME, Iwashyna TJ, Phillips GS, Lemeshow S, Osborn T, Terry KM, Levy MM (2017) Time to treatment and mortality during mandated emergency care for sepsis. N Engl J Med 376:2235–2244

Sohail A, Iqbal AA, Sahini N, Chen F, Tantawy M, Waqas SFH, Winterhoff M, Ebensen T, Schultz K, Geffers R et al (2022) Itaconate and derivatives reduce interferon responses and inflammation in influenza A virus infection. PLoS Pathog 18:e1010219

Soto-Heredero G, Gómez de las Heras MM, Gabandé-Rodríguez E, Oller J, Mittelbrunn M (2020) Glycolysis–a key player in the inflammatory response. FEBS J 287:3350–3369

Tavares LP, Teixeira MM, Garcia CC (2017) The inflammatory response triggered by Influenza virus: a two edged sword. Inflamm Res 66:283–302

Te Velthuis AJW, Long JS, Barclay WS (2018) Assays to measure the activity of influenza virus polymerase. Methods Mol Biol 1836:343–374

World Health Organization (2010) WHO guidelines for pharmacological management of pandemic influenza A (H1N1) 2009 and other influenza viruses. World Health Organization, Geneva

Yan Y-Q, Fu Y-J, Wu S, Qin H-Q, Zhen X, Song B-M, Weng Y-S, Wang P-C, Chen X-Y, Jiang Z-Y (2018) Anti-influenza activity of berberine improves prognosis by reducing viral replication in mice. Phytother Res 32:2560–2567

Yu J, Sun X, Goie JYG, Zhang Y (2020) Regulation of host immune responses against influenza A virus infection by mitogen-activated protein kinases (MAPKs). Microorganisms 8:1067

## Acknowledgements

The authors are grateful to Prof. Pieter Hiemstra (Laboratory for Respiratory Cell Biology and Immunology of the Department of Pulmonology of the Leiden University Medical Center (LUMC)) for his collaboration in setting up the ex vivo culture model of primary human lung epithelial cells. We also thank Benoit Briard and Sandra Khau (CEPR, Inserm, Tours) for their advice on Incucyte® imaging and cell death assays. This work was partially supported by grants from Inserm, Université de Tours, Région Centre-Val de Loire (FLU-MET#2018-00124196; to MS-T), FEDER Euro-FERI (to MS-T and CP) and the ANR programs ANR-21-CE18-0061 ("SuccesS"; to MS-T) and ANR-22-ASTR-0021 ("VIROMETABLOCK"; to MS-T, POV, LPC, VL and CM).

## Author contributions

**Adeline Cezard**: Data curation; Formal analysis; Validation; Investigation; Visualization; Writing—original draft; Writing—review and editing. **Déborah Brea-Diakite**: Formal analysis; Validation; Investigation; Visualization; Writing—review and editing. **Virginie Vasseur**: Formal analysis; Validation; Investigation; Writing—review and editing. **Alan Wacquiez**: Formal analysis; Investigation. **Loic Gonzalez**: Formal analysis; Investigation. **Ronan Le Goffic**: Formal analysis; Investigation; Visualization; Writing—review and editing. **Bruno Da Costa**: Investigation. **Ambre Tinard**: Investigation. **Delphine Fouquenet**: Investigation. **Séverine Heumel**: Investigation. **Arnaud Machelart**: Formal analysis; Investigation; Writing—review and editing. **Eik Hoffmann**: Formal analysis; Investigation; Writing—review and editing. **Priscille Brodin**: Methodology; Writing—review and editing. **François Trottein**: Formal analysis; Writing—review and editing. **Cyrille Mathieu**: Formal analysis; Methodology; Writing—review and editing. **Lola Canus**: Investigation. **Florentine Jacolin**: Investigation. **Pierre-Olivier Vidalain**: Formal analysis; Methodology; Writing—review and editing. **Laure Perrin-Cocon**: Formal analysis; Methodology; Writing—review and editing. **Vincent Lotteau**: Formal analysis. **Julien Burlaud-Gaillard**: Investigation. **Dominique Tertigas**: Investigation; Methodology; Writing—review and editing. **Michael G Surette**: Formal analysis. **Antoine Legras**: Resources. **Damien Sizaret**: Resources. **Thomas Baranek**: Formal analysis; Writing—review and editing. **Christophe Paget**: Formal analysis; Writing—review and editing. **Antoine Guillon**: Conceptualization; Formal analysis; Writing—review and editing. **Mustapha Si-Tahar**: Conceptualization; Formal analysis; Supervision; Funding acquisition; Writing—original draft; Project administration; Writing—review and editing.

Source data underlying figure panels in this paper may have individual authorship assigned. Where available, figure panel/source data authorship is

listed in the following database record: biostudies:S-SCDT-10_1038-S44321-026-00379-8.

## Disclosure and competing interests statement

MS-T and AG declared the deposit of a patent application related to the anti-influenza activity of *cis*-aconitate (reference: WO 2024126742).

# Expanded View Figures

**Figure EV1.   Tolerance of bronchial epithelial cells to metabolite exposure.**

(**A**) BEAS-2B cells were treated with medium alone or with 3.4 mM *cis*-aco (CA), itaconate (Ita), oxaloacetate (Oxa), isocitrate (IsoC), citrate (Cit), fumarate (Fum), pyruvate (Pyr) or glucose (Glc) for 16 h. Cytotoxicity was assessed using the MTS assay. (**B–E**) BEAS-2B cells were treated (blue) or not (white) with 1.2 or 2.3 or 3.4 mM *cis*-aco for 20 h (**B**) or with 3.4 mM *cis*-aco for 6 or 24 h (**C–E**). Effects on cell viability (**B**), proliferation (**C**), mitochondrial labeling (**D**), and ROS production (**D**) were analyzed using Live Dead, Ki67, mitotracker and DHR123 staining, respectively. (**F, G**) PBEC were treated with 2.3, 3.4 or 5.7 mM *cis*-aco for 16 h, and cytotoxicity was assessed using the MTS assay and propidium iodide labeling. (**H**) PBEC in air–liquid interface (ALI) were treated with 5.7 mM *cis*-aco for 44 h, and cytotoxicity was assessed using the MTS assay. Data are presented as the mean ± SEM from 4 (**B–F**) or 5 (**A**) independent experiments. The number of data points shown in each bar plot corresponds to the number of independent experiments performed for that condition. Statistical analyses were performed using the Kruskal–Wallis test with Dunn's multiple comparison test (**A, B, F**), the Friedman test with Dunn's multiple comparison test (**C–E**) or the Wilcoxon matched-pairs signed rank test (**G, H**). Statistical significance: $*P < 0.05$, $**P < 0.01$, $***P < 0.001$, $****P < 0.0001$.

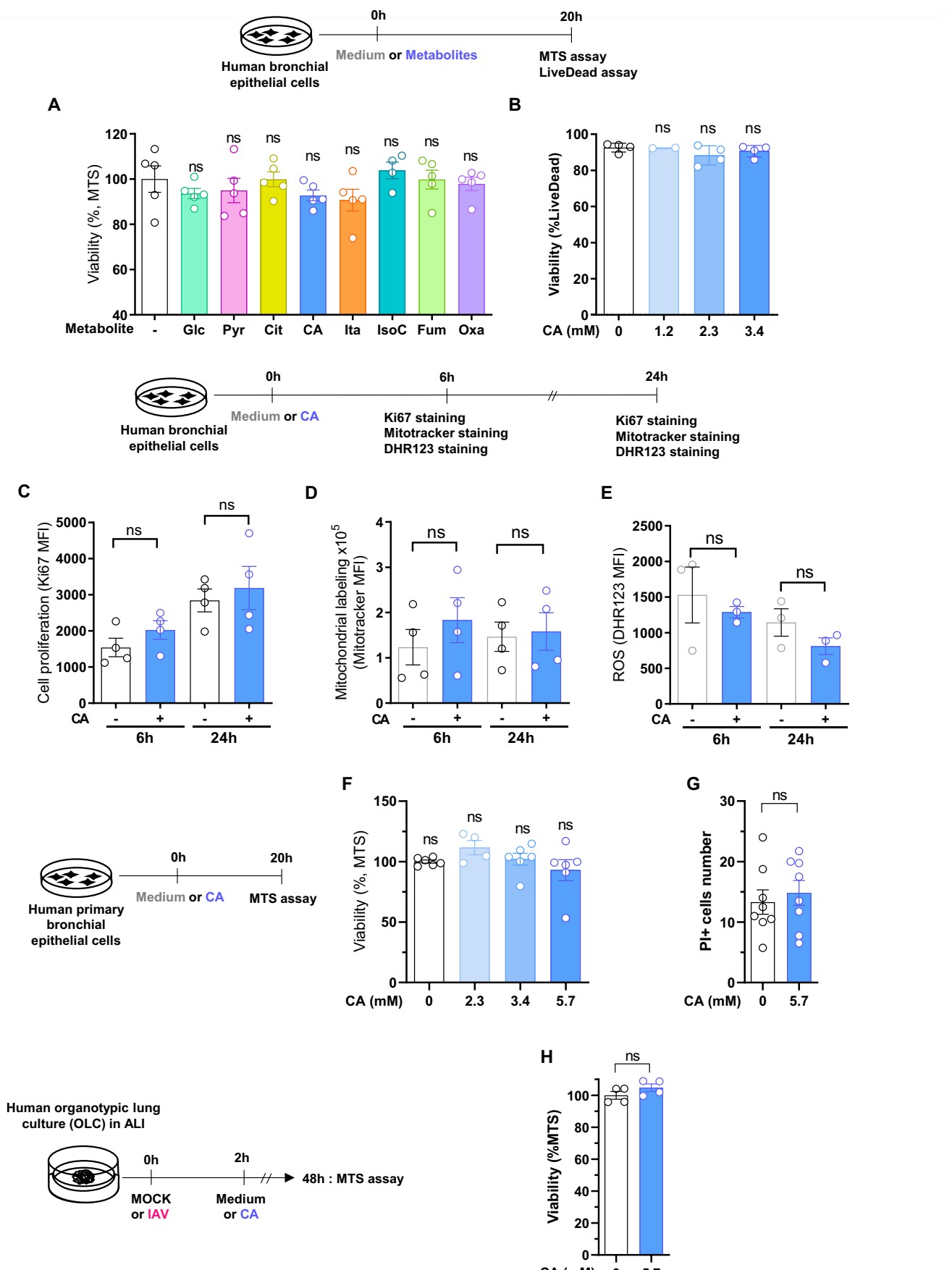

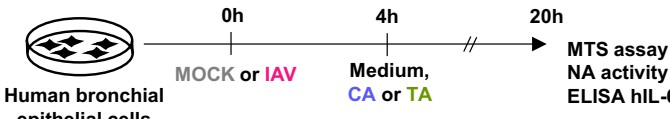

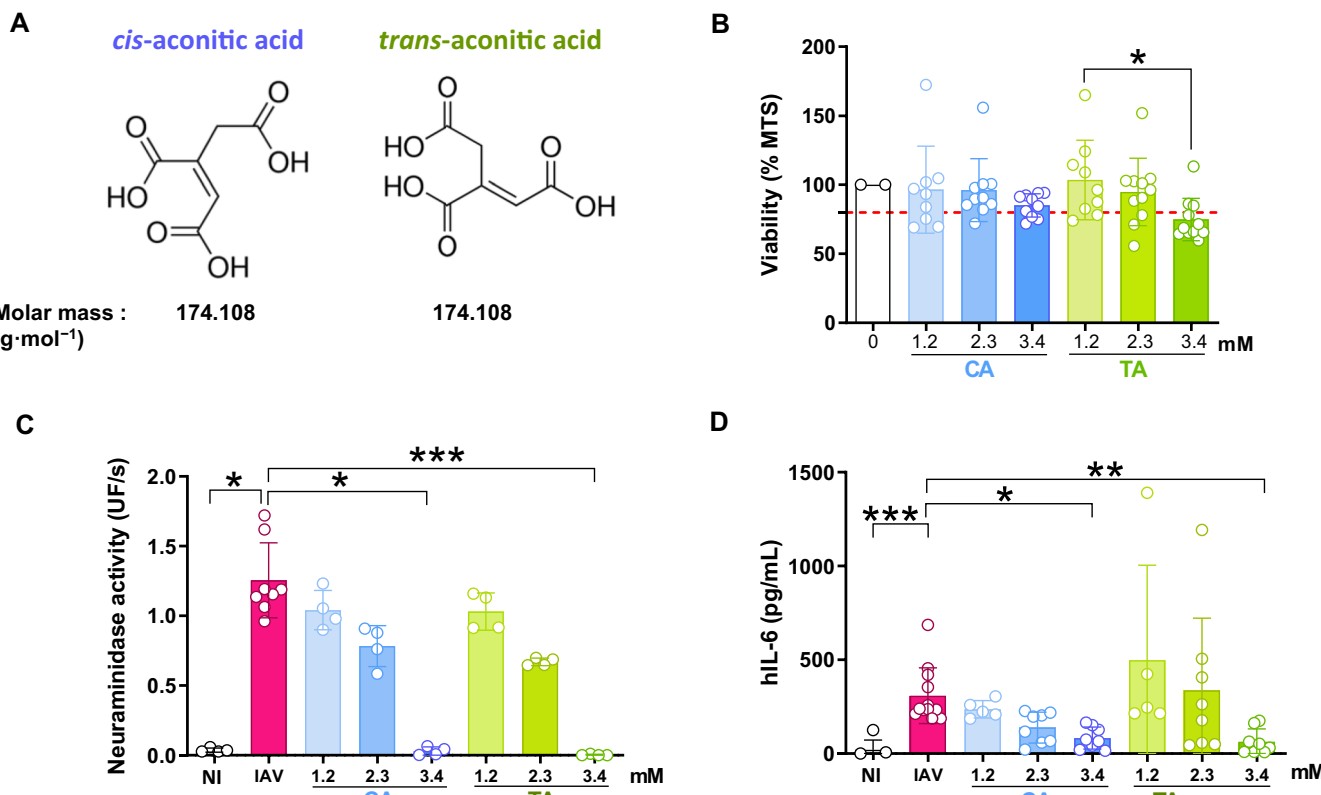

**Figure EV2. Comparison of anti-influenza effects of *cis*-aco and *trans*-aco.**

BEAS-2B cells were incubated with medium alone or with increasing concentrations of *cis*-aconitic (CA) or *trans*-aconitic acid (TA) for 20 h. The chemical structures of these metabolites are shown in (**A**). (**B**) Cell viability was assessed using the MTS assay. A viability threshold of 80% was set (red dashed line). (**C, D**) BEAS-2B cells were infected with influenza A/Scotland/20/74 (H3N2) virus (IAV) at an MOI = 1 (pink bars) or left uninfected (NI) for 4 h, followed by treatment or not with increasing concentrations of CA (blue) or TA (green) for 16 h. (**C**) Viral particle production was assessed using a neuraminidase activity assay. (**D**) IL-6 levels in cell supernatants were measured by ELISA. Data are presented as the mean ± SEM and represent cumulative results from 4 or 5 independent experiments. The number of data points shown in each bar plot corresponds to the number of independent experiments performed for that condition. Statistical analysis was performed using the Kruskal–Wallis test with Dunn's multiple comparison test. Statistical significance: *$P < 0.05$, **$P < 0.01$, ***$P < 0.001$, ****$P < 0.0001$.

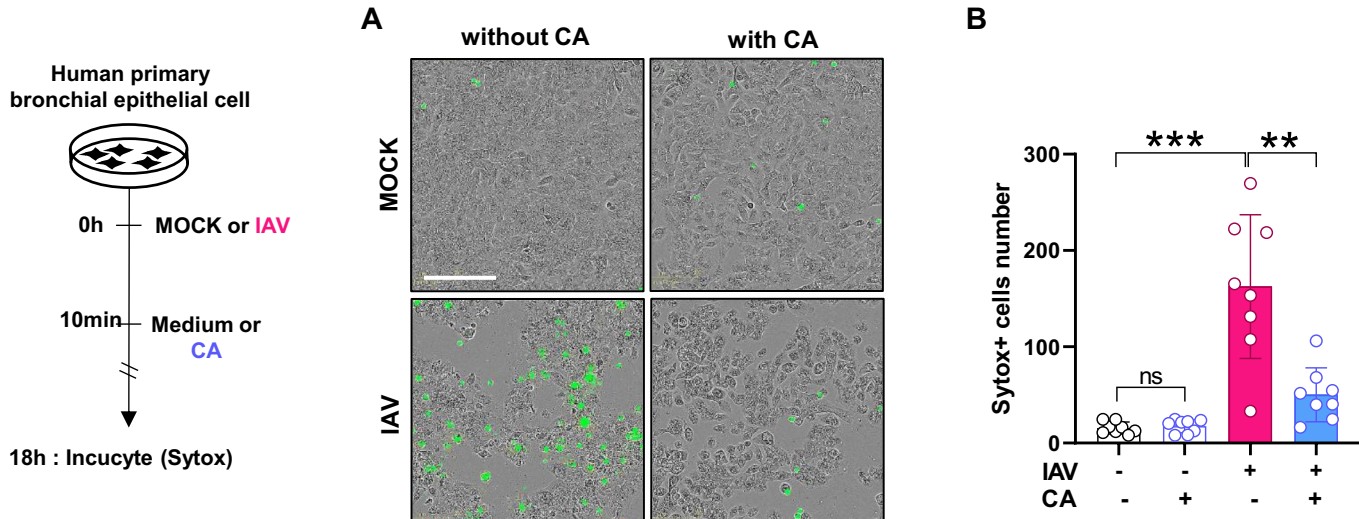

**Figure EV3.  *Cis*-aco protects against IAV-induced cell death in human primary bronchial epithelial cells (PBEC).**

(**A, B**) PBEC were infected with virus A/Scotland/20/74 (H3N2) at an MOI = 1, and treated or not with 3.4 mM *cis*-aco (CA) 10 min p.i. SYTOX™ labeling was monitored over 18 h. Representative images ((**A**); scale bar: 200 μm) and quantification of labeling (**B**) at 18 h p.i. were used to assess cell death. Data are presented as the mean ± SEM from duplicate PBEC derived from 4 independent patients analyzed. The number of data points shown in each bar plot corresponds to the number of independent experiments performed for that condition. Statistical analysis was performed using the Friedman test with Dunn's multiple comparison test. Statistical significance: *$P < 0.05$, **$P < 0.01$, ***$P < 0.001$, ****$P < 0.0001$.

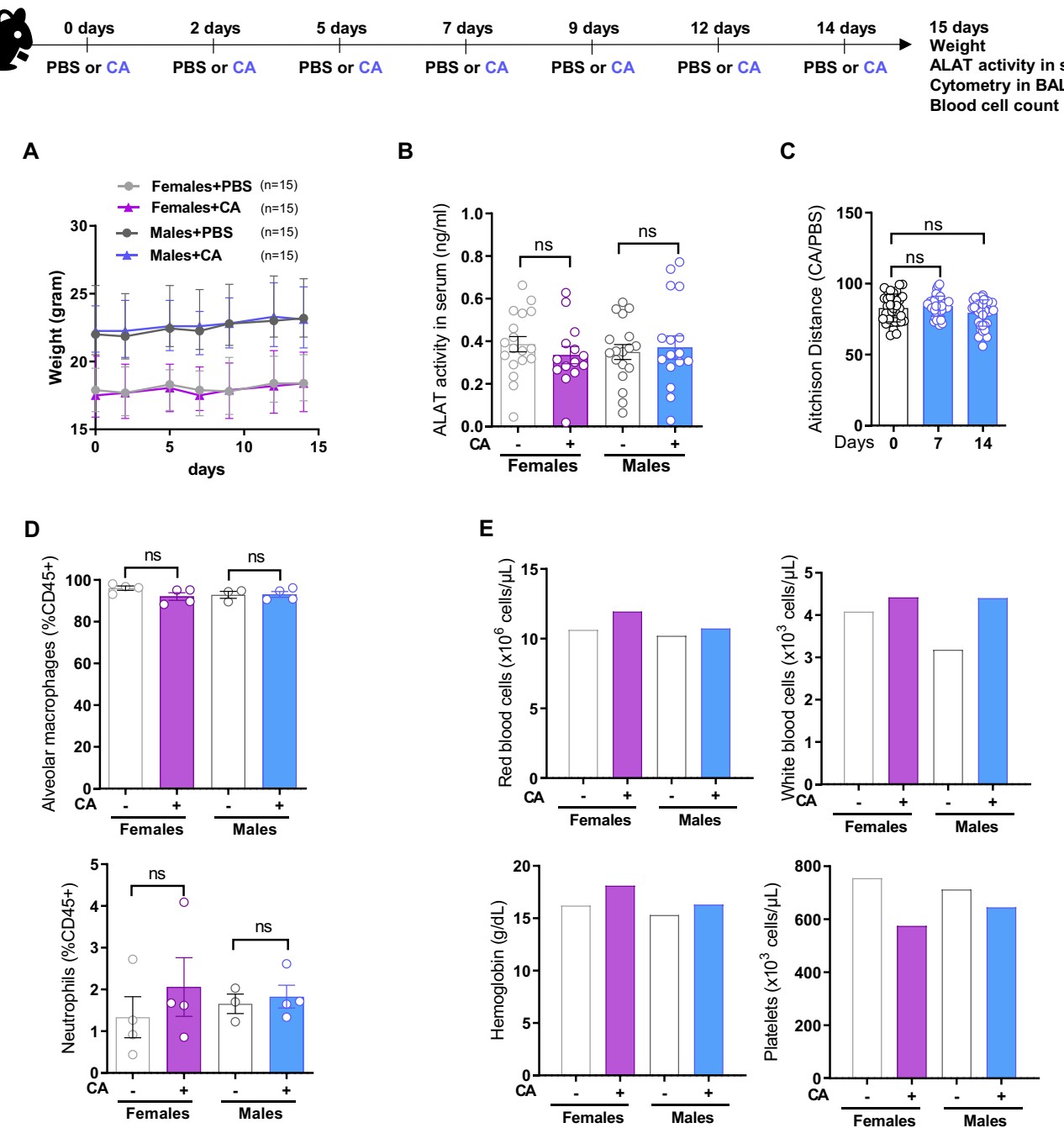

**Figure EV4. Tolerance and safety of *cis*-aco in vivo.**

Seven-week-old female and male C57Bl/6 mice were intranasally instilled with PBS or 30 mg/kg *cis*-aco (CA) for 15 days, according to the schedule outlined. (**A**) Body weight was monitored throughout the study. (**B**) Mice were euthanized on day 15 and serum was collected to determine ALAT activity levels. (**C**) Microbiota alterations were assessed in mouse fecal pellets, after genomic DNA extraction and 16S rRNA sequencing. Microbial community diversity was quantified using Aitchison distance values, a measure of beta diversity. Each point represents the Aitchison distance between a CA-treated mouse and a PBS-treated mouse at day 0, 7, or 14. (**D**) The number of immune cells in BAL fluids was assessed by flow cytometry. (**E**) Blood cell analysis was also conducted. Data from (**E**) represent pooled results from 3 mice. All other data are presented as the mean ± SEM and are cumulative (**A**–**C**) or representative (**D**) of 3 independent experiments. The number of data points shown in each bar plot corresponds to the number of independent experiments performed for that condition. Statistical analysis was performed using the Mann–Whitney test (**B**, **D**) or the Kruskal–Wallis test with Dunn's multiple comparison test. Statistical significance: *$P < 0.05$, **$P < 0.01$, ***$P < 0.001$, ****$P < 0.0001$.

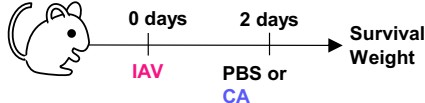

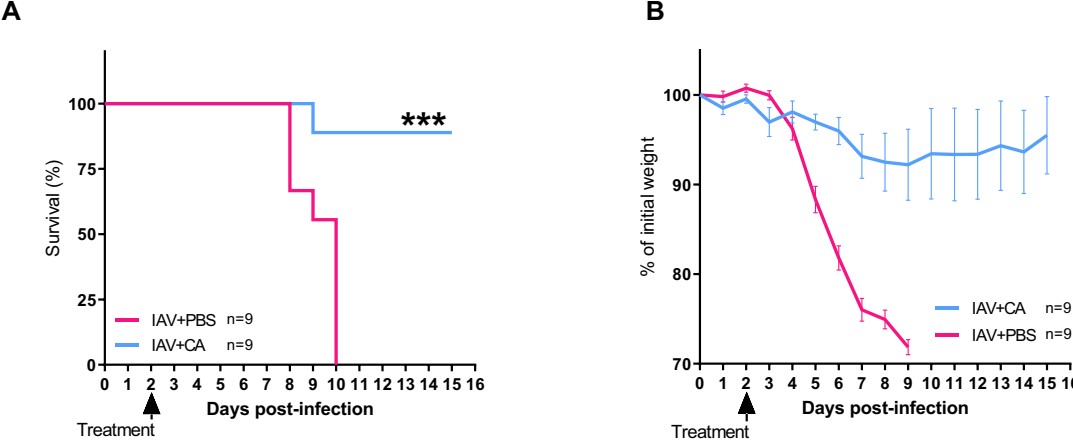

**Figure EV5. Confirmation of the protective effects of *cis*-aco in IAV-infected Balb/c mice.**

To validate the findings observed in C57Bl/6 mice (see Fig. 3A–D) in a different mouse strain and sex, we conducted an experiment using male Balb/c mice. These (NF-κB transgenic) Balb/c animals were infected with 300 PFU of A/Scotland/20/74 (H3N2) IAV and treated intranasally with 30 mg/kg *cis*-aco (CA) 2 days p.i. Survival (**A**) and body weight (**B**) were monitored daily. All data are presented as the mean ± SEM. Statistical analysis was performed using the Log-rank (Mantel–Cox) test. Statistical significance: *$P < 0.05$, **$P < 0.01$, ***$P < 0.001$, ****$P < 0.0001$.

