## [Peer Review File · EMBO Molecular Medicine]

Cis-aconitate protects against influenza by targeting viral polymerase and ERK/AKT/NF- κ B signaling

Adeline Cezard, Deborah Bréa-Diakite, Virginie Vasseur, Alan Wacquiez, Loic Gonzalez, Ronan Le Goffic, Bruno Da Costa, Ambre Tinard, Delphine Fouquenot, Séverine Heumel, Arnaud Machelart, Eik Hoffmann, Priscille Brodin, Francois Trottein, Cyrille MATHIEU, Lola Canus, Florentine Jacolin, Pierre-Olivier Vidalain, Laure Perrin-Cocon, Vincent Lotteau, Julien Burlaud-Gaillard, Dominique Tertigas, Michael Surette, Antoine Legras, Damien Sizaret, Thomas Baranek, Christophe Paget, Antoine Guillon, and Mustapha SI-TAHAR

Corresponding author: Mustapha SI-TAHAR (si-tahar@univ-tours.fr)

Review Timeline:

Submission Date:	30th Jan 25
Editorial Decision:	11th Mar 25
Revision Received:	1st Oct 25
Editorial Decision:	21st Nov 25
Revision Received:	31st Dec 25
Accepted:	15th Jan 26

Editor: Zeljko Durdevic

Transaction Report:

11th Mar 2025

Dear Dr. Si-Tahar,

Thank you for the submission of your manuscript to EMBO Molecular Medicine. We have now received feedback from the three reviewers who agreed to evaluate your manuscript. All three referees recognize interest of the study but also raise important concerns that should be addressed in a major revision. If you would like to discuss further the points raised by the referees, I am available to do so via email or video. Let me know if you are interested in this option.

We would welcome the submission of a revised version within three months for further consideration. Please let us know if you require longer to complete the revision.

I look forward to receiving your revised manuscript.

Yours sincerely,

Zeljko Durdevic

We require:

- 1) A .docx formatted version of the manuscript text (including legends for main figures, EV figures and tables). Please make sure that the changes are highlighted to be clearly visible.
- 2) Individual production quality figure files as .eps, .tif, .jpg (one file per figure). For guidance, download the 'Figure Guide PDF': (<https://www.embopress.org/page/journal/17574684/authorguide#figureformat>).
- 3) A .docx formatted letter INCLUDING the reviewers' reports and your detailed point-by-point responses to their comments. As part of the EMBO Press transparent editorial process, the point-by-point response is part of the Review Process File (RPF), which will be published alongside your paper.
- 4) A complete author checklist, which you can download from our author guidelines (<https://www.embopress.org/page/journal/17574684/authorguide#submissionofrevisions>). Please insert information in the checklist that is also reflected in the manuscript. The completed author checklist will also be part of the RPF.
- 5) Please note that all corresponding authors are required to supply an ORCID ID for their name upon submission of a revised manuscript.
- 6) It is mandatory to include a 'Data Availability' section after the Materials and Methods. Before submitting your revision, primary datasets produced in this study need to be deposited in an appropriate public database, and the accession numbers and

database listed under 'Data Availability'. Please remember to provide a reviewer password if the datasets are not yet public (see <https://www.embopress.org/page/journal/17574684/authorguide#dataavailability>).

12) Author contributions: You will be asked to provide CRediT (Contributor Role Taxonomy) terms in the submission system. These replace a narrative author contribution section in the manuscript.

13) A Conflict of Interest statement should be provided in the main text.

14) Every published paper now includes a 'Synopsis' to further enhance discoverability. Synopses are displayed on the journal webpage and are freely accessible to all readers. They include a short stand first (maximum of 300 characters, including space) as well as 2-5 one-sentences bullet points that summarizes the paper. Please write the bullet points to summarize the key NEW findings. They should be designed to be complementary to the abstract - i.e. not repeat the same text. We encourage inclusion of key acronyms and quantitative information (maximum of 30 words / bullet point). Please use the passive voice. Please attach

these in a separate file or send them by email, we will incorporate them accordingly.

15) Include a Reagents and Tools Table as part of the Methods section, which can be downloaded from our author guidelines (<https://www.embopress.org/page/journal/17574684/authorguide#structuredmethods>)

***** Reviewer's comments *****

Referee #1 (Comments on Novelty/Model System for Author):

This is a nice paper but some experimental controls are missing. Further clarifications are also required (see my review) that will likely need so metabolic analysis. That said, I am overall supportive at this stage.

Referee #1 (Remarks for Author):

This study investigates the antiviral and anti inflammatory properties of cis-aconitate (cis-aco) as a potential therapeutic for influenza. The authors propose cis-aco as a promising candidate for influenza treatment, particularly in cases where traditional antivirals (like oseltamivir) are ineffective. Overall this is a very interesting study, however several aspects of the study require further clarification and additional controls are required to strengthen the manuscript.

Major comments:

- Are levels of itaconate or cis-aco , or other metabolites screened altered in lung of infected mice? How about expression of appropriate metabolic enzymes?
- Is aconitase upregulated during infection?
- Relating to the above, is aconitase or cis-aco increased in human IAV treated ECs?
- Were metabolic responses between BEAS2B and primary cells different?
- What is the half life of aconitase in vivo?
- Please make it clearer in text the source of primary ECs (although this is in methods, it should be mentioned in text
- In murine models, please make it clearer how cis-aco was given and frequency/dose in schematics
- Please provide CA only controls for all mouse studies
- The human data in figure 8 seems a bit out of the blue and very hard to directly relate this timing to the murine model, I am not sure if it is useful in this context.

Referee #2 (Comments on Novelty/Model System for Author):

While very promising and potentially exciting, the novelty and impact of this work is questionable at the moment due to the lack of rigor and technical quality of certain key experiments.

More specifically, the study lacks key control experiments to [1] evaluate the impact of cis-aconitate (CA) treatment on the intracellular levels of this metabolites and its downstream metabolites in the TCA cycle and [2] establish the antiviral and anti-inflammatory effects of CA are independent of other TCA metabolites with similar functions, particularly itaconate and succinate. Additionally, certain findings are built on a limited set of data, several of which are from experiments that lack rigor and are of poor technical quality due to high variation and/or limited number of replicates, lack of statistical significance, or insufficient evidence.

Referee #2 (Remarks for Author):

In this study, the authors identify cis-aconitate (CA) as a TCA cycle metabolite with both antiviral and anti-inflammatory properties during influenza infection. They employ a range of in vitro, ex vivo, and in vivo models, leading to the following key findings:

- 1.CA treatment is effective against multiple influenza A virus strains as well as an influenza B virus.
- 2.The antiviral and anti-inflammatory functions of CA are independent of itaconate.
- 3.CA's antiviral activity stems from its interference with viral polymerase.
- 4.CA mediates its anti-inflammatory effects by modulating key inflammatory signaling pathways and reducing the secretion of inflammatory mediators.

5. CA treatment reduces viral replication and inflammation *in vivo* and is effective even when administered at a later stage of infection, enhancing its clinical relevance.

While the study is potentially very exciting and has several strengths, including the diverse experimental models and analytical approaches employed, there are several concerns that limit its impact and relevance.

The most pressing of these concerns is that the independent function of CA from other TCA cycle metabolites is not sufficiently established. More detailed comments are listed below:

Major Concerns:

1. Establishing the Effect of CA Treatment on the TCA Cycle and Its Regulation During Influenza Infection.

The authors demonstrate that CA treatment does not impact cell viability, proliferation, or ROS production (Fig. EV1). However, its effect on the TCA cycle and its regulation during influenza infection remain unexamined. To address this, they should analyze changes in TCA cycle metabolites and key enzymes in both control and infected cells, with and without CA treatment.

This is essential to determine whether CA administration alters cellular metabolic programs without compromising viability and proliferation, which could have broader implications for cell function and antiviral responses. Additionally, such an analysis would provide a clearer context for CA's role during influenza infection in airway epithelial cells.

Finally, since the downstream TCA cycle metabolites, itaconate and succinate, exhibit similar antiviral and anti-inflammatory effects, this assessment would offer valuable insights into their regulation in this setting.

2. Evaluating the Antiviral Activity of CA.

a. There are inconsistencies in the methods used to assess CA's antiviral activity across the different experimental models used in the study, which hinder cross-comparison. The authors should provide viral titers as measured by plaque assays across all experimental model, using a consistent unit of measurement [preferably (pfu/ml) instead of (%untreated) shown in (Fig.2g), for more transparency]. As an example of such inconsistencies, in (Fig.1) when examining CA's antiviral activity in the bronchial epithelial cell lines and primary cells:

- Antiviral activity is evaluated by viral titers in BEAS-2B cells and NA activity in the primary cells using vastly different replicate numbers. [These issues recur in (Fig.2b and g) in the 3D culture and the OLC models].
- In (Fig.1i and j), CA is given at a 5.7mM concentration, which was not tested for its effect on cell viability.

b. The authors state that "CA impairs viral polymerase activity, suppressing viral mRNA expression and protein synthesis to inhibit replication across a range of influenza subtypes". This claim lacks rigorous support.

- In (Fig.5g) the experiment in which the minigenome system is used to examine the effect of treatment on the viral polymerase is poorly described in terms of rationale, experimental approach, and interpretation.
- Western blot analyses of viral proteins shown in (Figs.5d and e) should be repeated with equal or at least comparable numbers of replicates per condition, with a minimum of 3 replicates to ensure statistical validity.
- To trace CA's effect on viral replication more accurately, viral mRNA expression and protein levels should be measured for the same viral protein. If possible, gene expression analysis of at least one of the proteins examined by microscopy and western blot should be included.

3. Establishing the Independence of CA's Antiviral and Anti-Inflammatory Functions from Downstream TCA Cycle Metabolites.

This aspect should be firmly established early in the study to reinforce the novelty of the work. While this is particularly important for itaconate, a comparison with succinate should also be addressed either experimentally or in the discussion, considering the authors' prior publication.

a. *In vitro* evidence for CA's independence from itaconate is insufficient due to concerns regarding CAD siRNA inhibition (Fig. 7b).

- The regulation of TCA cycle enzymes upstream and downstream of CA has not been examined (as suggested earlier). Therefore it is not possible to rule out whether changes in the enzymatic activity downstream of CA could affect its intracellular levels, either during infection and/or CA treatment.

- The data in (Fig. 7b) show no statistical significance, and the high variability in the IAV-infected group makes interpretation difficult.

- The figure legend states that (Fig. 7b) data are cumulative from three independent experiments. However, with only three data points, it appears that each experiment had a single replicate. If so, siRNA suppression should be evaluated using more replicates across at least two independent experiments.

b. The *in vivo* experiment (Fig. 7f) was only performed once and should be independently replicated. Moreover, the readouts should include lung viral titers and body weight loss to clarify whether improved survival results from antiviral, anti-inflammatory

effects, or both.

4. Assessing CA's Anti-Inflammatory Role.

a. The NF- κ B reporter mouse model results are striking, but follow-up experiments on inflammatory signaling are limited and lacking in rigor.

- Western blot analysis of lung tissue (from uninfected and infected mice, with or without CA treatment) should be performed to confirm the regulation in key proteins involved in the NF- κ B signaling pathways.
- Similarly, the regulation of the ERK/AKT signaling pathway should be evaluated in vivo from lung tissue, as the current in vitro data (Figs. 6a and 6b) exhibit high variability, casting doubt on conclusions.

b. On page 7, line 172, the authors state that CA suppresses inflammatory mediator secretion, but only expression levels have been measured. Secretion should be quantified via ELISA from serum and, if possible, BALF.

c. Flow cytometry data focus only on NK, NKT cell, and neutrophil populations, without justification for excluding macrophages and other relevant cell types. A broader immune cell characterization is needed. Additionally, the antibodies used to identify each cell population should be indicated in both the figure and result section, and the scatter plots showcasing the gating strategy should be included in the EV figure.

d. Histology in (Fig. 4d) is limited to a single magnification. Lower-magnification images should be included to facilitate an unbiased examination of lung damage.

Other General Concerns and Minor Comments:

- The reduction in viral titers following CA treatment in (Figs. 1 e-h) are rather modest. The authors list cis-aconitate's effectiveness against IBVs as one of its main advantages over succinate treatment. This point would be strengthened by examining the effect of CA treatment on IBV titers in a different model (e.g. primary cells), or by modulating the dose of the virus and/or treatment.
- (Figs 2e-f) do not allow for a proper evaluation of the tissue structure and integrity. If possible, the authors should include either H/E staining or confocal staining that include more cell-specific markers, using a similar approach to that used in (Fig. 2a).
- The data shown in (Fig. 8) is presented in a somewhat misleading way, as it builds anticipation for clinical data involving CA treatment. It is more suited as a panel within (Fig 3).
- Similarly, the authors might want to reconsider the arrangement of their data across figures to improve the flow and consolidate their findings.

Referee #3 (Remarks for Author):

The study by Cezard and colleagues highlights the TCA cycle intermediate cis-aconitate as a promising influenza treatment, which is not only effective at early, but also at later stages of infection. The application of cis-aconitate impairs the generation of viral RNA and protein expression together with a reduction of the associated inflammatory signaling. The antiviral activity of cis-aconitate appears not to be associated with the ACOD1/itaconate axis. Regarding cellular metabolism including TCA cycle during influenza virus infection could the authors discuss potential mechanisms besides itaconate as contributors to the observed effects? I would also recommend adding a few aspects on differences and similarities between the antiviral activity of itaconate and cis-aconitate in the discussion section.

Some specific comments. Line 120 states the percent reduction in virus titer for the infection of the organotypic lung culture. It would also be helpful to add for example in the manuscript text for figure 2e-h a percent reduction or fold reduction to get some information on the range of reduction in viral titers for different influenza virus types. CXCL10 levels were not reduced by cis-aconitate (Fig. 4b). This represents an important pro-inflammatory chemokine during influenza virus infection. Would that add some new aspects on the efficacy or potential mechanism of action of cis-aconitate? Could that be due to the application of cis-aconitate at 2 dpi as compared to earlier application time points? The dataset on the minigenome assay indicates the interference with the polymerase activity as an important aspect of the manuscript. I would suggest to add a few points on the background and informative value of this minigenome assay in the results section. As a minor point, please check, line 100, Fig 3a-c as reference.

- **POINT-BY-POINT REPLY TO :**

- **Reviewer 1 : page 2**
- **Reviewer 2 : page 11**
- **Reviewer 1 : page 23**

Referee #1

This is a nice paper but some experimental controls are missing. Further clarifications are also required (see my review) that will likely need so metabolic analysis. That said, I am overall supportive at this stage.

We thank the reviewer #1 very much for her/his positive opinion and constructive input.

Referee #1 (Remarks for Author):

This study investigates the antiviral and anti-inflammatory properties of cis-aconitate (cis-aco) as a potential therapeutic for influenza. The authors propose cis-aco as a promising candidate for influenza treatment, particularly in cases where traditional antivirals (like oseltamivir) are ineffective. Overall this is a very interesting study, however several aspects of the study require further clarification and additional controls are required to strengthen the manuscript.

Major comments:

1. Are levels of itaconate or cis-aco, or other metabolites screened altered in lung of infected mice?

We thank the reviewer for raising this point. To address it, we performed an untargeted metabolomic analysis of murine lungs collected at 2, 4, and 8 days post-influenza infection. The global pulmonary metabolome is presented in Figure S1a.

As shown in Figure S1b below, the levels of both itaconate and cis-aconitate (expressed as normalized peak intensities) did not change significantly over the course of infection compared with mock-challenged animals. This information has been added in the revised manuscript (page 12, line 298 and supplementary Fig. S1).

Figure Supplementary S1. Cezard *et al.*

Figure S1. Pulmonary metabolome alterations in influenza-infected mice.

(a–e) Seven-week-old female C57Bl/6 mice were intranasally infected with 200 pfu of influenza A/Scotland/20/74 (H3N2) virus (IAV). Lungs were collected at 2, 4, and 8 days post-infection (p.i.) for metabolomic profiling by mass spectrometry. Data include results from 5 individual mice from 1 independent experiment. Principal component analysis (PCA) was used to visualize the global spectral distribution of the pulmonary metabolome (panel a). Targeted quantification of itaconate and cis-aconitate is shown in panel b. Statistical analyses were performed using multiple *t*-tests, with a significance threshold set at $-\log(p\text{-value}) \geq 1.5$ (corresponding to $p \leq 0.005$) and a \log_2 fold change ≥ 3 (upregulated) or ≤ -3 (downregulated)

2. How about expression of appropriate metabolic enzymes? - Is aconitase upregulated during infection?

As shown below in Figure S2, we analyzed the pulmonary expression of key enzymes of the TCA cycle by qRT-PCR in mice either infected or not with IAV, and treated intranasally or not with cis-aco (30 mg/kg). Lungs were collected 4 days post-infection for analysis. Our data show that the expression of cis-aconitate decarboxylase (CAD) is markedly upregulated following influenza infection (125-fold increase compared to mock, $p = 0.0008$), consistent with previously published observations [Preusse et al. PMID : 24341411]. In contrast, expression of aconitase 1 (ACO1), isocitrate dehydrogenase 2 (IDH2), and succinate dehydrogenase (SDH) is significantly reduced in infected lungs (fold changes of -4 , -2.6 , and -2.2 , respectively; $p = 0.0015$, 0.004 , and 0.0365). This information is now included in the revised manuscript at page 12, lines 299, 307, and in Supplementary Fig. S2).

Figure Supplementary S2. Cezard *et al.*
Figure S2. Influenza-induced alterations in TCA cycle enzyme expression and their modulation by cis-aconitate treatment.

Seven-week-old female C57Bl/6 mice were intranasally infected with 200 pfu of influenza A/Scotland/20/74 (H3N2) virus (IAV) and treated intranasally or not with 30 mg/kg cis-aconitate (CA) at day 2 post-infection (p.i.). Lungs were collected at day 4 p.i. and the expression of key TCA cycle enzymes was quantified by RT-qPCR. Statistical analyses were performed using the Kruskal–Wallis test with Dunn’s multiple comparison test. Data are presented as mean \pm SEM and each dot represents one individual mouse (pooled from 3 independent experiments).

3. Relating to the above, is aconitase or cis-aco increased in human IAV treated ECs?

To address this question, we investigated the impact of influenza infection and cis-aconitate treatment on the TCA cycle in human bronchial epithelial cells. Cells were either mock-infected or infected with IAV, and treated 4 h post-infection with cis-aconitate (3.4 mM) or left untreated. At 8 and 24 h post-infection, expression of TCA cycle enzymes was quantified by qRT-PCR (Figure S3a) while intracellular concentrations of citric, α -ketoglutaric, succinic, fumaric, and malic acids were measured by mass spectrometry (Figure S3b). In addition, mitochondrial respiration was analyzed after 20 h of cis-aconitate treatment (3.4 mM; Figure S3c).

Consistent with the *in vivo* results, cis-aconitate decarboxylase (CAD) expression was markedly upregulated 24 h post-influenza infection (27-fold increase, $p = 0.0109$; Figure S3a). Conversely, IDH2 expression was reduced (-2.7 -fold), while OGDH and MDH2 expression were both increased at 24 h (10-fold, $p = 0.0049$, and modestly elevated, respectively; Figure S3a). The intracellular concentration of cis-aconitate remained below 43 nM, irrespective of infection status (data not shown). Similarly, the abundance of other TCA intermediates (citric, α -ketoglutaric, succinic, fumaric, and malic acids) was not significantly modulated by infection (Figure S3b). All of these data are now included in the revised manuscript (page 12, lines 300-309; Supplementary Fig. S3).

Figure Supplementary S3. Cezard *et al.*

Figure S3. Influenza-induced alterations of TCA cycle enzyme expression and the effect of cis-aconitate (CA) treatment in human bronchial epithelial cells.

BEAS-2B cells were infected with influenza A/Scotland/20/74 (H3N2) at a multiplicity of infection (MOI) of 1 for 4 h, then washed and treated with 3.4 mM cis-aconitate or left untreated. Expression of TCA cycle enzymes was quantified by RT-qPCR at 8 and 24 h post-infection (a), and intracellular levels of key TCA cycle metabolites were measured by mass spectrometry at 24 h post-infection (b). Mitochondrial respiration was assessed using the MitoStress test on an XFe24 analyzer (c–g), including real-time monitoring of oxygen consumption rate (OCR) in cells treated or not with cis-aconitate for 20 h (d), basal respiration prior to oligomycin injection (e), ATP-linked OCR calculated as the difference between basal OCR and OCR after oligomycin (f), and maximal respiration following FCCP injection with spare respiratory capacity calculated as the difference between maximal and basal OCR (g). Results are presented as mean \pm SEM from 4 independent experiments (a–b) or individual well values (d–g). Statistical analyses were performed using the Kruskal–Wallis test with Dunn’s multiple comparison test (a–b) or the Wilcoxon matched-pairs signed rank test (d–g).

4. Were metabolic responses between BEAS2B and primary cells different?

We thank the reviewer for this important question. To address it, we analyzed transcriptomic data from Bertrams *et al.* [PMID: 36446841], comparing the response to influenza infection across three models: primary airway epithelial cells, primary alveolar macrophages, and human lung tissue explants, as shown in Figure S4. Overall, only a few mitochondrial genes were significantly up- or downregulated (highlighted in red and blue, respectively). Notably, cis-aconitate decarboxylase (CAD) was consistently induced across all three primary models, in agreement with the response observed in BEAS-2B cells (*cf.* Figure S3), indicating that this aspect of the metabolic response is conserved between the immortalized cell line and primary human cells. These data are now included in the revised manuscript (page 12, line 302; Supplementary Fig. S4).

Figure Supplementary S4. Cezard *et al.*

Figure S4. Comparison of metabolic enzyme expression changes induced by influenza infection across different primary lung cell types.

Expression of metabolic enzymes in alveolar macrophages, airway epithelial cells, and human lung tissue explants was analyzed based on data from Bertrams *et al.* [77]. Upregulated genes are shown in red, and downregulated genes in blue.

5. *What is the half-life of aconitase in vivo?*

To our knowledge, there are currently no data on the half-life of aconitase *in vivo*. However, Christiano *et al.* [PMID: 25466257] reported a half-life of aconitase in yeast, which was determined to be 8.7 hours. While this may not directly translate to mammalian systems, it provides a reference point for enzyme stability.

6. *Please make it clearer in text the source of primary ECs (although this is in methods, it should be mentioned in text).*

As suggested, we have clarified the source of primary epithelial cells in the main text (line 105): “Primary bronchial epithelial cells (PBECS) were isolated from normal bronchial tissues of lung cancer patients undergoing lobectomy at the University hospital (CHRU) of Tours. Cells were scraped from the luminal surface of cancer-free tissue and cultured under conditions promoting mucociliary differentiation.”

7. *In murine models, please make it clearer how cis-aco was given and frequency/dose in schematics.*

Accordingly, we have added the abbreviation “IN” in all schemes to indicate intranasal administration. The dose of cis-aconitate was kept consistent throughout all experiments and is now explicitly indicated in all figure legends and main text.

8. *Please provide CA-only controls for all mouse studies.*

We fully agree with the reviewer on the importance of including cis-aconitate–only controls in the *in vivo* experiments. To specifically address this point, we provide below Figure R1 (for the reviewer’s consideration only), which displays the complete protein array corresponding to the original Figure 4b, now including all control conditions (mock and cis-aco alone).

In addition, panels c, d, and f of the revised Figure 4 have been updated to incorporate mock conditions with and without cis-aconitate, thereby ensuring a more comprehensive presentation of the results

Regarding NF-κB-Luciferase mice, due to the limited availability of this model, only mock data (without cis-aco) have been added (see revised Figure 4f).

For the survival experiment, treatment with cis-aco alone did not affect body weight and caused no mortality in wild-type mice (as described in Figure EV4).

Concerning CAD^{-/-} mice, it was not possible to perform a cis-aconitate-only experiment due to the very limited number of animals available during the review period, as these mice were allocated to address other critical questions raised by the reviewers.

Figure R1. Cezard *et al.*

Figure R1. Levels of mediators measured in BAL fluids from IAV-infected mice. Seven-week-old female C57Bl/6 mice were intranasally challenged with PBS or infected with 200 pfu of influenza A/Scotland/20/74 (H3N2) virus (IAV) and subsequently treated, or not, at 2 days post-infection (p.i.) with 30 mg/kg cis-aconitate (CA) administered intranasally. Mice were euthanized at 4 days p.i. and the levels of 50 mediators were quantified in BAL fluids using a protein array.

9. The human data in figure 8 seems a bit out of the blue and very hard to directly relate this timing to the murine model, I am not sure if it is useful in this context.

Thank you for your comment. We agree that a direct comparison between human and murine data in Figure 8 is not straightforward, as the exact timing of infection in patients is uncertain. Our study of the time from symptom onset to hospital presentation (Fig. 8a) indicates that influenza patients typically arrive around 4 days after symptoms begin. To mirror this scenario, we initiated treatment in the murine model at day 4 post-infection. While a perfect temporal match is not possible, we believe this approach provides useful context for interpreting the results.

Referee #2

3. *Establishing the Effect of CA Treatment on the TCA Cycle and Its Regulation During Influenza Infection.*

The authors demonstrate that CA treatment does not impact cell viability, proliferation, or ROS production (Fig. EV1). However, its effect on the TCA cycle and its regulation during influenza infection remain unexamined. To address this, they should analyze changes in TCA cycle metabolites and key enzymes in both control and infected cells, with and without CA treatment.

This is essential to determine whether CA administration alters cellular metabolic programs without compromising viability and proliferation, which could have broader implications for cell function and antiviral responses. Additionally, such an analysis would provide a clearer context for CA's role during influenza infection in airway epithelial cells.

Finally, since the downstream TCA cycle metabolites, itaconate and succinate, exhibit similar antiviral and anti-inflammatory effects, this assessment would offer valuable insights into their regulation in this setting.

Accordingly, we assessed the impact of cis-aconitate (cis-aco) treatment on the TCA cycle in human bronchial epithelial cells, both with and without influenza infection (see Fig. S3, page 5-6 of this point-by-point reply). At 24 h post-infection, cis-aconitate decarboxylase (CAD) expression was markedly upregulated (27-fold increase, $p=0.0109$; Fig. S3a) whereas IDH2 expression was decreased by 2.7-fold. In human IAV-infected epithelial cells, mass spectrometry analyses showed that cis-aco levels remained below 43 nM regardless of infection (data not shown). Moreover, concentrations of key metabolites including citric acid, ketoglutaric acid, succinic acid, fumaric acid, and malic acid were not significantly modulated by infection (Fig. S3b).

Consistent with these observations, influenza infection in murine lungs also led to a marked upregulation of CAD and a downregulation of IDH2 (Fig. S2).

Mass spectrometry showed that the serum metabolome of infected mice differed from controls, with itaconate -but not succinate- modestly increased under IAV conditions (3.12-fold, $p=0.0452$; see Fig. R2a–b below). These data are provided to address the reviewer's comment and are not included in the manuscript.

These results, in agreement with the literature [Ohno et al. PMID: 32616893 ; Sohail et al. PMID: 35025971; Smallwood et al. PMID: 28538182] suggest that influenza infection shunts the TCA cycle through CAD upregulation and enhanced itaconate production, thereby disrupting normal TCA cycle homeostasis.

Interestingly, cis-aco treatment partially reversed these changes. In both epithelial cells and murine lungs, cis-aco tended to reduce CAD expression. In addition, cis-aco strongly upregulated citrate synthase, ACO2, IDH2, and SDHA expression in human bronchial epithelial cells, regardless of infection status (Fig. S3a), and increased IDH2 expression in murine lungs (Fig. S2). While cis-aco treatment did not increase TCA metabolite concentrations in epithelial cells (Fig. S3b), Seahorse assays revealed that it enhanced mitochondrial respiration and ATP production (Fig. S3c–g). Similarly, cis-aco did not increase serum itaconate levels in either naïve or influenza-infected mice (Fig. R2b).

Taken together, these results indicate that cis-aco tends to restore TCA cycle homeostasis disrupted by influenza infection by normalizing key enzyme expression and enhancing respiration and ATP production, independently of increased metabolite accumulation.

Figure R2. Effects of influenza infection and cis-aco treatment on TCA cycle metabolites in mice.

Seven-week-old female C57Bl/6 mice were intranasally infected with 200 PFU of influenza A/Scotland/20/74 (H3N2) virus (IAV) and treated intranasally or not with 30 mg/kg of cis-aconitate (CA) at 2 days post-infection (p.i.). Mice were euthanized at day 4 p.i., and TCA cycle metabolites were quantified by mass spectrometry in bronchoalveolar lavage fluid (BAL), lung tissue, and serum. Principal component analysis (PCA) was generated from the quantified serum metabolome (a), and itaconate and succinate levels were extracted from these data (b). Statistical analyses were performed using the Kruskal–Wallis test with Dunn's multiple comparison test. Data are presented as mean \pm SEM, with each white dot representing one individual mouse (n=5 from a single experiment).

4. Evaluating the Antiviral Activity of CA.

a. There are inconsistencies in the methods used to assess CA's antiviral activity across the different experimental models used in the study, which hinder cross-comparison. The authors

should provide viral titers as measured by plaque assays across all experimental model, using a consistent unit of measurement [preferably (pfu/ml) instead of (%untreated) shown in (Fig.2g), for more transparency]. As an example of such inconsistencies, in (Fig.1) when examining CA's antiviral activity in the bronchial epithelial cell lines and primary cells:

- Antiviral activity is evaluated by viral titers in BEAS-2B cells and NA activity in the primary cells using vastly different replicate numbers. [These issues recur in (Fig.2b and g) in the 3D culture and the OLC models].

Due to inter-individual variability in patient-derived primary cells and lung tissues, we initially presented the data normalized to the untreated infected condition to better highlight the effect of cis-aco. However, in response to the reviewer's request, we now provide the corresponding viral titration data expressed in PFU/mL or FFU/mL in the revised Fig. 1i and Fig. 2b–g. We believe these additional data address the reviewer's concern and enable a more transparent cross-comparison across models.

i. In (Fig.1i and j), CA is given at a 5.7 mM concentration, which was not tested for its effect on cell viability.

The effect of cis-aconitate (CA) at 5.7 mM on cell viability was documented in the expanded file Fig. EV1f.

To further support this, we added complementary measurements in primary bronchial epithelial cells (PBEC) at this concentration using propidium iodide (PI) staining (see Fig. EV1g). In addition, we provide an assessment of the safety of this dose in PBEC cultured at the air–liquid interface (ALI) using a MTS assay (Fig. EV1h).

b. The authors state that "CA impairs viral polymerase activity, suppressing viral mRNA expression and protein synthesis to inhibit replication across a range of influenza subtypes". This claim lacks rigorous support.

We thank the reviewer for this important comment. Our mini-genome experiments suggest that cis-aconitate can impair viral polymerase activity, leading to reduced viral mRNA expression and protein synthesis. We agree, however, that these assays were conducted exclusively with the influenza A/Scotland/20/74 (H3N2) strain. The reviewer is therefore correct that our original statement was too broad. We have revised the text accordingly to better reflect the extent of our data.

The revised sentence now reads: « CA appears to impair viral polymerase activity, resulting in decreased viral mRNA expression and protein synthesis, as observed for the influenza A/Scotland/20/74 (H3N2) strain ». See page 2, line 36-37.

i. In (Fig.5g) the experiment in which the minigenome system is used to examine the effect of treatment on the viral polymerase is poorly described in terms of rationale, experimental approach, and interpretation.

The influenza virus minigenome system is a well-established molecular tool that enables the study of viral polymerase activity (PB1, PB2, PA) and nucleoprotein (NP) function without the need for infectious virus. In this system, cultured cells are transfected with plasmids encoding PB1, PB2, PA, and NP, together with a minigenome plasmid carrying a reporter gene (firefly luciferase) flanked by viral untranslated regions (UTRs). The reporter RNA is transcribed under the control of an RNA polymerase I promoter, generating viral-like RNA. The reconstituted polymerase complex recognizes this RNA and initiates transcription and replication, thereby mimicking the intracellular processes of the authentic virus. Reporter gene expression thus serves as a direct readout of polymerase activity.

A concise description of this system has been added in the Results section of the manuscript (see page 8, lines 194-200) to clarify the rationale, experimental setup, and interpretation for readers who may not be familiar with this approach.

ii. Western blot analyses of viral proteins shown in (Figs.5d and e) should be repeated with equal or at least comparable numbers of replicates per condition, with a minimum of 3 replicates to ensure statistical validity.

To improve clarity, the representation of the untreated condition in Fig. 5e has been updated to reflect the variability across experiments, using the average of all independent experiments. Each independent data point is indicated by a white dot. Importantly, all conditions now include at least 3 independent replicates to ensure statistical validity. In addition, for the reviewer's consideration, we provide below the raw Western blot images used for quantification (Fig. R3).

Figure R3. Raw Western blot images used for quantification in Fig. 5e. Results from each experiment (Exp.) are presented in separate columns and each viral protein is shown in a separate row

iii. To trace CA's effect on viral replication more accurately, viral mRNA expression and protein levels should be measured for the same viral protein. If possible, gene expression analysis of at least one of the proteins examined by microscopy and western blot should be included.

To more accurately monitor the effect of CA on viral replication, we ensured that the viral proteins M1 and NP were analyzed using all the techniques presented in Figure 5, namely immunofluorescence, Western blot and qRT-PCR.

3. Establishing the Independence of CA's Antiviral and Anti-Inflammatory Functions from Downstream TCA Cycle Metabolites.

This aspect should be firmly established early in the study to reinforce the novelty of the work. While this is particularly important for itaconate, a comparison with succinate should also be addressed either experimentally or in the discussion, considering the authors' prior publication.

We thank the reviewer for this comment. We fully agree that establishing the independence of CA's antiviral and anti-inflammatory functions early in the study is important to highlight the novelty of our work. Succinate and CA indeed differ in their functional profiles: succinate primarily exhibits antiviral activity whereas CA demonstrates both antiviral and anti-inflammatory effects. This distinction is critical for understanding the therapeutic potential of CA. Consistently, succinate is not effective in vivo at later stages of infection (e.g., day 2 post-infection, data not shown), whereas CA enhances survival at these time points. Importantly, the doses of CA used in our study are lower than those of succinate: in vitro, succinate was used at 25 mM (4 mg/mL) compared with 3.4 mM (0.6 mg/mL) for CA; in vivo, succinate was used at 840 mM (200 mg/kg) versus 86 mM (30 mg/kg) for CA. These differences further support the peculiar efficacy of CA.

From an antiviral perspective, succinate inhibits the nuclear-to-cytoplasmic translocation of the viral nucleoprotein, whereas CA (cis-aconitate) acts by inhibiting viral polymerase activity. Consequently, even if cis-aconitate were to increase succinate levels via the TCA cycle, any potential antiviral effect of succinate would occur after cis-aconitate has already suppressed viral replication.

Moreover, the concentrations of cis-aconitate used in our study are far below those required to generate a biologically relevant increase in succinate. Consistent with this, cis-aconitate treatment did not induce significant changes in the serum metabolome (Fig. S5), with neither succinate nor itaconate levels being upregulated (Fig. S5b). In conclusion, CA possesses distinct anti-inflammatory properties that differentiate it from succinate, enabling it to be effective at lower doses and within a therapeutically relevant window. This information complements the discussion in the original manuscript (see page 15, line 373-381).

a. In vitro evidence for CA's independence from itaconate is insufficient due to concerns regarding CAD siRNA inhibition (Fig. 7b).

i. The regulation of TCA cycle enzymes upstream and downstream of CA has not been examined (as suggested earlier). Therefore it is not possible to rule out whether changes in the enzymatic activity downstream of CA could affect its intracellular levels, either during infection and/or CA treatment.

As addressed in our responses to

Questions 1 and 2 by Reviewer

#1, we have examined the regulation of TCA cycle enzymes both upstream and downstream of CA. Please refer to Figures S1 and S2, which present these studies *in vitro* and *in vivo*, respectively.

ii. The data in (Fig. 7b) show no statistical significance, and the high variability in the IAV-infected group makes interpretation difficult.

iii. The figure legend states that (Fig. 7b) data are cumulative from three independent experiments. However, with only three data points, it appears that each experiment had a single replicate. If so, siRNA suppression should be evaluated using more replicates across at least two independent experiments

Figure 7b has been updated to provide meaningful statistical analysis validating CAD siRNA-mediated suppression compared to WT. In the basal state, CAD expression in this cellular model is very low (mean Cp >35), which makes it difficult to detect significant suppression by siRNA. However, during IAV infection, CAD expression is strongly upregulated (mean Cp <30), allowing clear validation of siRNA-mediated knockdown (mean Cp >32). These data confirm effective suppression of CAD under conditions where its expression is physiologically relevant

b. The in vivo experiment (Fig. 7f) was only performed once and should be independently replicated. Moreover, the readouts should include lung viral titers and body weight loss to clarify whether improved survival results from antiviral, anti-inflammatory effects, or both.

We thank the reviewer for this comment. Regarding the requested repetition of the *in vivo* IRG1 knockout experiment (Fig. 7f), the experiment was challenging to perform due to delays in generating sufficient IRG1 KO mice at our collaborators' facility (CIIL, Lille, France). The experiment was eventually carried out this September, but a technical issue with the viral inoculum resulted in a dose too low to induce severe disease, preventing meaningful conclusions.

Importantly, our original submission already included IRG1 KO data based on 8–10 mice per group (control and infected \pm cis-aconitate). These results are statistically robust and fully support the overall conclusions of the study. In combination with the large body of new experimental data addressing the other reviewer concerns, we believe that the revised manuscript provides a comprehensive and rigorous basis for evaluation.

4. Assessing CA's Anti-Inflammatory Role.

a. The NF- κ B reporter mouse model results are striking, but follow-up experiments on inflammatory signaling are limited and lacking in rigor.

- Western blot analysis of lung tissue (from uninfected and infected mice, with or without CA treatment) should be performed to confirm the regulation in key proteins involved in the NF- κ B signaling pathways.

- Similarly, the regulation of the ERK/AKT signaling pathway should be evaluated in vivo from lung tissue, as the current in vitro data (Figs. 6a and 6b) exhibit high variability, casting doubt on conclusions

We thank the reviewer for this important suggestion. In response, we performed Western blot analysis of phosphorylated ERK1/2 in lung tissue from uninfected and infected mice, with or without CA treatment (see Fig. R4 below, for the reviewer's consideration). Similar to the human bronchial epithelial cell model, influenza infection induced ERK1/2 phosphorylation. Cis-aconitate treatment reduced this phosphorylation, as well as basal phosphorylation levels (2.75- and 1.57-fold decreases, $p = 0.0485$ and 0.0057 , respectively, for MOCK vs MOCK+CA and IAV vs IAV+CA).

Figure R4. Cis-aconitate reduces ERK1/2 phosphorylation in vivo.

Seven-week-old female C57Bl/6 mice were intranasally infected with 200 pfu of influenza A/Scotland/20/74 (H3N2) virus (IAV) and treated, or not, 2 days post-infection (p.i.) with 30 mg/kg cis-aconitate (CA) intranasally. Mice were euthanized at day 4 p.i. (a) Representative Western blots showing phosphorylated ERK1/2 (P-ERK1/2) with β -actin as a loading control. (b) Quantification of protein signals normalized for each experiment. Statistical analyses were performed using a paired *t*-test. Data are presented as mean \pm SEM and include results from 3 independent experiments, with each white dot representing an individual mouse.

b. On page 7, line 172, the authors state that CA suppresses inflammatory mediator secretion, but only expression levels have been measured. Secretion should be quantified via ELISA from serum and, if possible, BALF.

We acknowledge the reviewer's concern. However, Figure 4b in the original manuscript, based on a protein array, already demonstrated that CA reduces the secretion of inflammatory mediators in mouse BAL following influenza infection. Nevertheless, to further validate these findings as requested, we performed classical ELISAs for myeloperoxidase (MPO), interleukin-6 (IL-6), and the chemokine KC using the same in vivo protocol (Fig. R5, for the reviewer's consideration only). These ELISA results fully confirm and are consistent with the observations from the protein array.

Figure R5. Cis-aconitine reduces secretion of pro-inflammatory mediators in vivo. Seven-week-old female C57Bl/6 mice were intranasally infected with 200 pfu of influenza A/Scotland/20/74 (H3N2) virus (IAV) and treated, or not, 2 days post-infection (p.i.) with 30 mg/kg cis-aconitine (CA) intranasally. Mice were euthanized at day 4 p.i. Secretion of myeloperoxidase (MPO), interleukin-6 (IL-6), and the chemokine KC (KC) was measured in BAL fluid by ELISA. Statistical analyses were performed using the Wilcoxon matched-pairs signed-rank test. Data are presented as mean \pm SEM and include results from three independent experiments, with each white dot representing an individual mouse

c. Flow cytometry data focus only on NK, NKT cell, and neutrophil populations, without justification for excluding macrophages and other relevant cell types. A broader immune cell characterization is needed. Additionally, the antibodies used to identify each cell population should be indicated in both the figure and result section, and the scatter plots showcasing the gating strategy should be included in the EV figure.

As suggested by the reviewer, the number and activation status of alveolar macrophages have been added to Fig. 4c. The antibodies used to identify each cell population are now indicated in both the figure and the Results section.

Additionally, the gating strategy for the flow cytometry analysis is provided in the new supplementary Fig. S6 (see also below)

Supplementary Figure S6. Cezard *et al.*

Figure S6. Flow cytometry gating strategy. Representative surface gating used to define immune cell subsets shown in Fig. 5 and Supplementary Figure EV4.

d. Histology in (Fig. 4d) is limited to a single magnification. Lower-magnification images should be included to facilitate an unbiased examination of lung damage.

As suggested, we have included lower-magnification images (see Fig. R6 below) corresponding to the histology shown in the original Figure 4d. However, we believe that these lower-magnification images provide only a general overview and do not show the pathological features as clearly as the higher-magnification images in the initial figure.

Figure R6. Cis-aconitate reduces lung inflammation and tissue damage in IAV-infected mice.

Seven-week-old female C57Bl/6 mice were intranasally infected with 200 pfu of influenza A/Scotland/20/74 (H3N2) virus (IAV) and treated, or not, 2 days post-infection (p.i.) with 30 mg/kg cis-aconitate (CA) intranasally. Mice were euthanized at 8 days p.i. Lung sections were stained with H&E, and tissue lesions were assessed and scored (scale bar: 500 µm). Data represent results from two independent experiments.

Other General Concerns and Minor Comments:

- The reduction in viral titers following CA treatment in (Figs. 1 e-h) are rather modest. The authors list cis-aconitate's effectiveness against IBVs as one of its main advantages over succinate treatment. This point would be strengthened by examining the effect of CA treatment on IBV titers in a different model (e.g. primary cells), or by modulating the dose of the virus and/or treatment.

We agree with the reviewer that examining the effect of CA treatment on IBV titers in a different model would be valuable. To address this, we tested primary airway epithelial cells grown in 2D culture (see Figure R7 below). In this more physiologically relevant system, cis-aconitate was found to decrease viral production following infection with influenza A (H3N2) as well as influenza B viruses (Yamagata and Victoria lineages). These results confirm the broad anti-influenza activity of cis-aconitate. Importantly, the data from influenza B Victoria in primary epithelial cells have been incorporated into the Figure 1i of the revised manuscript.

a.

Figure R7. Cis-aconitate exhibits anti-influenza A and B activity in human primary bronchial epithelial cells (PBEc) in 2D culture.

PBEc were infected with influenza A/Scotland/20/74 (H3N2), influenza B Yamagata (B/Paris/234/2013), or influenza B Victoria (B/Bretagne) at an MOI of 1 for 4 h, followed by treatment with or without 3.4 mM cis-aconitate (CA) for 20 h. Viral titers in the supernatant were quantified at 20 h post-infection by TCID₅₀. Data are presented as mean ± SEM and represent cumulative results from 4 independent experiments using cells from 4 different donors. Statistical analyses were performed using the Wilcoxon matched-pairs signed-rank test.

- (Figs 2e-f) do not allow for a proper evaluation of the tissue structure and integrity. If possible, the authors should include either H/E staining or confocal staining that include more cell-specific markers, using a similar approach to that used in (Fig. 2a).

As suggested, we provide in Fig. R8 below a microscopy image of OLC stained with H&E and annotated by an anatomopathologist to allow proper evaluation of tissue structure and integrity.

Figure R8. Microscopic image of OLC stained with H&E.

The data shown in (Fig. 8) is presented in a somewhat misleading way, as it builds anticipation for clinical data involving CA treatment. It is more suited as a panel within (Fig 3).

- Similarly, the authors might want to reconsider the arrangement of their data across figures to improve the flow and consolidate their findings.

We thank the reviewer for the suggestion regarding the reorganization of the figures. In the very initial version of our manuscript, we structured the data to first present evidence supporting the independent effects of cis-aconitate, followed by its impact on various stages of the viral cycle, activity against multiple influenza strains and anti-inflammatory and anti-cell death properties. These findings were then validated in more translational models (i.e. primary cells, OLCs, and in vivo) before concluding with a clinical perspective.

After careful consideration and extensive discussions among all co-authors, we revised the structure to first present the descriptive data, followed by the mechanistic aspects. We believe that this revised organization provides a clearer narrative flow, allowing readers to progressively build their understanding of cis-aconitate's multifaceted effects.

We hope this explanation clarifies our rationale for the current figure arrangement.

Referee #3 (Remarks for Author):

The study by Cezard and colleagues highlights the TCA cycle intermediate cis-aconitate as a promising influenza treatment, which is not only effective at early, but also at later stages of infection. The application of cis-aconitate impairs the generation of viral RNA and protein expression together with a reduction of the associated inflammatory signaling.

The antiviral activity of cis-aconitate appears not to be associated with the CAD/itaconate axis. Regarding cellular metabolism including TCA cycle during influenza virus infection could the authors discuss potential mechanisms besides from itaconate as contributors to the observed effects? I would also recommend adding a few aspects on differences and similarities between the antiviral activity of itaconate and cis-aconitate in the discussion section.

To address this point, we investigated the impact of cis-aconitate on the TCA cycle both in uninfected and influenza-infected human bronchial epithelial cells (HBECs), as well as in infected mice (see supplementary Fig. S2). At 24 hours post-infection, we observed a marked upregulation of cis-aconitate decarboxylase (CAD) in HBECs (27-fold increase, $p = 0.0109$; Fig. S2a), accompanied by downregulation of IDH2 (2.7-fold decrease). These changes were also detected in vivo, with increased CAD and reduced IDH2 expression in the lungs of infected mice (Fig. S1). Serum metabolomics confirmed a distinct metabolic profile in infected animals, consistent with a major metabolic reprogramming during influenza infection (Fig. S5b).

Cis-aconitate treatment appeared to counteract this metabolic imbalance. In both HBECs and mouse lungs, cis-aconitate reduced CAD expression (Figs. S2 and S1). In HBECs, it also upregulated key TCA cycle enzymes (CS, ACO2, IDH2, SDHA), irrespective of infection status (Fig. S2a), and restored IDH2 expression in infected lungs when administered post-infection. Although cis-aconitate did not significantly increase TCA cycle metabolites in HBECs (Fig. S2b), Seahorse analysis revealed enhanced mitochondrial respiration and ATP production (Fig. S3b). Importantly, cis-aconitate did not increase serum itaconate levels in either naïve or infected mice (Fig. S5b), indicating that its antiviral effects are independent of this metabolite.

In summary, these results indicate that cis-aconitate helps restore TCA cycle homeostasis disrupted by influenza infection by modulating TCA cycle enzyme expression and improving mitochondrial bioenergetics.

With regard to comparison with itaconate, both metabolites are linked to the TCA cycle and share some immunomodulatory and antiviral properties. However, our data highlight important differences: whereas itaconate has been reported to inhibit SDH and IDH2, thereby suppressing respiration, reducing ROS, and enhancing AKT phosphorylation [Heinz et al., PMID: 36038039; Sohail et al., PMID: 35025971], cis-aconitate upregulated SDHA and IDH2 expression, increased mitochondrial respiration and ATP production (Fig. S3a-b), and did not reduce ROS (Fig. EV1e) nor activate AKT phosphorylation (Fig. 6a-b).

Overall, these findings indicate that cis-aconitate acts through a distinct, and partly opposite, mechanism to itaconate, highlighting the originality of its antiviral mode of action.

A part of these considerations has now been integrated into the revised Discussion section to address the reviewer's comment (see pages 12-13, lines 296-320).

Some specific comments. Line 120 states the percent reduction in virus titer for the infection of the organotypic lung culture. It would also be helpful to add for example in the manuscript text for figure 2e-h a percent reduction or fold reduction to get some information on the range of reduction in viral titers for different influenza virus types.

Due to inter-individual variability in patient-derived primary cells and organotypic lung cultures, we initially presented the data normalized to the untreated infected condition to clearly illustrate the effect of cis-aconitate. In response to the reviewer's suggestion, we have now included the corresponding viral titration data expressed in PFU/mL or FFU/mL in the revised Fig. 1i and Fig. 2b–g. These additions provide a more clearer representation and allow direct comparison across models.

Overall, our data show that cis-aconitate consistently reduces viral titers by about 1 log across different influenza virus strains (H3N2, H1N1, Influenza B) and various models, including bronchial cell lines, primary cells, organotypic lung cultures and mice.

CXCL10 levels were not reduced by cis-aconitate (Fig. 4b). This represents an important pro-inflammatory chemokine during influenza virus infection. Would that add some new aspects on the efficacy or potential mechanism of action of cis-aconitate? Could that be due to the application of cis-aconitate at 2 dpi as compared to earlier application time points?

We thank the reviewer for this comment. CXCL10 (IP-10) is a pleiotropic molecule with both pro-inflammatory and antiviral functions, acting as a sentinel in host defense and contributing to the development of protective T-cell responses following viral infection. Recent studies have also highlighted a critical role for CXCL10 - as well as CXCL9 - in antiviral immune responses even in the absence of T and B cells [Trifilo et al. PMID: 14694090].

Interestingly, in our protein array, we observed that CXCL10/IP-10 and CXCL9 are only moderately reduced by cis-aconitate treatment, suggesting that signaling pathways involving these related chemokines are largely preserved. This preservation may allow CXCL9 and CXCL10 to contribute to the antiviral activity of cis-aconitate, while this metabolite downregulates ERK- and NF- κ B-dependent inflammatory signaling, as shown in our study both in vitro and in vivo.

We are currently investigating these inhibitory mechanisms in more detail using single-cell RNA sequencing of primary airway epithelial cells, infected or not with influenza virus and treated or not with cis-aconitate. This ongoing work will allow us to address the reviewer's question more precisely. At this stage however, any further conclusions would be speculative, and we prefer not to include them in the manuscript.

The dataset on the minigenome assay indicates the interference with the polymerase activity as an important aspect of the manuscript. I would suggest to add a few points on the background and informative value of this minigenome assay in the results section. As a minor point, please check, line 100, Fig 3a-c as reference.

We thank the reviewer for this comment. The influenza virus minigenome system allows the study of viral polymerase activity (PB1, PB2, PA, and NP) without using infectious virus by reconstituting the viral ribonucleoprotein complex (vRNP) in cultured cells.

Cells are transfected with expression plasmids for the polymerase proteins and a minigenome plasmid encoding a reporter gene (firefly luciferase) flanked by viral UTRs. Reporter expression directly reflects polymerase activity, thus providing a quantitative measure of viral transcription and replication.

A brief description of this system has been added to the Results section (lines 196-202) for clarity. We also corrected the reference error. Thank you very much for your attention.

21st Nov 2025

Dear Dr. Si-Tahar,

Thank you for the submission of your manuscript to EMBO Molecular Medicine and please accept my apologies for the delay in getting back to you, which is due to the fact that both referees needed more time to complete their reviews. I am pleased to inform you that we will be able to accept your manuscript pending the following final amendments:

- 1) Authors: We note name discrepancies in our submission system and in the manuscript. Séverine Heumel in our system and Severine Heumel in the manuscript; Laure Perrin in our system and Laure Perrin-Cocon in the manuscript. Please correct.
- 2) Figures: Please upload main figures and EV figures as individual high-resolution files. Please check "Author Guidelines" for more information:
<https://www.embopress.org/page/journal/17574684/authorguide#figureformat>
<https://www.embopress.org/page/journal/17574684/authorguide#expandedview>
- 3) Author checklist: Please complete the form by selecting appropriate responses in all column D fields.
- 4) In the main manuscript file, please do the following:
 - Please address all comments suggested by our data editors listed below:
 - o Figure legends:
 1. Please define the annotated p values ****/**/*/* as well as provide the exact p-values for the same in the legends of figures 1B, C, D, E, F, G, H, I, J, K; 2B-D; 3A, C; 4A, C, E, G; 5C, E, F, G; 6B, C, E, F; 7B, C, D, E; 8B, EV2 C, D; EV3 B, EV5A, S2, S3A, D, E, F, G as appropriate.
 - Please correct the order and heading of the sections in the manuscript text to: Abstract / Keywords / The Paper Explained / Introduction / Results / Discussion / Methods / Data Availability / Acknowledgements / Disclosure and Competing Interests Statement / References / Main Figure Legends / Tables / Expanded View Figure Legends
 - Please include callouts for the panels in Fig EV2, 3 and 5.
 - In Methods, provide the statement that in addition to WMA Declaration of Helsinki the experiments involving human participants also conformed to the principles set out in the Department of Health and Human Services Belmont Report.
 - Indicate in legends exact n and exact p values, not a range, along with the statistical test used. To keep the figures "clear" some authors found providing an Appendix table Sx with all exact p-values preferable. You are welcome to do this if you want to.
 - Please upload Reagents and Tools Table as a separate file. Structured Methods section includes Reagents and Tools Table followed by a Methods and Protocols section. More information on how to adhere to this format as well as downloadable templates (.docx) for the Reagents and Tools Table can be found in our author guidelines:
<https://www.embopress.org/page/journal/17574684/authorguide#structuredmethods>
An example of a paper with Structured Methods can be found here:
<https://www.embopress.org/doi/full/10.1038/s44320-024-00037-6#sec-4>
 - Rename "Declaration of competing interest" to "Disclosure Statement & Competing Interests" and place it after the "Acknowledgements". We updated our journal's competing interests policy in January 2022 and request authors to consider both actual and perceived competing interests. Please review the policy <https://www.embopress.org/competing-interests> and update your competing interests if necessary.
 - Author contributions: Please remove it from the manuscript and specify author contributions in our submission system. CRediT has replaced the traditional author contributions section because it offers a systematic machine-readable author contributions format that allows for more effective research assessment. You are encouraged to use the free text boxes beneath each contributing author's name to add specific details on the author's contribution. More information is available in our guide to authors:
<https://www.embopress.org/page/journal/17574684/authorguide#authorshipguidelines>
 - In data availability statement replace the current sentence with "This study includes no data deposited in external repositories."
 - Correct the reference citation in the text and reference list. In the text a reference should be cited by author and year of publication. Include a space between a word and the opening parenthesis of the reference that follows. In the reference list, citations should be listed in alphabetical order. Where there are more than 10 authors on a paper, 10 will be listed, followed by "et al.". Please check "Author Guidelines" for more information.
<https://www.embopress.org/page/journal/17574684/authorguide#referencesformat>
 - 5) Source data: My colleague Annika Diederich requested the source data on 21.10.2025 and sent Source Data Checklist (attached to this letter). As source data are now mandatory for all manuscripts published in our journal, please provide source data underlying the figure panels, upload them as one folder per figure. Also, completed Source Data Checklist should be uploaded.
 - 6) Tables: Please move Tables 1 and 2 to the main manuscript file and place them between Main Figure Legends and Expanded View Figure Legends.
 - 7) Appendix: Please rename the file to Appendix and remove the figure titles Supplementary Figure S1. Cezard et al. etc. In the figure legends rename Figure S1 etc. to Appendix Figure S1 etc. and add a table of contents with page numbers on the title page. Please update the callouts in the main text.
 - 8) Funding: Please merge it with Acknowledgments.
 - 9) The Paper Explained: Please provide "The Paper Explained" and add it to the main manuscript text. Please check "Author

Guidelines" for more information. <https://www.embopress.org/page/journal/17574684/authorguide#researcharticleguide>

10) Synopsis: Every published paper now includes a 'Synopsis' to further enhance discoverability. Synopses are displayed on the journal webpage and are freely accessible to all readers. They include separate synopsis image and synopsis text.

- Synopsis image: Please provide a visual abstract as a high-resolution jpeg file 550 px-wide x (300-600)-px high to illustrate your article.

- Synopsis text: Please provide a short standfirst (maximum of 300 characters, including space) as well as 2-5 one sentence bullet points that summarise the paper as a .doc file. Please write the bullet points to summarise the key NEW findings. They should be designed to be complementary to the abstract - i.e. not repeat the same text. We encourage inclusion of key acronyms and quantitative information (maximum of 30 words / bullet point). Please use the passive voice.

11) As part of the EMBO Publications transparent editorial process initiative (see our Editorial at <http://embomolmed.embopress.org/content/2/9/329>), EMBO Molecular Medicine will publish online a Review Process File (RPF) to accompany accepted manuscripts. This file will be published in conjunction with your paper and will include the anonymous referee reports, your point-by-point response and all pertinent correspondence relating to the manuscript. Let us know whether you agree with the publication of the RPF and as here, if you want to remove or not any figures from it prior to publication. Please note that the Authors checklist will be published at the end of the RPF.

12) Please provide a point-by-point letter INCLUDING my comments as well as the reviewer's reports and your detailed responses (as Word file).

I look forward to reading a new revised version of your manuscript as soon as possible.

Yours sincerely,

Zeljko Durdevic

Zeljko Durdevic
Senior Editor
EMBO Molecular Medicine

*** Instructions to submit your revised manuscript ***

1) a .docx formatted version of the manuscript text (including Figure legends and tables)

2) Separate figure files*

3) supplemental information as Expanded View and/or Appendix. Please carefully check the authors guidelines for formatting Expanded view and Appendix figures and tables at <https://www.embopress.org/page/journal/17574684/authorguide#expandedview>

4) a letter INCLUDING the reviewer's reports and your detailed responses to their comments (as Word file).

5) The paper explained: EMBO Molecular Medicine articles are accompanied by a summary of the articles to emphasize the

major findings in the paper and their medical implications for the non-specialist reader. Please provide a draft summary of your article highlighting

6) Author contributions: You will be asked to provide CRediT (Contributor Role Taxonomy) terms in the submission system. These replace a narrative author contribution section in the manuscript.

7) EMBO Molecular Medicine now requires a complete author checklist (<https://www.embopress.org/page/journal/17574684/authorguide>) to be submitted with all revised manuscripts. Please use the checklist as guideline for the sort of information we need WITHIN the manuscript. The checklist should only be filled with page numbers where the information can be found. This is particularly important for animal reporting, antibody dilutions (missing) and exact values and n that should be indicated instead of a range.

8) Every published paper now includes a 'Synopsis' to further enhance discoverability. Synopses are displayed on the journal webpage and are freely accessible to all readers. They include a short stand first (maximum of 300 characters, including space) as well as 2-5 one sentence bullet points that summarise the paper. Please write the bullet points to summarise the key NEW findings. They should be designed to be complementary to the abstract - i.e. not repeat the same text. We encourage inclusion of key acronyms and quantitative information (maximum of 30 words / bullet point). Please use the passive voice. Please attach these in a separate file or send them by email, we will incorporate them accordingly.

You are also welcome to suggest a striking image or visual abstract to illustrate your article. If you do please provide a jpeg file 550 px-wide x 300-600px high.

9) A Conflict of Interest statement should be provided in the main text

10) Please note that we now mandate that all corresponding authors list an ORCID digital identifier. This takes <90 seconds to complete. We encourage all authors to supply an ORCID identifier, which will be linked to their name for unambiguous name identification.

Currently, our records indicate that the ORCID for your account is 0000-0002-5792-7742.

Please click the link below to modify this ORCID:
Link Not Available

11) Include a Reagents and Tools Table as part of the Methods section, which can be downloaded from our author guidelines (<https://www.embopress.org/page/journal/17574684/authorguide#structuredmethods>)

Graphs 800-1,200 DPI
Photos 400-800 DPI
Colour (only CMYK) 300-400 DPI"

*Additional important information regarding figures and illustrations can be found at <https://bit.ly/EMBOPressFigurePreparationGuideline>. See also figure legend preparation guidelines: <https://www.embopress.org/page/journal/17574684/authorguide#figureformat>

***** Reviewer's comments *****

Referee #1 (Remarks for Author):

Thank you for adequately addressing my comments.

Referee #3 (Remarks for Author):

The authors have convincingly addressed all points raised by the reviewers. The revisions are clear, have improved the manuscript and provided further support for the conclusions of the manuscript.

The authors addressed the remaining editorial issues.

15th Jan 2026

Dear Dr. Si-Tahar,

We are pleased to inform you that your manuscript is accepted for publication and is now being sent to our publisher to be included in the next available issue of EMBO Molecular Medicine.

You may qualify for financial assistance for your publication charges - either via a Springer Nature fully open access agreement or an EMBO initiative. Check your eligibility: <https://link.springer.com/journal/44321/how-to-publish-with-us>

Zeljko Durdevic
Senior Editor
EMBO Molecular Medicine

>>> Please note that it is EMBO Molecular Medicine policy for the transcript of the editorial process (containing referee reports and your response letter) to be published as an online supplement to each paper. If you do NOT want this, you will need to inform the Editorial Office via email immediately. More information is available here: <https://link.springer.com/partners/embo-press/editorial-policies#Peer%20review>